# Discontinuities in Sparse Mixture-of-Experts: A Measure-Stochastic Analysis

## Abstract

Sparse Mixture-of-Experts (SMoE) architectures are now widely deployed in state-of-the-art language and vision models, where conditional routing allows scaling to very large networks. However, this very Top-$k$ expert selection that enables conditional routing also renders the SMoE map inherently discontinuous. In the vicinity of these discontinuity surfaces, even inputs that are arbitrarily close may activate substantially different sets of experts resulting in significantly different outputs. In this work we give a rigorous geometric and stochastic analysis of these discontinuities. We first classify them by order, determined by the number of tied experts at a switching event. Using measure-theoretic slicing arguments, we establish asymptotic volume estimates for the thickened discontinuity surfaces, showing that lower-order discontinuity sets dominate, whereas higher-order ones occupy a vanishingly small relative volume. Next, modeling random perturbations in the input space via a diffusion process, we prove that the path eventually encounter a discontinuity, and moreover that the first hit almost surely occurs on an order-1 discontinuity with explicit finite-time probability bounds. We further derive occupation-time bounds that quantify the duration the random path spend in the neighborhoods of each discontinuity order. These theoretical results imply that inputs are more likely to lie near lower order discontinuities. Motivated by this insight, we propose a simple smoothing mechanism that can be directly applied to existing SMoEs, softly incorporating experts near discontinuities; our analysis guarantees that the added computational overhead remains small while providing localized smoothing near discontinuities, and experiments across language and vision tasks show that smoothing not only enforces continuity of the SMoE map but also enhances empirical performance.

## 1 Introduction

The Transformer architecture (Vaswani, 2017) has been successfully applied to a wide range of tasks, most notably in language (Devlin et al., 2019a; Radford et al., 2019; Hoffmann et al., 2022; Chowdhery et al., 2023), vision (Bao et al., 2022b; Dosovitskiy et al., 2021; Bao et al., 2022a; Liu et al., 2023), and other tasks (Radford et al., 2021; Chen et al., 2020; Tan & Bansal, 2019; Lu et al., 2019). However, scaling Transformers to very large models demands substantial computational resources and extended training time. To alleviate this, the Sparse Mixture-of-Experts (Jacobs et al., 1991) (SMoE) has been introduced as an architectural extension, replacing the standard feed-forward layers with sparsely activated expert modules, thereby enabling scaling while controlling computational overhead. The most common mechanism for this selection is Top-$k$ sparse gating, which has been widely adopted in large pretrained language models (Narayanan et al., 2021; Liu et al., 2024a; Shazeer et al., 2017; Rajbhandari et al., 2022) and vision models (Chen et al., 2023; Lin et al., 2024; Liu et al., 2024b).

Despite its practical success, Top-$k$ gating introduces inherent discontinuities in the input–output map of SMoEs. While sparsity is achieved by activating only $k$ experts, inputs that are nearly identical may be routed to substantially different expert sets near the switching boundaries, leading to uncontrolled variation in the outputs. Prior works (Chen et al., 2022; Wang et al., 2024; Shazeer et al., 2017) have acknowledged the existence of such discontinuities, but to the best of our knowledge, no systematic theoretical analysis of their structure and properties has been undertaken. Several recent studies have focused on mitigating the problem in practice by making MoE routing differentiable.

SMEAR (Muqeeth et al., 2023) does so by merging experts, and Soft MoE (Puigcerver et al., 2024) by mixing tokens across experts. While effective in removing hard switches, these methods compromise the causal structure required for autoregressive language modeling and are therefore limited in generation tasks. More recently, ReMoE (Wang et al., 2024) replaced Top-$k$ gating with ReLU-based gating, but this approach requires retraining the gating from scratch due to its fundamental difference from Top-$k$ gating and includes a costly initialization phase that is nearly as expensive as training a dense model. For a theoretical discussion between our paper and other continuous routing method, please refer to Section B.2 in Appendix B.

## 2 PROBLEM FORMULATION

Top-$k$ gating partitions the input space into regions with fixed active experts, and discontinuities occur where scores tie at the top-$k$ threshold. A pairwise tie between one active and one inactive expert gives an order-1 discontinuity; simultaneous ties among more experts yield higher-order ones. Though measure-zero (Proposition A.3), inputs near them are unstable since tiny perturbations can switch the active set.

We address two questions. *Geometry*: how often do different tie patterns occur, and how much space lies near their boundaries? *Dynamics*: under random perturbations, does a trajectory remain in its region or hit a boundary, and of which order?

**Contributions.** Addressing the questions above from both geometric and stochastic viewpoints, our main contributions are:

1. **Asymptotic measure.** Discontinuities are classified by order (number of tied experts). Using slicing arguments, we show $\epsilon$-thickened order-1 sets dominate while higher orders vanish in relative measure. The result extends to $\ell_\infty$-thickening, enabling efficient logit-based tests with similar bounds.

2. **Stochastic behavior.** Modeling perturbations as diffusion, we prove trajectories almost surely hit a discontinuity in finite time, with the first hit almost surely order-1. We bound occupation time in $\epsilon$-neighborhoods, showing it decreases with order in the small-$\epsilon$ regime.

3. **Smoothing mechanism.** Based on these insights, we propose a simple method that enforces continuity in Top-$k$ SMoE and is demonstrated to be effective in practice.

## 3 SPARSE MIXTURE-OF-EXPERT AND DISCONTINUITIES

### 3.1 BACKGROUND ON SPARSE MIXTURE-OF-EXPERTS

The Mixture-of-Experts (MoE) framework defines a model as a collection of expert functions combined through a gating mechanism. Formally, one considers an input space $(\mathbb{X}, \mathcal{B}(\mathbb{X}), \lambda^D)$ and an output space $(\mathbb{Y}, \mathcal{B}(\mathbb{Y}), \lambda^{D'})$. Here $\lambda^D$ and $\lambda^{D'}$ denote the Lebesgue measures on $\mathbb{R}^D$ and $\mathbb{R}^{D'}$.

A gating function $G : \mathbb{X} \to \Delta_{M-1}$ maps each input to a point on the $(M-1)$-dimensional probability simplex, assigning nonnegative weights to $M$ expert functions $\{E_i : \mathbb{X} \to \mathbb{Y}\}_{i=1}^M$. The MoE map is then given by

$$f(x) = \sum_{i=1}^{M} G_i(x) E_i(x).$$

In practice, the gating weights are often derived from a linear scoring function $z : \mathbb{X} \to \mathbb{R}^M$, where $z_i(x) = \langle W_g^{(i)}, x \rangle + b_g^{(i)}$. The most widely used variant is the Top-$k$ Sparse Mixture-of-Experts (SMoE), where only the $k$ largest scores are retained. In this case, the gate takes the form

$$G_i(x) = \frac{\exp(z_i(x)) \, \mathbf{1}_{\{i \in S_k(x)\}}}{\sum_{j \in S_k(x)} \exp(z_j(x))},$$

with $S_k(x)$ denoting the indices of the $k$ largest components of $z(x)$.

The resulting model is sparse, since only $k$ experts contribute for each input. This sparsity makes SMoEs computationally efficient and widely used in large-scale language and vision models, but also introduces discontinuities in the input–output map, which is the focus of this work.

## 3.2 DISCONTINUITIES IN SPARSE MIXTURE-OF-EXPERTS

In a Sparse Mixture-of-Experts (SMoE), the gating scores are affine functions

$$z_i(x) = \langle W_g^{(i)}, x \rangle + b_g^{(i)}, \qquad i = 1, \ldots, M.$$

For each $k$-subset $\mathbb{S} \subseteq \{1, \ldots, M\}$, we define the open cell

$$\mathcal{C}_{\mathbb{S}} = \{\, x \in \mathbb{X} : \; z_i(x) > z_j(x) \text{ for all } i \in \mathbb{S}, \; j \notin \mathbb{S} \,\},$$

which consists of all inputs where the same $k$ experts form the top set. The collection $\{\mathcal{C}_{\mathbb{S}} : |\mathbb{S}| = k\}$ partitions $\mathbb{X}$ into regions of constant active set, and the SMoE map is smooth within each region. The complement

$$\Gamma = \mathbb{X} \setminus \bigcup_{|\mathbb{S}|=k} \mathcal{C}_{\mathbb{S}}$$

is the *discontinuity set*, where ties occur between active and inactive experts. Crossing such a boundary produces a jump in the output map $f(x)$, making $\Gamma$ the source of all discontinuities in SMoEs.

However, not all discontinuities are alike. The simplest case is a pairwise tie: the $k$-th and $(k{+}1)$-th largest gate scores coincide, so that an infinitesimal change swaps membership of the Top-$k$ set. More generally, simultaneous ties among multiple scores give rise to higher-order discontinuities.

**Definition 3.1** (Order statistics of the scores)**.** Given scores $z_1(x), \ldots, z_M(x)$ at $x \in \mathbb{X}$, define the order statistics

$$z_{[1]}(x) \; \geq \; z_{[2]}(x) \; \geq \; \cdots \; \geq \; z_{[M]}(x)$$

, i.e. the sorted values of $\{z_i(x)\}_{i=1}^{M}$ in nonincreasing order.

**Definition 3.2** (Order-$n$ discontinuity)**.** Fix $1 < k < M$. A point $x \in \mathbb{X}$ is an *order-$n$ discontinuity* if there exists an index set $J = \{i_1, \ldots, i_{n+1}\} \subseteq \{1, \ldots, M\}$ such that

$$z_{i_1}(x) = z_{i_2}(x) = \cdots = z_{i_{n+1}}(x) = z_{[k]}(x) = z_{[k+1]}(x),$$

that is, $n+1$ distinct scores tie exactly at the threshold between the $k$-th and $(k{+}1)$-th largest values. For each such index set $J$, we define the corresponding discontinuity component

$$\Gamma_J^{(n)} = \{\, x \in \mathbb{X} : \; z_i(x) = z_{[k]}(x) = z_{[k+1]}(x) \; \forall i \in J \,\},$$

and the full set of order-$n$ discontinuities as

$$\Gamma^{(n)} = \bigcup_{\substack{J \subseteq \{1, \ldots, M\} \\ |J| = n+1}} \Gamma_J^{(n)}.$$

*Remark* 3.3. For readability, Definition 3.2 leaves implicit the affine inequality constraints that specify the active top-$k$ set; the equivalent, explicit formulation appears in Definition A.5. These inequalities imply $\Gamma_J^{(n)}$ is a finite union of translated affine cones contained in $(D - n)$-dimensional subspace.

Given a subset $J = \{i_1, \ldots, i_{n+1}\}$ of expert indices, we use $J$ to specify *which* experts are tied in score. Concretely, the order-$n$ tie condition

$$z_{i_1}(x) = z_{i_2}(x) = \cdots = z_{i_{n+1}}(x)$$

is equivalent to the $n$ independent equalities $z_{i_s}(x) = z_{i_1}(x)$ for $s = 2, \ldots, n + 1$, i.e.

$$\left( W_g^{(i_s)} - W_g^{(i_1)} \right)^\top x \; = \; b_g^{(i_1)} - b_g^{(i_s)}.$$

Stacking these rows defines the linear system

$$A_J x = d_J, \qquad A_J = \begin{pmatrix} (W_g^{(i_2)} - W_g^{(i_1)})^\top \\ \vdots \\ (W_g^{(i_{n+1})} - W_g^{(i_1)})^\top \end{pmatrix}, \quad d_J = \begin{pmatrix} b_g^{(i_1)} - b_g^{(i_2)} \\ \vdots \\ b_g^{(i_1)} - b_g^{(i_{n+1})} \end{pmatrix}.$$

Thus $J$ encodes the labels of the tied experts, and $A_J x = d_J$ describes the affine flat

$$S_J^{(n)} := \{x \in \mathbb{R}^D : A_J x = d_J\}$$

on which exactly those experts in $J$ have equal logits. In the later part, sometimes we write $A_J^{(n)}, d_J^{(n)}$ to denote that it corresponding to order-$n$ discontinuity.

# 4 ASYMPTOTIC MEASURE OF THICKENING DISCONTINUITIES

**Euclidean $\epsilon$-thickening of discontinuities.** Although the discontinuity set $\Gamma$ itself has Lebesgue measure zero in $\mathbb{X}$ (Proposition A.3), it is not immediately clear how large the surrounding region of "near discontinuities" can be. For instance, on the real line the rationals have measure zero, yet their $\epsilon$-neighborhood is the whole line. This motivates studying the neighborhoods of these discontinuities.

**Definition 4.1** (Euclidean $\epsilon$-thickening)**.** For a set $A \subseteq \mathbb{R}^D$ and $\epsilon > 0$, the Euclidean $\epsilon$-*thickening* of $A$ is defined as

$$T_\epsilon(A) := \{x \in \mathbb{R}^D : \operatorname{dist}(x, A) < \epsilon\},$$

where $\operatorname{dist}(x, A) := \inf_{y \in A} \|x - y\|$ is the Euclidean distance.

For brevity, we will refer to the Euclidean $\epsilon$-thickening as the $\epsilon$-thickening from now on. We write $T_\epsilon(\Gamma^{(n)})$ for the $\epsilon$-thickening of order-$n$ discontinuities. Quantifying the volume of these neighborhoods is central to understanding how much of the input space lies close to discontinuities.

In this section we investigate how much of the input space $\mathbb{X}$ is occupied by the $\epsilon$–thickening of order-$n$ discontinuity sets. Since these sets are generally unbounded, we restrict to their intersection with the ball $B^D(0, R)$ centered at the origin. Our first goal is to establish asymptotic upper bounds for their volume inside $B^D(0, R)$, together with their normalized volume, i.e. the ratio relative to $\lambda^D(B^D(0, R))$.

For brevity, all proofs in this section are deferred to Appendix A.4. We also write $\omega_d = \lambda^d(B^d(0, 1))$ for the volume of the $d$-dimensional unit ball.

**Theorem 4.2** (Asymptotic measure for $T_\epsilon(\Gamma^{(n)})$)**.** *Fix $1 \leq n < D$ and $\epsilon > 0$. Let $\bigcup_J S_J \supset \Gamma^{(n)}$ be the union of all subspaces with codimension $n$ containing the order-$n$ discontinuities, where each*

$$S_J = \{x \in \mathbb{R}^D : A_J x = d_J\}, \qquad A_J \in \mathbb{R}^{n \times D}, \ \operatorname{rank}(A_J) = n,$$

*indexed by $J$. For each $J$, define the closest point of $S_J$ to the origin by*

$$x_J^\star = A_J^\top (A_J A_J^\top)^{-1} d_J,$$

*and let $\delta_J \in \mathbb{R}^n$ be its coordinate in the normal direction to $S_J$, so that $\|\delta_J\| = \operatorname{dist}(0, S_J)$.*

*If $R > \max_J\{\|\delta_J\|\} + \epsilon$, then*

$$\lambda^D\big(T_\epsilon(\Gamma^{(n)}) \cap B^D(0, R)\big) \leq \omega_{D-n}\,\omega_n\,|J|\epsilon^n\,R^{D-n} + \sum_J O\Big((\|\delta_J\| + \epsilon)^2\,\epsilon^n\,R^{D-n-2}\Big),$$

*and*

$$\frac{\lambda^D\big(T_\epsilon(\Gamma^{(n)}) \cap B^D(0, R)\big)}{\lambda^D(B^D(0, R))} \leq \frac{\omega_{D-n}\,\omega_n}{\omega_D}\,|J|\epsilon^n R^{-n} + \sum_J O\Big((\|\delta_J\| + \epsilon)^2\,\epsilon^n\,R^{-n-2}\Big).$$

*Remark* 4.3. Theorem 4.2 shows that the thickening measure scales as $\epsilon^n R^{D-n}$, since order-$n$ discontinuities lie on codimension-$n$ flats with $\epsilon^n$ volume in normal and $R^{D-n}$ in tangential directions. After normalization, the contribution decays as $(\epsilon/R)^n$, so higher-order discontinuities vanish asymptotically.

While Theorem 4.2 shows that higher–order discontinuities thickening vanish asymptotically, it does so one order at a time. We now sharpen this by establishing asymptotic ratios between the $\epsilon$–thickenings of order-$n$ and order-$m$ discontinuities.

**Theorem 4.4** (Relative Volume of $\epsilon$–Thickenings Across Orders). *Fix integers $1 \leq m, n < D$ and $\epsilon > 0$. For each $r \in \{m, n\}$, suppose*

$$\Gamma^{(r)} \subseteq \bigcup_{J \in \mathcal{J}_r} S_J^{(r)}, \qquad S_J^{(r)} = \{x \in \mathbb{R}^D : A_J^{(r)} x = d_J^{(r)}\}, \ \ \mathrm{rank}(A_J^{(r)}) = r,$$

*with finite $\mathcal{J}_r$. Assume moreover that each slice $\Gamma_J^{(r)} := \Gamma^{(r)} \cap S_J^{(r)}$ is a (possibly unbounded) polyhedral subset of the flat $S_J^{(r)}$. Define*

$$U_r(R) := \lambda^D\big(T_\epsilon(\Gamma^{(r)}) \cap B^D(0, R)\big).$$

*For each $J \in \mathcal{J}_r$, set*

$$\alpha_{J,r} := \lim_{R \to \infty} \frac{\lambda^{D-r}(\Gamma_J^{(r)} \cap B^D(0, R))}{\omega_{D-r} R^{D-r}} \in \Big[\max_{\mathbb{S}} \tfrac{1}{2} I_{4s_{\mathbb{S},J,r}^2(1 - s_{\mathbb{S},J,r}^2)}\Big(\tfrac{d-1}{2}, \tfrac{1}{2}\Big), 1\Big],$$

*with $s_{\mathbb{S},J,r}$ defined as in Lemma A.16 and Lemma A.17.*

*Then*

$$\frac{U_n(R)}{U_m(R)} = \frac{\sum_{J \in \mathcal{J}_n} \alpha_{J,n} \, \omega_{D-n} \, \omega_n}{\sum_{J \in \mathcal{J}_m} \alpha_{J,m} \, \omega_{D-m} \, \omega_m} \left(\frac{\epsilon}{R}\right)^{n-m} \left(1 + O\left(\frac{1}{R}\right)\right).$$

*Remark* 4.5. Theorem 4.4 shows that the ratio between $\epsilon$–thickenings of order-$n$ and order-$m$ discontinuities decays as $(\epsilon/R)^{n-m}$, so higher–order sets become negligible compared to lower–order ones as $R$ grows. This scaling reflects that a codimension-$n$ flat contributes $\epsilon^n$ volume in normal directions and $R^{D-n}$ in tangential ones, with the prefactor $\frac{\omega_{D-n} \omega_n}{\omega_{D-m} \omega_m}$ giving the dimensional correction. The slice densities $\alpha_{J,r}$ measure the fraction of each tie-flat occupied by admissible regions, and Lemma A.16 with Lemma A.17 guarantees these densities are strictly positive under linear independence of the gating weights.

$\ell_{\infty,\epsilon}$**-thickening of discontinuities.** Directly checking whether $x \in \mathbb{X}$ lies within the Euclidean $\epsilon$–neighborhood of an order-$n$ discontinuity is expensive, since it requires proximity tests against all order-$n$ subspaces. We therefore introduce a more tractable $\ell_\infty$–based thickening.

**Definition 4.6** ($\ell_{\infty,\epsilon}$–thickening). Let $\Gamma \subseteq \mathbb{X}$ and let $z : \mathbb{X} \to \mathbb{R}^M$ denote the vector of gating logits. Define the $\ell_\infty$–distance from $x$ to $\Gamma$ by

$$\mathrm{dist}_\infty(x, \Gamma) := \inf_{y \in \Gamma} \|z(x) - z(y)\|_\infty.$$

The corresponding $\ell_{\infty,\epsilon}$–thickening of $\Gamma$ is

$$T_\epsilon^{(\infty)}(\Gamma) := \{x \in \mathbb{X} : \mathrm{dist}_\infty(x, \Gamma) \leq \epsilon\}.$$

Intuitively, this is the set of inputs whose gating logits lie within $\epsilon$ (in $\ell_\infty$) of a discontinuity. By Proposition A.20, it suffices to check whether some non top-$k$ logit is within $\epsilon$ of $z_{[k]}(x)$, giving an efficient proximity test directly in logit space.

**Theorem 4.7** (Relative Volume of $\ell_{\infty,\epsilon}$-thickening Across Orders). *Fix integers $1 \leq m, n < D$ and $\epsilon > 0$. For each $r \in \{m, n\}$, suppose*

$$\Gamma^{(r)} \subseteq \bigcup_{J \in \mathcal{J}_r} S_J^{(r)}, \qquad S_J^{(r)} = \{x \in \mathbb{R}^D : A_J^{(r)} x = d_J^{(r)}\}, \ \ \mathrm{rank}(A_J^{(r)}) = r,$$

*with finite $\mathcal{J}_r$. Each slice $\Gamma_J^{(r)} := \Gamma^{(r)} \cap S_J^{(r)}$ is a polyhedral subset of the flat $S_J^{(r)}$. Set*

$$U_r(R) := \lambda^D\big(T_\epsilon^{(\infty)}(\Gamma^{(r)}) \cap B^D(0, R)\big),$$

*and for each $J \in \mathcal{J}_r$ let*

$$\alpha_{J,r} := \lim_{R \to \infty} \frac{\lambda^{D-r}(\Gamma_J^{(r)} \cap B^D(0, R))}{\omega_{D-r} R^{D-r}} \in \Big[\max_{\mathbb{S}} \tfrac{1}{2} I_{4s_{\mathbb{S},J,r}^2(1 - s_{\mathbb{S},J,r}^2)}\Big(\tfrac{d-1}{2}, \tfrac{1}{2}\Big), 1\Big],$$

$$\kappa_{J,r} := \big(\det(A_J^{(r)}(A_J^{(r)})^\top)\big)^{-1/2},$$

*with $s_{\mathbb{S},J,r}$ defined as in Lemma A.16 and Lemma A.17*

*Then*

$$\frac{U_n(R)}{U_m(R)} \;=\; \frac{\sum\limits_{J\in\mathcal{J}_n} \kappa_{J,n}\,\alpha_{J,n}}{\sum\limits_{J\in\mathcal{J}_m} \kappa_{J,m}\,\alpha_{J,m}}\; \frac{\omega_{D-n}}{\omega_{D-m}}\; \left(\frac{2\epsilon}{R}\right)^{n-m}\; \left(1+O\left(\frac{1}{R}\right)\right).$$

*Remark* 4.8. Theorem 4.7 shows that the ratio between $\ell_{\infty,\epsilon}$–thickenings of order-$n$ and order-$m$ discontinuities decays as $(\epsilon/R)^{n-m}$, so higher–order sets remain negligible at large scales. Compared to the Euclidean case, the prefactor includes $\kappa_{J,r}$, reflecting the axis-aligned nature of $\ell_\infty$ tubes and their sensitivity to slice orientation. The densities $\alpha_{J,r}$ again capture the admissible fraction, while the $O(R^{-1})$ term accounts for finer geometry.

## 5    RANDOM PERTURBATION PROCESS: HITTING AND OCCUPATION TIME NEAR DISCONTINUITIES

In this section, we analyze how a random perturbation process, such as an adversarial actor making small stochastic updates, can drive $x_0$ from the open top-$k$ cell $\mathcal{C}_{\mathbb{S}}$ (the region where the active set $\mathbb{S}$ is fixed) to a discontinuity boundary. Neighborhoods of these boundaries are precisely where small changes can flip the top-$k$ active set. For simplicity, we assume a time-independent, invertible diffusion coefficient $\sigma \in \mathbb{R}^{d\times d}$, so the input evolves as the Itô diffusion

$$dx_t = \sigma\, dB_t, \qquad x_0 \in \mathcal{C}_{\mathbb{S}},$$

where $B_t$ is standard $d$-dimensional Brownian motion. Under this model we first derive explicit probabilistic bounds on the boundary hitting time. For brevity, all proofs in this section are deferred to Appendix A.5.

**Theorem 5.1** (Exit through order-1 discontinuities with hitting-time bound). *Let $x_t$ solve the diffusion process in Equation 5, with $\mathcal{C}_{\mathbb{S}}$ is the open polyhedral cell associated with the $k$-subset $\mathbb{S}$, given by*

$$\mathcal{C}_{\mathbb{S}} = \bigcap_{i\in\mathbb{S},\, j\notin\mathbb{S}} \left\{ x \in \mathbb{R}^d :\; (W_g^{(i)} - W_g^{(j)})^\top x > b_g^{(j)} - b_g^{(i)} \right\}.$$

*Denote $a^{(i,j)} := W_g^{(i)} - W_g^{(j)}$, $d^{(i,j)} := b_g^{(j)} - b_g^{(i)}$, and $c^{(i,j)} := \|\sigma^\top a^{(i,j)}\|$, and assume uniform nondegeneracy $c^{(i,j)} > 0$ for all $i,j$. Define the minimal normalized distance to the boundary by*

$$r_{\min} \;:=\; \min_{i\in\mathbb{S},\, j\notin\mathbb{S}} \frac{a^{(i,j)\top} x_0 - d^{(i,j)}}{\|\sigma^\top a^{(i,j)}\|} \;>\; 0.$$

*Let*

$$\tau_{\mathbb{S}} := \inf\{t \geq 0 :\; x_t \notin \mathcal{C}_{\mathbb{S}}\}$$

*be the exit time. Then the following hold:*

*1. (**Exit location.**) Almost surely,*

$$\mathbb{P}\big(x_{\tau_{\mathbb{S}}} \in \Gamma^{(1)}\big) = 1, \qquad \mathbb{P}\big(x_{\tau_{\mathbb{S}}} \in \Gamma^{(n)}\big) = 0 \quad \text{for all } n \geq 2,$$

*i.e. exit occurs on an order-1 discontinuity with probability one.*

*2. (**Hitting-time bound.**) For every $t > 0$,*

$$\mathbb{P}(\tau_{\mathbb{S}} \leq t) \;\geq\; 2(1 - \Phi(r_{\min}/\sqrt{t})),$$

*where $\Phi(x) = \frac{1}{\sqrt{2\pi}} \int_{-\infty}^{x} e^{-u^2/2}\, du$ is the standard normal CDF. Moreover, by continuity of the sample paths and Lemma A.25,*

$$x_{\tau_{\mathbb{S}}} \in \Gamma \quad \text{almost surely}.$$

*Remark* 5.2. Theorem 5.1 shows that higher–order discontinuities $\Gamma^{(n)}$, $n \geq 2$, are almost surely never hit by the diffusion, i.e. $\mathbb{P}(x_{\tau_{\mathbb{S}}} \in \Gamma^{(n)}) = 0$. The key proof idea is that projecting onto directions orthogonal to each discontinuity subspace yields an $n$-dimensional Brownian motion, which for $n \geq 2$ almost surely does not hit a fixed point (Lemma A.27). Hence exits occur only along order-1 boundaries corresponding to pairwise logit ties. Moreover, the bound $\mathbb{P}(\tau_{\mathbb{S}} \leq t) \geq 2(1-\Phi(r_{\min}/\sqrt{t}))$ highlights how the minimal normalized distance $r_{\min}$ governs the law of $\tau_{\mathbb{S}}$, linking separating hyperplane geometry with exit-time behavior.

In summary, order-1 discontinuities are almost surely hit in finite time (Proposition A.3), while higher orders $n \geq 2$ are not, though diffusion may still linger near them. To quantify this, we use the $\epsilon$-thickening and state the following theorem.

**Theorem 5.3** (Occupation time near order-$n$ discontinuities). *Let $x_t$ solve the diffusion equation 5 with initial condition $x_0 \in \mathcal{C}_\mathbb{S}$, where $\mathcal{C}_\mathbb{S}$ is an open top-k cell. Assume*

$$\Gamma^{(n)} \subseteq \bigcup_{J \in \mathcal{J}_n} S_J^{(n)}, \qquad S_J^{(n)} := \{x \in \mathbb{R}^D : A_J^{(n)} x = d_J^{(n)}\}, \quad \operatorname{rank} A_J^{(n)} = n.$$

*For each $J$, choose an orthonormal basis $N_J$ of $(S_J^{(n)})^\perp$ and set*

$$\Sigma_{\perp,J} := N_J^\top \Sigma N_J, \quad \lambda_{\min,J} := \lambda_{\min}(\Sigma_{\perp,J}), \quad s_J := N_J^\top y \ (y \in S_J^{(n)}), \quad \mu_J := N_J^\top x_0.$$

*Define*

$$K_{J,n} := \frac{\omega_n}{(2\pi)^{n/2}\sqrt{\det(\Sigma_{\perp,J})}}, \qquad \delta_{J,\epsilon} := \left\| \Sigma_{\perp,J}^{-1/2}(s_J - \mu_J) \right\| - \frac{\epsilon}{\sqrt{\lambda_{\min,J}}}, \qquad b_{J,\epsilon} := \frac{(\delta_{J,\epsilon})_+^2}{2}.$$

*Let*

$$A_\epsilon^{(n)}(T; \Gamma) := \int_0^T \mathbf{1}\{x_t \in T_\epsilon(\Gamma^{(n)})\} \, dt.$$

*Then, for all $T > 0$,*

$$\mathbb{E}\left[A_\epsilon^{(n)}(T; \Gamma)\right] \leq \begin{cases} \sum_J K_{J,n} \, \epsilon^n \, b_{J,\epsilon}^{1-\frac{n}{2}} \, \Gamma(n/2 - 1, b_{J,\epsilon}/T), & n > 2, \\ \sum_J K_{J,2} \, \epsilon^2 \, E_1(b_{J,\epsilon}/T), & n = 2, \\ 2\left(\sum_J K_{J,1}\right) \epsilon \sqrt{T}, & n = 1. \end{cases}$$

*where $\Gamma(\cdot, \cdot)$ is the upper incomplete gamma function and $E_1(z) = \int_z^\infty e^{-u} u^{-1} \, du$.*

*Remark* 5.4. Theorem 5.3 gives an upper bound on the expected occupation time that the diffusion $X_t$ spends inside the $\epsilon$-thickening of order-$n$ discontinuities. The leading factor $\epsilon^n$ reflects the codimension-$n$ geometry of the thickening. As $n$ increases, $\epsilon^n$ decays exponentially for $0 < \epsilon < 1$. Moreover, the sum over slices $J$ is finite, so the upper bound decreases with $n$ in the small-$\epsilon$ regime.

# 6 CONTINUITY VIA $\ell_{\infty,\epsilon}$-THICKENING LOCAL SMOOTHING

From Section 5, random perturbations in the input space almost surely intersect a discontinuity boundary, with low-order ones encountered most often. Motivated by this, we propose smoothing the SMoE map whenever the input lies in an $\ell_{\infty,\epsilon}$–thickening of a discontinuity set. Unlike Euclidean $\epsilon$–thickening, the $\ell_{\infty,\epsilon}$ version allows efficient proximity testing via gating logits, making it both theoretically justified and computationally practical.

$\ell_{\infty,\epsilon}$ **local smoothing (Figure 2 from Appendix A.2).** From Proposition A.20, we established that local smoothing within an $\ell_{\infty,\epsilon}$–thickening requires only inputs $x \in T_\epsilon^{(\infty)}(\Gamma)$ such that there exists a non top-k index $i$ with

$$0 < z_{[k]}(x) - z_i(x) < \epsilon.$$

We propose to smooth non top-$k$ logits $z_i(x)$ within the $\epsilon$-strip and discard those below it, while keeping all top-$k$ logits unchanged. The smoothing is applied uniformly, but only affects logits $z_i(x)$ that satisfy the specified inequality. A key consequence is that if $x$ lies in the $\ell_{\infty,\epsilon}$–thickening of an order-$n$ discontinuity, at most $n$ additional experts can be activated. Since the measure of higher-order $\ell_{\infty,\epsilon}$–thickenings decays rapidly (Theorem 4.7), a small $\epsilon$ ensures that the expected number of extra experts remains low.

We define the *log-smoothstep* $h : \mathbb{R} \to \mathbb{R}$ by

$$h(u) = -\infty \, \mathbf{1}_{\{u \leq 0\}} + 0 \cdot \mathbf{1}_{\{u \geq 1\}} + \log\left(\frac{u^a}{u^a + (1-u)^b}\right) \mathbf{1}_{\{0 < u < 1\}}, \qquad a, b > 0.$$

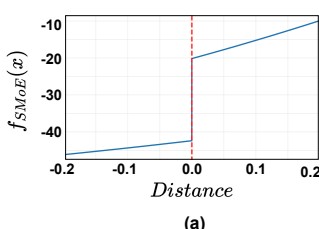 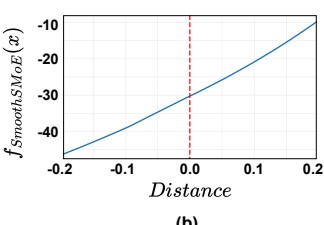 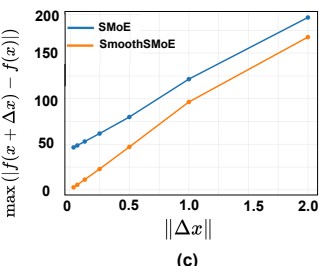

Figure 1: Effect of $\ell_{\infty,\epsilon}$ smoothing on discontinuity boundaries. (a) Standard SMoE shows a jump at the boundary. (b) SmoothSMoE, with identical weights, removes the jump and yields continuity. (c) Continuity check: maximum output difference vs. perturbation $\|\Delta x\|$. For SmoothSMoE (orange) it vanishes as $\|\Delta x\| \to 0$, while for SMoE (blue) it remains nonzero.

Given $\epsilon > 0$, we define the smoothed coefficient

$$m_i(x) \;=\; h\big((z_i(x) - z_{[k]}(x) + \epsilon)/\epsilon\big).$$

The smoothed gating logit is then defined as

$$\hat{z}_i(x) \;=\; z_i(x) + m_i(x).$$

As shown in Figure 2 from Appendix A.2, the soft margin discards logits below the cutoff, smoothly boosts those within the margin, and leaves those above the cutoff unchanged. Although $h$ is continuous, the continuity of $x \mapsto z_{[k]}(x)$ is not immediate; Proposition A.7 establishes this and hence the continuity of the smoothed SMoE.

**Boundary loss for adaptive $\epsilon$.** Choosing $\epsilon$ is nontrivial since smoothing acts in logit space. We therefore introduce a *boundary loss* that adaptively tunes $\epsilon$ under a fixed budget of extra experts. Let $\mathcal{K}$ be the average number of activated experts with threshold $\epsilon$ (top-$k$ plus those within $\epsilon$ of $z_{[k]}$), and $k^*$ the target budget. With a learning coefficient $\alpha > 0$, we define

$$\mathcal{L}_{\text{boundary}} \;=\; \alpha\,\epsilon\,\big(\mathcal{K} - k^*\big).$$

Minimizing $\mathcal{L}_{\text{boundary}}$ naturally adjusts $\epsilon$: when $\mathcal{K} > k^*$ the loss drives $\epsilon$ down, and when $\mathcal{K} < k^*$ it drives $\epsilon$ up. In practice, we set $k^* = k + 0.5$, allowing on average half an additional expert for boundary smoothing. For a geometric intuition behind our theoretical results and smoothing mechanism, please see Section B.1.

## 7 EMPIRICAL RESULTS

In this section, we empirically investigate the behaviour of the $\ell_{\infty,\epsilon}$ local smoothing method. We first demonstrate, through a small experiment, that the vanilla top-$k$ SMoE map exhibits nontrivial discontinuity, while $\ell_{\infty,\epsilon}$ local smoothing effectively enforces continuity in the SMoE map. We further show that the proposed smoothing can also yield improvements over its top-$k$ counterpart when applied to other tasks. Appendix B.4 shows how the boundary loss adapts $\epsilon$ and controls the average number of active experts. The complete experimental setup and training hyperparameters are reported in Appendix C.

### 7.1 $\ell_{\infty,\epsilon}$ LOCAL SMOOTHING VS. VANILLA SMOE NEAR DISCONTINUITY BOUNDARIES

To visualize the effect of $\ell_{\infty,\epsilon}$ local smoothing, we analyze a 4-layer SMoE pretrained on CIFAR-10 and compare it with SmoothSMoE initialized from the same weights with 32 experts and top-4 routing, isolating stochastic effects. Focusing on Layer 3, we select a random input point with a large discontinuity gap based on its orthogonal projection onto the nearest boundary, and then evaluate the model's output along the normal direction. As shown in Figure 1(a), SMoE exhibits a sharp jump,

Table 1: Perplexity (PPL) of SmoothSMoE compared to baseline models on clean and attacked WikiText-103 datasets. Means and standard deviations are computed over 3 random seeds.

| Model | WikiText-103 | | Attacked WikiText-103 | |
|---|---|---|---|---|
| | Valid PPL ↓ | Test PPL ↓ | Valid PPL ↓ | Test PPL ↓ |
| *SMoE* | $33.79 \pm 0.07$ | $35.52 \pm 0.13$ | $42.21 \pm 0.08$ | $44.18 \pm 0.12$ |
| *ReMoE* | $33.60 \pm 0.14$ | $35.35 \pm 0.12$ | $42.19 \pm 0.19$ | $44.00 \pm 0.45$ |
| SmoothSMoE | $\mathbf{32.72 \pm 0.08}$ | $\mathbf{34.35 \pm 0.22}$ | $\mathbf{40.99 \pm 0.26}$ | $\mathbf{42.85 \pm 0.29}$ |

where tiny perturbations cause large output changes, while SmoothSMoE in Figure 1(b) removes this jump and yields a continuous map. Figure 1(c) plots the maximum output difference of a fixed dimension versus perturbation magnitude $\|\Delta x\|$ along a normal direction. For SMoE (blue) the difference persists as $\|\Delta x\| \to 0$, whereas for SmoothSMoE (orange) it vanishes, confirming that smoothing restores continuity. Additional results, including visualizations for other layers, are given in Appendix B.3.

## 7.2 LANGUAGE MODELING ON WIKITEXT-103 AND ENWIKI-8

We follow Pham et al. (2024) for language modeling pretraining on WikiText-103 (Merity et al., 2017a) and EnWiki-8 (Mahoney, 2006) using a Switch Transformer (Fedus et al., 2022) with 16 experts and top-2 routing, reporting PPL on WikiText-103 and BPC on EnWiki-8. Robustness on WikiText-103 is tested by training on the clean corpus and evaluating on attacked versions (Han et al., 2024). We include ReMoE (Wang et al., 2024) as another baseline to compare against other continuous routing methods; for this baseline, we allow dense expert training for the first 2 epochs before enforcing the sparsity loss. As shown in Table 1, SmoothSMoE reduces WikiText-103 validation/test PPL from 33.79/35.52 to 32.72/34.35 (improvements of 1.07 and 1.17), and similarly lowers Attacked WikiText-103 validation/test PPL from 42.21/44.18 to 40.99/42.85 compared to SMoE. ReMoE yields slightly lower perplexity than SMoE but is consistently outperformed by SmoothSMoE, ranking second across all four metrics. On EnWiki-8 (Table 4, Appendix B), SmoothSMoE achieves 1.122 BPC vs. 1.153 for SMoE, confirming gains across both standard and robust language modeling.

Table 2: Results on GLUE benchmarks. Means and standard deviations are computed over 5 random seeds.

| Model | RTE | MRPC | COLA | QNLI | MNLI | Average |
|---|---|---|---|---|---|---|
| *SMoE* (K=16, k=2) | $73.28 \pm 1.02$ | $89.17 \pm 0.42$ | $64.25 \pm 1.49$ | $\mathbf{92.56 \pm 0.05}$ | $86.60 \pm 0.06$ | $81.17$ |
| *ReMoE* (K=16, k=2) | $73.10 \pm 0.74$ | $88.60 \pm 1.90$ | $64.9 \pm 1.2$ | $92.53 \pm 0.14$ | $86.69 \pm 0.13$ | $81.18$ |
| SmoothSMoE (K=16, k=2) | $\mathbf{73.40 \pm 0.85}$ | $\mathbf{90.15 \pm 0.60}$ | $\mathbf{65.41 \pm 0.39}$ | $92.40 \pm 0.12$ | $\mathbf{86.90 \pm 0.20}$ | $\mathbf{81.65}$ |
| *SMoE* (K=16, k=4) | $73.85 \pm 1.17$ | $89.26 \pm 1.29$ | $63.90 \pm 0.62$ | $92.20 \pm 0.14$ | $86.49 \pm 0.15$ | $81.14$ |
| *ReMoE* (K=16, k=4) | $72.20 \pm 1.35$ | $89.49 \pm 0.49$ | $\mathbf{65.07 \pm 1.61}$ | $92.51 \pm 0.03$ | $86.51 \pm 0.12$ | $81.16$ |
| SmoothSMoE (K=16, k=4) | $\mathbf{74.60 \pm 1.11}$ | $89.88 \pm 0.87$ | $64.82 \pm 0.41$ | $\mathbf{92.53 \pm 0.28}$ | $\mathbf{86.82 \pm 0.12}$ | $\mathbf{81.73}$ |

## 7.3 IMAGE CLASSIFICATION ON DOMAINBED BENCHMARK

We evaluate smoothing on vision tasks using DomainBed (Gulrajani & Lopez-Paz, 2020). Following Guo et al. (2024), GMoE (Li et al., 2023) is built from a ViT-S/16 backbone (Dosovitskiy et al., 2021) pretrained on ImageNet-1K. We add our $\ell_{\infty,\epsilon}$ local smoothing to GMoE and compare against the original across four DomainBed tasks. As shown in Table 3, SmoothGMoE achieves steady improvements over GMoE across most benchmarks, with an average gain of 0.56% and a notable 2.1% increase on TerraInc. The larger datasets show consistent improvements, suggesting that smoothing is especially effective in large-data regimes by activating extra experts near ties and stabilizing optimization.

---

[1]All baseline results in Table 3 are from Li et al. (2023), except DomainNet, which we carefully tuned and reproduced.

Table 3: Mean accuracy (%) on DomainBed with ViT-S/16. Mean and standard deviation are computed over 5 random seeds.

| Algorithms | PACS | VLCS | OfficeHome | TerraInc | DomainNet | Average |
|---|---|---|---|---|---|---|
| *GMoE*[1] | $\mathbf{87.7 \pm 0.2}$ | $79.6 \pm 0.4$ | $73.1 \pm 0.3$ | $45.4 \pm 0.3$ | $48.4 \pm 0.1$ | $66.84$ |
| SmoothGMoE | $87.6 \pm 0.32$ | $\mathbf{79.9 \pm 0.2}$ | $\mathbf{73.46 \pm 0.41}$ | $\mathbf{47.5 \pm 0.91}$ | $\mathbf{48.8 \pm 0.1}$ | $\mathbf{67.4}$ |

## 7.4 GLUE BENCHMARK: LANGUAGE INFERENCE AND CLASSIFICATION TASKS

We evaluate our smoothing mechanism on natural language understanding using five GLUE tasks (Wang et al., 2018): CoLA (Warstadt et al., 2019), MRPC (Dolan & Brockett, 2005), MNLI (Wang et al., 2018), QNLI, and RTE (Bentivogli et al., 2009). Following experiment settings in MoEfication (Zhang et al., 2022) and EMoE (Qiu et al., 2023), we augment BERT-large (Devlin et al., 2019b) by replacing one FFN layer with our MoE layer and compare against SMoE baselines, reporting validation performance. As shown in Table 2 in Appendix B, SmoothSMoE achieves higher accuracy on almost all tasks and settings, with the largest gain of $1.32\%$ on RTE. Averaged across each top-$k \in \{2, 4\}$ yields a consistent improvement of $0.25\%$–$0.42\%$, indicating that smoothing benefits SMoE models for language understanding on the GLUE benchmark.

We evaluate our smoothing mechanism on natural language understanding using five GLUE tasks (Wang et al., 2018): CoLA (Warstadt et al., 2019), MRPC (Dolan & Brockett, 2005), MNLI (Wang et al., 2018), QNLI, and RTE (Bentivogli et al., 2009). Following experiment settings in MoEfication (Zhang et al., 2022) and EMoE (Qiu et al., 2023), we augment BERT-large (Devlin et al., 2019b) by replacing one FFN layer with our MoE layer and compare against SMoE baselines, reporting validation performance. We additionally include ReMoE (Wang et al., 2024) as a continuous-routing baseline to broaden the comparison. As shown in Table 2, SmoothSMoE achieves the strongest performance across both $k=2$ and $k=4$ settings, improving over SMoE and ReMoE on nearly all tasks. The largest gain is observed on MRPC (up to $1.55\%$), and smoothing also yields consistent improvements on RTE, CoLA, and MNLI. Averaged across all GLUE tasks, SmoothSMoE improves over the next best baseline by $0.47\%$ for $k=2$ and $0.57\%$ for $k=4$, indicating that smoothing provides robust benefits for language understanding.

## 8 CONCLUSION

In this paper, we provide a theoretical investigation of discontinuities in Sparse Mixture-of-Experts from both geometric and stochastic perspectives. On the geometric side, we classify discontinuities by order and, using measure-theoretic slicing arguments, derive asymptotic volume bounds for both Euclidean $\epsilon$-thickenings and $\ell_{\infty, \epsilon}$-thickenings around these sets. On the stochastic side, we analyze the hitting times of discontinuities as well as the occupation times of a random diffusion process in their neighborhoods. Building on these insights, we propose a simple smoothing mechanism that can be applied directly to SMoEs and demonstrate its effectiveness across multiple tasks. One possible limitation of our analysis is that adversarial or structured perturbations may deviate from random diffusion, making them more challenging to study; addressing such cases remains an interesting direction for future work.

**Ethics Statement.** Given the nature of the work, we do not foresee any negative societal and ethical impacts of our work.

**Reproducibility Statement.** Source codes for our experiments are provided in the supplementary materials of the paper. The details of our experimental settings are given in Section C. All datasets used in this paper are publicly available.

**LLM usage.** In this paper, large language models (LLMs) were used solely as a tool to assist and refine the writing process. They helped with phrasing, clarity, and stylistic polishing, but all conceptual work, analyses, and conclusions were developed independently by the authors. The LLM served only to improve readability and presentation, without contributing to the research content itself

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

## A  APPENDIX

# Supplement to "Discontinuities in Sparse Mixture-of-Experts: A Measure-Stochastic Analysis"

**Table of Contents**

## A.1   MATH NOTATIONS

### Numbers and Arrays

| | |
|---|---|
| $a$ | A scalar (integer or real) |
| $\boldsymbol{a}$ | A vector |
| $\boldsymbol{A}$ | A matrix |
| $\mathbf{A}$ | A tensor |
| $\boldsymbol{I}_n$ | Identity matrix with $n$ rows and $n$ columns |
| $\boldsymbol{I}$ | Identity matrix with dimensionality implied by context |
| $\boldsymbol{e}^{(i)}$ | Standard basis vector $[0, \ldots, 0, 1, 0, \ldots, 0]$ with a 1 at position $i$ |
| $\mathrm{diag}(\boldsymbol{a})$ | A square, diagonal matrix with diagonal entries given by $\boldsymbol{a}$ |
| a | A scalar random variable |
| $\mathbf{a}$ | A vector-valued random variable |
| $\mathbf{A}$ | A matrix-valued random variable |

### Sets and Graphs

| | |
|---|---|
| $\mathbb{A}$ | A set |
| $\mathbb{R}$ | The set of real numbers |
| $\{0, 1\}$ | The set containing 0 and 1 |
| $\{0, 1, \ldots, n\}$ | The set of all integers between 0 and $n$ |
| $[a, b]$ | The real interval including $a$ and $b$ |
| $(a, b]$ | The real interval excluding $a$ but including $b$ |
| $\mathbb{A} \backslash \mathbb{B}$ | Set subtraction, i.e., the set containing the elements of $\mathbb{A}$ that are not in $\mathbb{B}$ |
| $\mathcal{G}$ | A graph |
| $Pa_{\mathcal{G}}(\mathrm{x}_i)$ | The parents of $\mathrm{x}_i$ in $\mathcal{G}$ |

### Indexing

| | |
|---|---|
| $a_i$ | Element $i$ of vector $\boldsymbol{a}$, with indexing starting at 1 |
| $a_{-i}$ | All elements of vector $\boldsymbol{a}$ except for element $i$ |
| $A_{i,j}$ | Element $i, j$ of matrix $\boldsymbol{A}$ |
| $\boldsymbol{A}_{i,:}$ | Row $i$ of matrix $\boldsymbol{A}$ |
| $\boldsymbol{A}_{:,i}$ | Column $i$ of matrix $\boldsymbol{A}$ |
| $A_{i,j,k}$ | Element $(i, j, k)$ of a 3-D tensor $\mathbf{A}$ |
| $\mathbf{A}_{:,:,i}$ | 2-D slice of a 3-D tensor |
| $\mathrm{a}_i$ | Element $i$ of the random vector $\mathbf{a}$ |

### Calculus

| | |
|---|---|
| $\dfrac{dy}{dx}$ | Derivative of $y$ with respect to $x$ |
| $\dfrac{\partial y}{\partial x}$ | Partial derivative of $y$ with respect to $x$ |
| $\nabla_{\boldsymbol{x}} y$ | Gradient of $y$ with respect to $\boldsymbol{x}$ |
| $\nabla_{\boldsymbol{X}} y$ | Matrix derivatives of $y$ with respect to $\boldsymbol{X}$ |
| $\nabla_{\mathbf{X}} y$ | Tensor containing derivatives of $y$ with respect to $\mathbf{X}$ |
| $\dfrac{\partial f}{\partial \boldsymbol{x}}$ | Jacobian matrix $\boldsymbol{J} \in \mathbb{R}^{m \times n}$ of $f : \mathbb{R}^n \to \mathbb{R}^m$ |
| $\nabla_{\boldsymbol{x}}^2 f(\boldsymbol{x})$ or $\boldsymbol{H}(f)(\boldsymbol{x})$ | The Hessian matrix of $f$ at input point $\boldsymbol{x}$ |
| $\displaystyle\int f(\boldsymbol{x})d\boldsymbol{x}$ | Definite integral over the entire domain of $\boldsymbol{x}$ |
| $\displaystyle\int_{\mathbb{S}} f(\boldsymbol{x})d\boldsymbol{x}$ | Definite integral with respect to $\boldsymbol{x}$ over the set $\mathbb{S}$ |

**Probability and Information Theory**

| | |
|---|---|
| $P(\mathrm{a})$ | A probability distribution over a discrete variable |
| $p(\mathrm{a})$ | A probability distribution over a continuous variable, or over a variable whose type has not been specified |
| $\mathrm{a} \sim P$ | Random variable a has distribution $P$ |
| $\mathbb{E}_{\mathrm{x} \sim P}[f(x)]$ or $\mathbb{E}f(x)$ | Expectation of $f(x)$ with respect to $P(\mathrm{x})$ |
| $\mathrm{Var}(f(x))$ | Variance of $f(x)$ under $P(\mathrm{x})$ |
| $\mathrm{Cov}(f(x), g(x))$ | Covariance of $f(x)$ and $g(x)$ under $P(\mathrm{x})$ |
| $H(\mathrm{x})$ | Shannon entropy of the random variable x |
| $D_{\mathrm{KL}}(P\|Q)$ | Kullback-Leibler divergence of P and Q |
| $\mathcal{N}(\boldsymbol{x}; \boldsymbol{\mu}, \boldsymbol{\Sigma})$ | Gaussian distribution over $\boldsymbol{x}$ with mean $\boldsymbol{\mu}$ and covariance $\boldsymbol{\Sigma}$ |

**Functions**

| | |
|---|---|
| $f : \mathbb{A} \to \mathbb{B}$ | The function $f$ with domain $\mathbb{A}$ and range $\mathbb{B}$ |
| $f \circ g$ | Composition of the functions $f$ and $g$ |
| $f(\boldsymbol{x}; \boldsymbol{\theta})$ | A function of $\boldsymbol{x}$ parametrized by $\boldsymbol{\theta}$. (Sometimes we write $f(\boldsymbol{x})$ and omit the argument $\boldsymbol{\theta}$ to lighten notation) |
| $\log x$ | Natural logarithm of $x$ |
| $\sigma(x)$ | Logistic sigmoid, $\dfrac{1}{1 + \exp(-x)}$ |
| $\zeta(x)$ | Softplus, $\log(1 + \exp(x))$ |
| $\|\boldsymbol{x}\|_p$ | $L^p$ norm of $\boldsymbol{x}$ |
| $\|\boldsymbol{x}\|$ | $L^2$ norm of $\boldsymbol{x}$ |
| $x^+$ | Positive part of $x$, i.e., $\max(0, x)$ |
| $\mathbf{1}_{\mathrm{condition}}$ | is 1 if the condition is true, 0 otherwise |

## A.2 MATHEMATICAL FORMULATION FOR MIXTURE OF EXPERTS

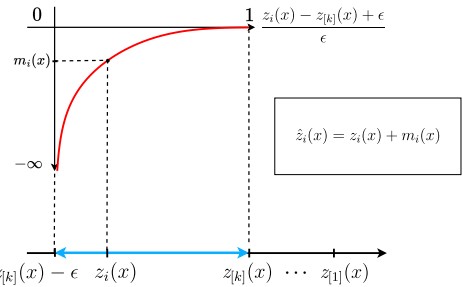

Figure 2: Illustration for gating logit smoothing within the $\ell_{\infty,\epsilon}$-thickening.

### A.2.1 MIXTURE-OF-EXPERTS

Let $\mathbb{X} = \mathbb{R}^D$ and $\mathbb{Y} = \mathbb{R}^{D'}$, each regarded as a finite-dimensional normed vector space with the Euclidean inner product. We equip them with their Borel $\sigma$-algebras $\mathcal{B}(\mathbb{X})$, $\mathcal{B}(\mathbb{Y})$, and with the standard Lebesgue measures $\lambda^D$, $\lambda^{D'}$, respectively. Then, we define the input space as $(\mathbb{X}, \mathcal{B}(\mathcal{X}), \lambda^D)$ and the output space as $(\mathbb{Y}, \mathcal{B}(\mathcal{Y}), \lambda^{D'})$.

Assume that we have $M$ experts. A gating function $G$ is a map

$$G : \mathcal{X} \to \Delta_{M-1},$$

where $\Delta_{M-1} = \{\alpha \in \mathbb{R}_{\geq 0}^M : \sum_{i=1}^M \alpha_i = 1\}$ denotes the $(M-1)$-dimensional probability simplex. For each input $x \in \mathcal{X}$, the vector $G(x) = (G_1(x), \ldots, G_M(x))$ specifies the weights assigned to the $M$ experts.

For $i = 1, \ldots, M$, each expert is given by a map

$$E_i : \mathbb{X} \to \mathbb{Y},$$

Then, we can write the Mixture-of-Experts as a function $f : \mathbb{X} \to \mathbb{Y}$ in the form

$$f(x) = \sum_{i=1}^M G_i(x) E_i(x)$$

### A.2.2 TOP-k SPARSE MIXTURE-OF-EXPERTS (SMOE)

We now state the 3 assumptions used in the proofs. They are mild and typically satisfied by pretrained SMoE models in practice; they exclude pathological corner cases and streamline the theoretical analysis.

**Assumption:**

1. The number of experts is smaller than the input dimension $(M < D)$.

2. The number of experts activated is positive and less than the full set of available expert $(1 \leq k < M)$.

3. $W_g \in \mathbb{R}^{M \times D}$ has full row rank.

*Remark* A.1. On the space $\mathbb{R}^{M \times D}$ we define the product measure $\lambda^{M \times D}$ induced by the row measures $\lambda^{W^{(i)}}$. If each $\lambda^{W^{(i)}}$ is absolutely continuous with respect to the Lebesgue measure $\lambda^D$ or has the Lebesgue measure itself, it follows that the set of weight matrices $W_g$ that are not full row rank has product $\lambda^{M \times D}$-measure zero.

Define $z : \mathcal{X} \to \mathbb{R}^M$ as the gating score function componentwise by

$$z_i(x) = \langle W_g^{(i)}, x \rangle + b_g^{(i)}, \qquad i = 1, \ldots, M,$$

so that $z(x) = (z_1(x), \ldots, z_M(x))$.

Fix $k \in \{1, \ldots, M\}$ and let $S_k(x) \subseteq \{1, \ldots, M\}$ be the indices of the $k$ largest entries of $z(x)$. The top-$k$ softmax gate $G : \mathcal{X} \to \Delta_{M-1}$ is

$$G_i(x) = \frac{\exp(z_i(x)) \, \mathbf{1}_{\{i \in S_k(x)\}}}{\sum_{j \in S_k(x)} \exp(z_j(x))}, \qquad i = 1, \ldots, M,$$

Then, the Sparse Mixture-of-Experts (SMoE) is the map $f : \mathcal{X} \to \mathcal{Y}$ defined by $f(x) = \sum_{i=1}^{M} G_i(x) E_i(x)$, where $G$ is the top-$k$ softmax gate and each expert map is $E_i : \mathcal{X} \to \mathcal{Y}$.

### A.3 DISCONTINUITIES OF TOP-K SPARSE MIXTURE-OF-EXPERTS

#### A.3.1 PARTITION INDUCED BY TOP-$k$ AFFINE GATING AND THE DISCONTINUITY SET.

Let $z_i(x) = \langle W_g^{(i)}, x \rangle + b_g^{(i)}$ for $i = 1, \ldots, M$ be the affine gatings, and fix $k \in \{1, \ldots, M\}$. For each $k$-subset $\mathbb{S} \subseteq \{1, \ldots, M\}$, define the open cell

$$\mathcal{C}_{\mathbb{S}} = \{ x \in \mathbb{X} : z_i(x) > z_j(x) \text{ for all } i \in S, \, j \notin S \}.$$

Then $\{\mathcal{C}_{\mathbb{S}}\}_{|\mathbb{S}|=k}$ is dense in $\mathbb{X}$, while the remaining points in $\mathbb{R}^D \backslash \bigcup_{|\mathbb{S}|=k} \mathcal{C}_{\mathbb{S}}$ constitute the discontinuity set of the Top-$k$ gating, which will be analyzed later.

**Proposition A.2.** $\mathcal{C}_{\mathbb{S}}$ *is a full-dimensional region in* $\mathbb{R}^D$, *i.e.* $\dim(\mathcal{C}_{\mathbb{S}}) = D$.

*Proof.* For $i \in \mathcal{S}, j \notin \mathcal{S}$, we have the following

$$z_i(x) > z_j(x) \iff (W_g^{(i)} - W_g^{(j)})^\top x > b_g^{(j)} - b_g^{(i)}.$$

Hence

$$\mathcal{C}_{\mathcal{S}} = \bigcap_{i \in \mathcal{S}, \, j \notin \mathcal{S}} \{ x \in \mathbb{R}^D : (W_g^{(i)} - W_g^{(j)})^\top x > b_g^{(j)} - b_g^{(i)} \}.$$

By Assumption 3, $(W_g^{(i)} - W_g^{(j)}) \neq 0$ for all $i \neq j$. Each inequality $z_i(x) > z_j(x)$ then defines a nontrivial open halfspace in $\mathbb{R}^D$. Their finite intersection gives $\mathcal{C}_{\mathcal{S}}$, which is an open subset of $\mathbb{R}^D$. So its affine hull equals $\mathbb{R}^D$ and

$$\dim(\mathcal{C}_{\mathcal{S}}) = \dim(\text{aff}(\mathcal{C}_{\mathcal{S}})) = D. \quad \square$$

On the relative interior $\text{relint}(\mathcal{C}_{\mathbb{S}})$ of $C_{\mathbb{S}}$, the active expert set is constant, $S_k(x) = \mathbb{S}$, and the gate is

$$G_i(x) = \frac{\exp(z_i(x)) \, \mathbf{1}_{\{i \in \mathbb{S}\}}}{\sum_{j \in \mathbb{S}} \exp(z_j(x))}.$$

For each $k$-subset $\mathbb{S}$ and $i \in \mathbb{S}, j \notin \mathbb{S}$, we define the boundary $\mathbb{F}_{\mathbb{S}, i, j}$ as follow

$$\mathbb{F}_{\mathbb{S}, i, j} = \Big\{ x \in \mathbb{X}^D : \; z_i(x) = z_j(x), \; z_i(x) \le z_\ell(x) \, \forall \ell \in \mathbb{S} \backslash \{i\}, \; z_m(x) \le z_j(x) \, \forall m \notin (\mathbb{S} \cup \{j\}) \Big\}.$$

Intuitively, this set is the boundary where the $k$-th largest score $z_i(x)$ from the active set $\mathbb{S}$ ties with the $(k+1)$-th largest score $z_j(x)$ from the inactive set, so that crossing such a boundary swaps $i$ and $j$ between active and inactive experts.

The discontinuous set of the Top-$k$ gating is the union

$$\Gamma = \bigcup_{|S|=k} \bigcup_{i \in S, \, j \notin S} \mathbb{F}_{\mathbb{S}, i, j}.$$

$\square$

**Proposition A.3.** *The discontinuous set* $\Gamma$ *has Lebesgue measure zero in* $\mathbb{R}^D$, *i.e.* $\lambda^D(\Gamma) = 0$.

*Proof.* For $i \neq j$, we define the tie set $\mathbb{H}_{ij} := \{x \in \mathbb{R}^D : z_i(x) = z_j(x)\}$ is an affine hyperplane given by

$$\{x \in \mathbb{R}^D : (W_g^{(i)} - W_g^{(j)})^\top x = b_g^{(j)} - b_g^{(i)}\},$$

with $W_g^{(i)} - W_g^{(j)} \neq 0$. Hence $\mathbb{H}_{ij}$ has Lebesgue measure zero. Each boundary piece $\mathbb{F}_{\mathbb{S},i,j}$ is a polyhedral subset of $\mathbb{H}_{ij}$, so $\lambda^D(\mathcal{F}_{\mathbb{S},i,j}) = 0$. Since

$$\Gamma = \bigcup_{|\mathbb{S}|=k} \bigcup_{i \in \mathbb{S},\, j \notin \mathbb{S}} \mathbb{F}_{\mathbb{S},i,j}$$

is a finite union of these $\mathbb{F}_{\mathbb{S},i,j}$ terms, hence, countable subadditivity gives us $\lambda^D(\Gamma) = 0$. $\qquad\square$

### A.3.2 ORDERS OF DISCONTINUITIES

Within the discontinuity set $\Gamma$ there are, in fact, different types of discontinuities. For instance, one may encounter a pairwise tie where only two scores satisfy $z_i(x) = z_j(x)$ with one index inside and one outside the top-$k$ set. Alternatively, higher-order ties may occur, such as a triple equality $z_{i'}(x) = z_{j'}(x) = z_{k'}(x)$.

To analyze these discontinuities, we classify them by order: a pairwise tie is called a order-1 discontinuity, a triple tie a order-3 discontinuity, and more generally an order-$n$ discontinuity corresponds to $n + 1$ scores becoming equal across the top-$k$ threshold.

**Definition A.4** (Order statistics of the scores). Given scores $z_1(x), \ldots, z_M(x)$ at $x \in \mathbb{X}$, define the order statistics

$$z_{[1]}(x) \geq z_{[2]}(x) \geq \cdots \geq z_{[M]}(x)$$

denote the order statistics, i.e. the sorted values of $\{z_i(x)\}_{i=1}^M$ in nonincreasing order, and ties are broken by lexical order of the original index.

**Definition A.5** (Order-$n$ discontinuity). Fix $1 < k < M$ and let the gating scores be affine maps

$$z(x) = W_g x + b_g, \qquad W_g \in \mathbb{R}^{M \times D},\ b_g \in \mathbb{R}^M,$$

with rows $a_i^\top$ and entries $b_i$, so $z_i(x) = a_i^\top x + b_i$.

A point $x \in \mathbb{X}$ is an *order-$n$ discontinuity* if there exists a tie set

$$I = \{i_1, \ldots, i_{n+1}\} \subseteq \{1, \ldots, M\}$$

such that the scores in $I$ tie exactly at the switching threshold,

$$z_{i_1}(x) = \cdots = z_{i_{n+1}}(x) = z_{[k]}(x) = z_{[k+1]}(x),$$

so that $x$ lies in the affine subspace

$$S_I = \left\{ x \in \mathbb{R}^D : (a_{i_r} - a_{i_1})^\top x = b_{i_1} - b_{i_r},\ r = 2, \ldots, n+1 \right\}.$$

At such a point, some but not all indices of $I$ belong to the Top-$k$ set $\mathbb{S}$. The corresponding discontinuity slice is the polyhedron

$$\Gamma_{I,\mathbb{S}}^{(n)} = \left\{ x \in S_I : (a_j - a_{i_1})^\top x > b_{i_1} - b_j,\ \forall j \in \mathbb{S} \setminus I;\quad (a_\ell - a_{i_1})^\top x < b_{i_1} - b_\ell,\ \forall \ell \in \mathbb{S}^{\complement} \setminus I \right\}.$$

The discontinuity component associated with $I$ is

$$\Gamma_I^{(n)} = \bigcup_{\substack{\mathbb{S} \subseteq \{1,\ldots,M\},\ |\mathbb{S}|=k \\ I \cap \mathbb{S} \neq \varnothing,\ I \cap \mathbb{S} \neq I}} \Gamma_{I,\mathbb{S}}^{(n)},$$

and the full order-$n$ discontinuity set is

$$\Gamma^{(n)} = \bigcup_{\substack{I \subseteq \{1,\ldots,M\} \\ |I|=n+1}} \Gamma_I^{(n)}.$$

**Proposition A.6** (Dimension of order-$n$ discontinuity sets). *For any tie set $I$ of size $n+1$, the associated discontinuity component $\Gamma_I^{(n)}$ lies in an affine subspace of codimension $n$. Consequently,*

$$\dim(\Gamma_I^{(n)}) = D - n.$$

*Proof.* Fix a tie set $I = \{i_1, \ldots, i_{n+1}\}$. By definition, $x \in \Gamma_I^{(n)}$ satisfies

$$(a_{i_r} - a_{i_1})^\top x = b_{i_1} - b_{i_r}, \qquad r = 2, \ldots, n+1,$$

which are $n$ linear equations.

Since the rows of $W_g$ are linearly independent, the difference vectors

$$\{\, a_{i_r} - a_{i_1} : r = 2, \ldots, n+1 \,\}$$

are also linearly independent. Thus the system has rank $n$, and the solution set $S_I$ is an affine subspace of codimension $n$.

The additional inequalities restrict $S_I$ to a polyhedral subset but do not reduce its dimension. Therefore every component $\Gamma_I^{(n)}$ has dimension $D - n$. $\square$

### A.3.3 SMOOTHED SMoE IS CONTINUOUS

In this part, we prove the fact that our Smoothed SMoE mapping is continuous under some mild assumptions. First, we assume that the base set of assumption in Section A.2.2 is satisfied. In addition, we assume that the set of expert mapping $E_i(x)$'s are continuous, which holds in practice when it is usally parameterized as an MLP network with ReLU activation.

**Proposition A.7.** *Let $\mathbb{X} = \mathbb{R}^D$ and $\mathbb{Y} = \mathbb{R}^{D'}$ be endowed with the standard Euclidean topology. Define the gating logits*

$$z_i(x) = \langle W_g^{(i)}, x \rangle + b_g^{(i)}, \qquad i = 1, \ldots, M,$$

*and the order statistics*

$$z_{[1]}(x) \geq z_{[2]}(x) \geq \cdots \geq z_{[M]}(x),$$

*with ties broken lexicographically by the original indices. Let $h : \mathbb{R} \to \mathbb{R}$ be continuous and set*

$$m_i(x) := h\left( \frac{z_i(x) - z_{[k]}(x) + \epsilon}{\epsilon} \right), \qquad \hat{z}_i(x) := z_i(x) + m_i(x).$$

*Define the gating scores and the SmoothSMoE*

$$G_i(x) := \frac{\exp(\hat{z}_i(x))}{\sum_{j=1}^M \exp(\hat{z}_j(x))}, \qquad f(x) := \sum_{i=1}^M G_i(x)\, E_i(x),$$

*where each expert map $E_i : \mathbb{X} \to \mathbb{Y}$ is continuous. Then $f$ is continuous.*

*Proof.* We write $>_{\text{lex}}$ for the strict total order on logits that respects values and breaks ties by index: $z_i(x) >_{\text{lex}} z_j(x)$ if either $z_i(x) > z_j(x)$ or $z_i(x) = z_j(x)$ and $i < j$. Thus

$$z_{[1]}(x) \; >_{\text{lex}} \; z_{[2]}(x) \; >_{\text{lex}} \; \cdots \; >_{\text{lex}} \; z_{[M]}(x).$$

Consider $z_{[k]} : \mathbb{X} \to \mathbb{R}$. Let $\mathcal{B} = \{(a - r, a + r) : a \in \mathbb{R}, r \in \mathbb{R}^{\geq 0}\}$ be a basis for the topology on $\mathbb{R}$, and fix $B = (a - r, a + r) \in \mathcal{B}$. Then

$$z_{[k]}^{-1}(B) = \{\, x \in \mathbb{X} : \ a - r < z_{[k]}(x) < a + r \,\}.$$

For any permutation $(i_1, \ldots, i_M)$ of $\{1, \ldots, M\}$,

$$U_{(i_1, \ldots, i_M)} := \{\, x \in \mathbb{X} : \ z_{i_1}(x) >_{\text{lex}} \cdots >_{\text{lex}} z_{i_M}(x) \,\}$$

is open, since it is a finite intersection of open half-spaces and subspaces. Also $\{x \in \mathbb{X} : \ a - r < z_{i_k}(x) < a + r\}$ is open because $z_{i_k}$ is affine (hence continuous). Consequently,

$$z_{[k]}^{-1}(B) = \bigcup_{(i_1, \ldots, i_M)} \left( U_{(i_1, \ldots, i_M)} \cap \{x : \ a - r < z_{i_k}(x) < a + r\} \right)$$

is a union of open sets, hence open. Since $z_{[k]}^{-1}(B)$ is open for any $B \in \mathcal{B}$, so $z_{[k]}$ is continuous (Munkres, 1997).

Since $z_{[k]}$ and each $z_i$ are continuous, the composition $m_i(x) = h\big((z_i(x) - z_{[k]}(x) + \epsilon)/\epsilon\big)$ is continuous, and so is $\hat{z}_i = z_i + m_i$. The softmax map is smooth, hence each $G_i$ is continuous. Finally, $f(x) = \sum_{i=1}^M G_i(x) E_i(x)$ is a finite sum of products of continuous functions, so $f$ is continuous. $\qquad\square$

### A.4 Asymptotic Measure of $\epsilon$-Thickened Discontinuities

In this part, we are interested in quantifying how much of the input space lies close to the discontinuities. While the discontinuity set itself has Lebesgue measure zero in the input space $\mathbb{X}$, it is not immediately clear how large the measure of an $\epsilon$-neighborhood of this set can be. For instance, on the real line the rationals form a measure-zero set, yet their closure is the entire line.

Motivated by this analogy, we now ask whether an $\epsilon$-thickening set around the discontinuities can occupy a non-negligible portion of the space. Our goal is to analyze this behavior separately for each order-$n$ discontinuity. To make this precise, we recall the classical notion of an $\epsilon$-thickening.

**Definition A.8** ($\epsilon$-thickening). For a set $A \subseteq \mathbb{R}^D$ and $\epsilon > 0$, the Euclidean $\epsilon$-*thickening* of $A$ is defined as
$$T_\epsilon(A) := \{ x \in \mathbb{R}^D : \text{dist}(x, A) < \epsilon \},$$
where $\text{dist}(x, A) := \inf_{y \in A} \|x - y\|$ is the Euclidean distance.

#### A.4.1 Base case: $\epsilon$-thickening measure of order-1 discontinuities in a bounded region

Consider the bounded ball $B_D(0, R) \subset \mathbb{X}$ of radius $R$ centered at the origin. We are interested in quantifying the asymptotic Lebesgue measure of the $\epsilon$-thickening set of the order-1 discontinuity restricted to this region, i.e.,
$$\lambda^D\big(T_\epsilon(\Gamma^{(1)}) \cap B(0, R)\big).$$
Intuitively, this corresponds to the volume of an $\epsilon$-thickening set surrounding the discontinuity facets $\Gamma^{(1)}$ within the bounded domain $B(0, R)$.

**Proposition A.9** (Measure of the $\epsilon$–thickening set of $\Gamma^{(1)}$ inside $B^D(0, R)$). *Let $\bigcup_{m=1}^M H_m \supset \Gamma^{(1)}$ be the union of all order-1 facets, where each*
$$H_m = \{x \in \mathbb{R}^D : a_m^\top x = d_m\}, \qquad a_m \neq 0,$$
*and define the $\epsilon$–thickening set (tube) of any $S \subset \mathbb{R}^D$ by*
$$T_\epsilon(S) := \{x \in \mathbb{R}^D : \text{dist}(x, S) < \epsilon\}.$$

*Write the distance from the origin to facet $m$ as $\delta_m := d_m / \|a_m\|$.*

*Let $\omega_{D-1} = \dfrac{\pi^{\frac{D-1}{2}}}{\Gamma(\frac{D+1}{2})}$ denote the volume of the unit $(D-1)$–ball. Then, for any $R > 0$:*

*For each $m = 1, \ldots, M$,*
$$\lambda^D\big(T_\epsilon(H_m) \cap B^D(0, R)\big) = \int_{\max\{-R,\, \delta_m - \epsilon\}}^{\min\{R,\, \delta_m + \epsilon\}} \omega_{D-1} \left(R^2 - u^2\right)^{\frac{D-1}{2}} du.$$

*Consequently,*
$$\lambda^D\big(T_\epsilon(\Gamma^{(1)}) \cap B^D(0, R)\big) \leq \sum_{m=1}^M \int_{\max\{-R,\, \delta_m - \epsilon\}}^{\min\{R,\, \delta_m + \epsilon\}} \omega_{D-1} \left(R^2 - u^2\right)^{\frac{D-1}{2}} du,$$

*Proof.* Fix $m$. Choose an orthonormal basis $e_1, \ldots, e_D$ such that $e_D = a_m / \|a_m\|$.

In these coordinates, every $x \in \mathbb{R}^D$ can be written as $x = (y, u)$, where $y \in \mathbb{R}^{D-1}$ lies in the subspace orthogonal to $a_m$, and $u = e_D^\top x \in \mathbb{R}$ is the coordinate in the normal direction. Then

$$H_m = \{(y, u) : u = \delta_m\}, \qquad \text{dist}((y, u), H_m) = |u - \delta_m|.$$

Let

$$E = T_\epsilon(H_m) \cap B^D(0, R) = \{(y, u) : |u - \delta_m| < \epsilon, \ \|y\|^2 + u^2 < R^2\}.$$

The measure of $E$ is

$$I = \lambda^D(E) = \int_{\mathbb{R}^D} \mathbf{1}_E(y, u) \, dy \, du.$$

By Fubini's theorem,

$$I = \int_{\mathbb{R}} \left( \int_{\mathbb{R}^{D-1}} \mathbf{1}_E(y, u) \, dy \right) du.$$

For each fixed $u$, the inner integral is the $(D-1)$–dimensional measure of the cross-section

$$\{y : \ \|y\|^2 < R^2 - u^2\} \cap \{|u - \delta_m| < \epsilon\}.$$

This is nonempty only if $|u| < R$ and $|u - \delta_m| < \epsilon$, i.e. $u \in (\max\{-R, \delta_m - \epsilon\}, \ \min\{R, \delta_m + \epsilon\})$.

Thus,

$$I = \int_{\max\{-R, \, \delta_m - \epsilon\}}^{\min\{R, \, \delta_m + \epsilon\}} \lambda^{D-1}\big(B^{D-1}(0, \sqrt{R^2 - u^2})\big) \, du.$$

Since $\lambda^{D-1}(B^{D-1}(0, r)) = \omega_{D-1} r^{D-1}$ for any radius $r > 0$, we obtain

$$I = \int_{\max\{-R, \, \delta_m - \epsilon\}}^{\min\{R, \, \delta_m + \epsilon\}} \omega_{D-1} \left(R^2 - u^2\right)^{\frac{D-1}{2}} du.$$

Finally, since

$$T_\epsilon(\Gamma^{(1)}) \subseteq \bigcup_{m=1}^M T_\epsilon(H_m),$$

subadditivity of Lebesgue measure gives the bound. $\qquad\square$

*Remark* A.10. In Proposition A.9 we adopt the convention that $\int_a^b (\cdot) = 0$ whenever $a \geq b$. This corresponds to the geometric situation where the $\epsilon$–thickening set lies entirely outside the ball, i.e. when the minimal distance from the origin to the set satisfies $\delta_m - \epsilon > R$. In that case we have $\lambda^D(T_\epsilon(H_m) \cap B^D(0, R)) = 0$.

**Proposition A.11** (Asymptotic measure of a facet's $\epsilon$–tube)**.** *Fix $\epsilon > 0$ and a facet*

$$H_m = \{x \in \mathbb{R}^D : \ a_m^\top x = d_m\}, \qquad a_m \neq 0,$$

*with signed distance $\delta_m := d_m / \|a_m\|$ from the origin. Assume $R > |\delta_m| + \epsilon$ so that the $\epsilon$–thickening slab of $H_m$ intersects the ball $B^D(0, R)$. Then:*

$$\lambda^D\big(T_\epsilon(H_m) \cap B^D(0, R)\big) = \frac{\omega_{D-1} R^D}{2} \left[ B_{\frac{(\delta_m + \epsilon)^2}{R^2}}\Big(\tfrac{1}{2}, \tfrac{D+1}{2}\Big) - \text{sgn}\Big(\tfrac{\delta_m - \epsilon}{R}\Big) B_{\frac{(\delta_m - \epsilon)^2}{R^2}}\Big(\tfrac{1}{2}, \tfrac{D+1}{2}\Big) \right],$$

*where $B_z(\alpha, \beta)$ is the incomplete beta function.*

*Dividing by the ball volume $\lambda^D(B^D(0, R)) = \omega_D R^D$, one has*

$$\frac{\lambda^D\big(T_\epsilon(H_m) \cap B^D(0, R)\big)}{\lambda^D(B^D(0, R))} = \frac{\omega_{D-1}}{2\omega_D} \left[ B_{\frac{(\delta_m + \epsilon)^2}{R^2}}\Big(\tfrac{1}{2}, \tfrac{D+1}{2}\Big) - \text{sgn}\Big(\tfrac{\delta_m - \epsilon}{R}\Big) B_{\frac{(\delta_m - \epsilon)^2}{R^2}}\Big(\tfrac{1}{2}, \tfrac{D+1}{2}\Big) \right].$$

*As $R \to \infty$ with $\delta_m, \epsilon$ fixed,*

$$\lambda^D\big(T_\epsilon(H_m) \cap B^D(0, R)\big) = 2\,\omega_{D-1}\,\epsilon\,R^{D-1} + O\Big((|\delta_m| + \epsilon)^2 \epsilon\,R^{D-3}\Big),$$

*and hence*

$$\frac{\lambda^D\big(T_\epsilon(H_m) \cap B^D(0, R)\big)}{\lambda^D(B^D(0, R))} = \frac{2\,\omega_{D-1}}{\omega_D} \frac{\epsilon}{R} + O\Big(\frac{(|\delta_m| + \epsilon)^2 \epsilon}{R^3}\Big).$$

*Proof.* From Proposition A.9, for each facet $H_m$ we have

$$\lambda^D\big(T_\epsilon(H_m) \cap B^D(0,R)\big) = \int_{\delta_m-\epsilon}^{\delta_m+\epsilon} \omega_{D-1} \, (R^2 - u^2)^{\frac{D-1}{2}} \, du,$$

whenever $R > |\delta_m| + \epsilon$ so that the integration interval lies inside $(-R, R)$.

Set $u = Rs$, so that $s \in \big[(\delta_m - \epsilon)/R, \ (\delta_m + \epsilon)/R\big]$ and $du = R\,ds$. Then

$$\lambda^D\big(T_\epsilon(H_m) \cap B^D(0,R)\big) = \omega_{D-1} R^D \int_{(\delta_m-\epsilon)/R}^{(\delta_m+\epsilon)/R} (1 - s^2)^{\frac{D-1}{2}} \, ds.$$

Let $\alpha = \frac{D+1}{2}$. Splitting the integral at $s = 0$ we obtain

$$I = \int_a^b (1 - s^2)^{\alpha-1} \, ds = \int_0^b (1 - s^2)^{\alpha-1} \, ds + \int_a^0 (1 - s^2)^{\alpha-1} \, ds,$$

where $a = (\delta_m - \epsilon)/R$ and $b = (\delta_m + \epsilon)/R$.

Using substitution $u = s^2, ds = \mathrm{sgn}(s)\frac{1}{2}u^{-1/2}du$, we obtain

$$\int_0^b (1 - s^2)^{\alpha-1} \, ds = \frac{1}{2}\,\mathrm{sgn}(b) \int_0^{b^2} u^{-\frac{1}{2}}(1 - u)^{\alpha-1} \, du = \frac{1}{2}\,\mathrm{sgn}(b)\, B_{b^2}\Big(\tfrac{1}{2}, \alpha\Big).$$

Similarly, for the second term,

$$\int_a^0 (1 - s^2)^{\alpha-1} \, ds = -\frac{1}{2}\,\mathrm{sgn}(a) \int_0^{a^2} u^{-\frac{1}{2}}(1 - u)^{\alpha-1} \, du = -\frac{1}{2}\,\mathrm{sgn}(a)\, B_{a^2}\Big(\tfrac{1}{2}, \alpha\Big).$$

Therefore

$$I = \frac{1}{2}\left[ \mathrm{sgn}(b)\, B_{b^2}\Big(\tfrac{1}{2}, \alpha\Big) - \mathrm{sgn}(a)\, B_{a^2}\Big(\tfrac{1}{2}, \alpha\Big) \right].$$

Substituting back yields the exact formula

$$\lambda^D\big(T_\epsilon(H_m) \cap B^D(0,R)\big) = \frac{\omega_{D-1}R^D}{2}\left[ B_{\frac{(\delta_m+\epsilon)^2}{R^2}}\Big(\tfrac{1}{2}, \tfrac{D+1}{2}\Big) - \mathrm{sgn}\Big(\tfrac{\delta_m-\epsilon}{R}\Big)\, B_{\frac{(\delta_m-\epsilon)^2}{R^2}}\Big(\tfrac{1}{2}, \tfrac{D+1}{2}\Big) \right].$$

Dividing by $\lambda^D(B^D(0,R)) = \omega_D R^D$ gives the normalized fraction.

For the asymptotics, put $z_\pm = ((\delta_m \pm \epsilon)^2/R^2) \to 0$ as $R \to \infty$. Using

$$B_z\Big(\tfrac{1}{2}, \alpha\Big) = \int_0^z u^{-1/2}(1 - u)^{\alpha-1} \, du = 2\,z^{1/2} + O(z^{3/2}) \quad (z \to 0),$$

we obtain

$$\mathrm{sgn}(b) B_{b^2}\Big(\tfrac{1}{2}, \alpha\Big) = 2\,\frac{\delta_m + \epsilon}{R} + O\Big(\frac{(\delta_m + \epsilon)^3}{R^3}\Big), \qquad -\mathrm{sgn}(a) B_{a^2}\Big(\tfrac{1}{2}, \alpha\Big) = -\,2\,\frac{\delta_m - \epsilon}{R} + O\Big(\frac{(\delta_m - \epsilon)^3}{R^3}\Big).$$

And

$$\lambda^D\big(T_\epsilon(H_m) \cap B^D(0,R)\big) = \frac{\omega_{D-1}R^D}{2}\left[\frac{4\epsilon}{R} + O\Big(\frac{(\delta_m + \epsilon)^3}{R^3} - \frac{(\delta_m - \epsilon)^3}{R^3}\Big)\right] = \frac{\omega_{D-1}R^D}{2}\left[\frac{4\epsilon}{R} + O\Big(\frac{(|\delta_m| + \epsilon)^2\epsilon}{R^3}\Big)\right],$$

$$= 2\,\omega_{D-1}\,\epsilon\,R^{D-1} + O\Big((|\delta_m| + \epsilon)^2\epsilon\,R^{D-3}\Big).$$

Dividing by $\omega_D R^D$ yields

$$\frac{\lambda^D\big(T_\epsilon(H_m) \cap B^D(0,R)\big)}{\lambda^D(B^D(0,R))} = \frac{2\,\omega_{D-1}}{\omega_D}\,\frac{\epsilon}{R} + O\Big(\frac{(|\delta_m| + \epsilon)^2\epsilon}{R^3}\Big)$$

as claimed. $\qquad\square$

**Corollary A.12** (Asymptotic measure of the $\epsilon$–tube of $\Gamma^{(1)}$). *Fix $\epsilon > 0$ and let $\bigcup_{m=1}^{M} H_m \supset \Gamma^{(1)}$ be the union of all order-1 facets. Then for any $R > 0$,*

$$\lambda^D\big(T_\epsilon(\Gamma^{(1)}) \cap B^D(0, R)\big) \;\leq\; \sum_{m=1}^{M} \lambda^D\big(T_\epsilon(H_m) \cap B^D(0, R)\big).$$

*In particular, if $R > |\delta_m| + \epsilon$ for each $m$, then by Proposition A.11,*

$$\lambda^D\big(T_\epsilon(\Gamma^{(1)}) \cap B^D(0, R)\big) \;\leq\; 2M\,\omega_{D-1}\,\epsilon\,R^{D-1} \;+\; O\!\left( \sum_{m=1}^{M} (|\delta_m| + \epsilon)^2 \epsilon\, R^{D-3} \right).$$

*Equivalently, dividing by $\lambda^D(B^D(0, R)) = \omega_D R^D$,*

$$\frac{\lambda^D\big(T_\epsilon(\Gamma^{(1)}) \cap B^D(0, R)\big)}{\lambda^D(B^D(0, R))} \;\leq\; \frac{2M\,\omega_{D-1}}{\omega_D}\,\frac{\epsilon}{R} \;+\; O\!\left( \sum_{m=1}^{M} \frac{(|\delta_m| + \epsilon)^2 \epsilon}{R^3} \right).$$

Corollary A.12 is not tight, as it bounds the asymptotic measure of $\Gamma^{(1)}$ using the aggregate bounds derived from the measures of the individual facets $H_m$. This section should therefore be viewed as a schematic illustration of our proof strategy rather than a final result. In the next section, we establish stronger bounds for general order-$n$ discontinuities, yielding a sharper characterization of their asymptotic measure.

### A.4.2 GENERALIZED CASE: $\epsilon$-THICKENING MEASURE OF ORDER-$n$ DISCONTINUITY IN A BOUNDED REGION

Having proved the result for order-1 discontinuity, now we aim to establish a similar result for general order-$n$ discontinuity for all $n \geq 1$.

**Upper bound on the measure of the $\epsilon$-thickening of the subspace $S_J$.**

**Proposition A.13** (Measure of the $\epsilon$–thickening set of $\Gamma^{(n)}$ inside $B^D(0, R)$). *Fix $1 \leq n < D$. Let $\bigcup_J S_J \supset \Gamma^{(n)}$ be the union of all order-$n$ subspaces containing the order-$n$ discontinuities, where each*

$$S_J = \{x \in \mathbb{R}^D : A_J x = d_J\}, \qquad A_J \in \mathbb{R}^{n \times D},\ \mathrm{rank}(A_J) = n,$$

*indexed by $J$, and let $\epsilon$–thickening set of any $S \subset \mathbb{R}^D$ be*

$$T_\epsilon(S) := \{x \in \mathbb{R}^D : \mathrm{dist}(x, S) < \epsilon\}.$$

*For each $J$, define the closest point of $S_J$ to the origin by*

$$x_J^\star = A_J^\top (A_J A_J^\top)^{-1} d_J,$$

*and let $\delta_J \in \mathbb{R}^n$ be its coordinate in the normal direction to $S_J$, so that $\|\delta_J\| = \mathrm{dist}(0, S_J)$. Choosing an orthogonal basis, any $x \in \mathbb{R}^D$ can then be written $x = (y, u)$ with $y \in \mathbb{R}^{D-n}$ tangent to $S_J$ and $u \in \mathbb{R}^n$ normal, and $S_J = \{(y, u) : u = \delta_J\}$.*

*For each $J$ and any $R > 0$,*

$$\lambda^D\big(T_\epsilon(S_J) \cap B^D(0, R)\big) = \int_{\substack{u \in \mathbb{R}^n:\\ \|u - \delta_J\| < \epsilon \\ \|u\| < R}} \omega_{D-n}\left(R^2 - \|u\|^2\right)^{\frac{D-n}{2}} du,$$

*Consequently,*

$$\lambda^D\big(T_\epsilon(\Gamma^{(n)}) \cap B^D(0, R)\big) \;\leq\; \sum_J \int_{\substack{u \in \mathbb{R}^n:\\ \|u - \delta_J\| < \epsilon \\ \|u\| < R}} \omega_{D-n}\left(R^2 - \|u\|^2\right)^{\frac{D-n}{2}} du.$$

*Proof.* Fix $J$. In the orthonormal coordinates $(y, u) \in \mathbb{R}^{D-n} \times \mathbb{R}^n$ with $S_J = \{(y, u) : u = \delta_J\}$, we have

$$E = T_\epsilon(S_J) \cap B^D(0, R) = \big\{(y, u) : \|u - \delta_J\| < \epsilon, \ \|y\|^2 + \|u\|^2 < R^2\big\}.$$

The measure of $E$ is

$$I = \lambda^D(E) = \int_{\mathbb{R}^D} \mathbf{1}_E(y, u) \, dy \, du,$$

where we decompose $x = (y, u)$ with $y \in \mathbb{R}^{D-n}$ tangent to $S_J$ and $u \in \mathbb{R}^n$ normal.

By Fubini's theorem,

$$I = \int_{\mathbb{R}^n} \left( \int_{\mathbb{R}^{D-n}} \mathbf{1}_E(y, u) \, dy \right) du.$$

For each fixed $u \in \mathbb{R}^n$, the inner integral is the $(D - n)$–dimensional measure of the cross–section

$$\{y : \ \|y\|^2 < R^2 - \|u\|^2\} \cap \{\|u - \delta_J\| < \epsilon\}.$$

This set is nonempty only if $\|u\| < R$ and $\|u - \delta_J\| < \epsilon$.

Thus,

$$I = \int_{\substack{u \in \mathbb{R}^n: \\ \|u - \delta_J\| < \epsilon, \, \|u\| < R}} \lambda^{D-n}\big(B^{D-n}(0, \sqrt{R^2 - \|u\|^2})\big) \, du.$$

Since $\lambda^{D-n}(B^{D-n}(0, r)) = \omega_{D-n} r^{D-n}$, we obtain

$$I = \int_{\substack{u \in \mathbb{R}^n: \\ \|u - \delta_J\| < \epsilon, \, \|u\| < R}} \omega_{D-n} \, (R^2 - \|u\|^2)^{\frac{D-n}{2}} \, du,$$

which yields the first identity.

The union bound directly follows from $T_\epsilon(\Gamma^{(n)}) \subseteq \bigcup_J T_\epsilon(S_J)$.

$\square$

**Proposition A.14** (Asymptotic measure for $T_\epsilon(S_J)$ and $T_\epsilon(\Gamma^{(n)})$)**.** *With the same setup as Proposition A.13, fix $1 \le n < D$ and $\epsilon > 0$.*

(i) *Single subspace $S_J$. If $R > \|\delta_J\| + \epsilon$, then*

$$\lambda^D\big(T_\epsilon(S_J) \cap B^D(0, R)\big) = \omega_{D-n} \, \omega_n \, \epsilon^n \, R^{D-n} \ + \ O\Big((\|\delta_J\| + \epsilon)^2 \, \epsilon^n \, R^{D-n-2}\Big),$$

*and consequently*

$$\frac{\lambda^D\big(T_\epsilon(S_J) \cap B^D(0, R)\big)}{\lambda^D(B^D(0, R))} = \frac{\omega_{D-n} \, \omega_n}{\omega_D} \left(\frac{\epsilon}{R}\right)^n \ + \ O\Big( \Big(\frac{\|\delta_J\| + \epsilon}{R}\Big)^2 \Big(\frac{\epsilon}{R}\Big)^n \Big).$$

(ii) *Union $\Gamma^{(n)}$. For $\Gamma^{(n)} \subset \bigcup_J S_J$, if $R > \max_J\{\|\delta_J\|\} + \epsilon$, then*

$$\lambda^D\big(T_\epsilon(\Gamma^{(n)}) \cap B^D(0, R)\big) \ \le \ \sum_J \lambda^D\big(T_\epsilon(S_J) \cap B^D(0, R)\big),$$

*so that*

$$\lambda^D\big(T_\epsilon(\Gamma^{(n)}) \cap B^D(0, R)\big) \ \le \ \omega_{D-n} \, \omega_n \, \epsilon^n \, R^{D-n} \, |J| \ + \ \sum_J O\Big((\|\delta_J\| + \epsilon)^2 \, \epsilon^n \, R^{D-n-2}\Big),$$

*and*

$$\frac{\lambda^D\big(T_\epsilon(\Gamma^{(n)}) \cap B^D(0, R)\big)}{\lambda^D(B^D(0, R))} \ \le \ \frac{\omega_{D-n} \, \omega_n}{\omega_D} \, |J| \left(\frac{\epsilon}{R}\right)^n \ + \ \sum_J O\Big( \Big(\frac{\|\delta_J\| + \epsilon}{R}\Big)^2 \Big(\frac{\epsilon}{R}\Big)^n \Big).$$

*Proof. (i) Single subspace $S_J$.* From Proposition A.13, for each $J$ we have

$$\lambda^D\big(T_\epsilon(S_J) \cap B^D(0, R)\big) = \int_{\substack{u \in \mathbb{R}^n: \\ \|u - \delta_J\| < \epsilon, \ \|u\| < R}} \omega_{D-n} \left(R^2 - \|u\|^2\right)^{\frac{D-n}{2}} du.$$

By the assumption $R > \|\delta_J\| + \epsilon$, so the $u$–region $\{u : \|u - \delta_J\| < \epsilon\}$ lies inside $\{\|u\| < R\}$. Expand for $\|u\| \ll R$:

$$\left(R^2 - \|u\|^2\right)^{\frac{D-n}{2}} = R^{D-n} \left(1 - \frac{\|u\|^2}{R^2}\right)^{\frac{D-n}{2}} = R^{D-n}\Big(1 + O(\|u\|^2/R^2)\Big).$$

Integrating over the $n$–ball $B^n(\delta_J, \epsilon)$ gives

$$\lambda^D\big(T_\epsilon(S_J) \cap B^D(0, R)\big) = \omega_{D-n} R^{D-n} \lambda^n\big(B^n(\delta_J, \epsilon)\big) + O\left(\omega_{D-n} R^{D-n-2} \int_{B^n(\delta_J, \epsilon)} \|u\|^2 \, du\right).$$

Now, the volume of the $n$–ball is explicit:

$$\lambda^n(B^n(\delta_J, \epsilon)) = \lambda^n(B^n(0, \epsilon)) = \omega_n \epsilon^n.$$

For the error term, note that for any $u \in B^n(\delta_J, \epsilon)$,

$$\|u\| \leq \|u - \delta_J\| + \|\delta_J\| \leq \epsilon + \|\delta_J\|,$$

so

$$\|u\|^2 \leq (\|\delta_J\| + \epsilon)^2.$$

Therefore

$$\int_{B^n(\delta_J, \epsilon)} \|u\|^2 \, du \leq (\|\delta_J\| + \epsilon)^2 \, \lambda^n(B^n(\delta_J, \epsilon)) = (\|\delta_J\| + \epsilon)^2 \, \omega_n \epsilon^n.$$

Substituting these into the previous expression gives

$$\lambda^D\big(T_\epsilon(S_J) \cap B^D(0, R)\big) = \omega_{D-n} \, \omega_n \, \epsilon^n \, R^{D-n} + O\Big((\|\delta_J\| + \epsilon)^2 \, \epsilon^n \, R^{D-n-2}\Big),$$

which is the claimed asymptotic expansion.

Dividing the asymptotic by $\lambda^D\big(B^D(0, R)\big) = \omega_D R^D$ gives

$$\frac{\lambda^D\big(T_\epsilon(S_J) \cap B^D(0, R)\big)}{\lambda^D\big(B^D(0, R)\big)} = \frac{\omega_{D-n} \, \omega_n}{\omega_D} \left(\frac{\epsilon}{R}\right)^n + O\Big(\left(\frac{\|\delta_J\| + \epsilon}{R}\right)^2 \left(\frac{\epsilon}{R}\right)^n\Big).$$

*(ii) Union $\Gamma^{(n)}$.* For $\Gamma^{(n)} \subset \bigcup_J S_J$, if $R > \max_J\{\|\delta_J\|\} + \epsilon$, then

$$\lambda^D\big(T_\epsilon(\Gamma^{(n)}) \cap B^D(0, R)\big) \leq \sum_J \lambda^D\big(T_\epsilon(S_J) \cap B^D(0, R)\big).$$

Applying the asymptotic expansion from part (i) to each term gives

$$\lambda^D\big(T_\epsilon(\Gamma^{(n)}) \cap B^D(0, R)\big) \leq \omega_{D-n} \, \omega_n \, \epsilon^n \, R^{D-n} \, |J| + \sum_J O\Big((\|\delta_J\| + \epsilon)^2 \, \epsilon^n \, R^{D-n-2}\Big).$$

Dividing both sides by $\lambda^D(B^D(0, R)) = \omega_D R^D$ yields

$$\frac{\lambda^D\big(T_\epsilon(\Gamma^{(n)}) \cap B^D(0, R)\big)}{\lambda^D(B^D(0, R))} \leq \frac{\omega_{D-n} \, \omega_n}{\omega_D} \, |J| \left(\frac{\epsilon}{R}\right)^n + \sum_J O\Big(\left(\frac{\|\delta_J\| + \epsilon}{R}\right)^2 \left(\frac{\epsilon}{R}\right)^n\Big).$$

$\square$

**Proposition A.15.** *With the same setup as Proposition A.13, fix $1 \leq m, n < D$, $\epsilon > 0$, and index sets $J_n, J_m$ with $m, n$ elements. Define*

$$I_n = \lambda^D\big(T_\epsilon(S_{J_n}) \cap B^D(0, R)\big) \qquad I_m = \lambda^D\big(T_\epsilon(S_{J_m}) \cap B^D(0, R)\big)$$

*Then*

$$\frac{I_n}{I_m} = \frac{\omega_{D-n}\omega_n}{\omega_{D-m}\omega_m} \left(\frac{\epsilon}{R}\right)^{n-m} \left(1 + O\big(\tfrac{(\|\delta_{J_n}\| + \|\delta_{J_m}\| + \epsilon)^2}{R^2}\big)\right).$$

*Proof.* By Proposition A.13, for any index set $J_k$ with $k$ elements:

$$I_k = \int_{\substack{u \in \mathbb{R}^k: \\ \|u - \delta_{J_k}\| < \epsilon, \ \|u\| < R}} \omega_{D-k}\big(R^2 - \|u\|^2\big)^{\frac{D-k}{2}} \, du, \qquad 1 \leq k < D,$$

when $R > \|\delta_{J_k}\| + \epsilon$.

Write $u = \delta_{J_k} + v$ with $\|v\| < \epsilon$ and set $\alpha = \frac{D-k}{2}$. Then

$$I_k = \omega_{D-k} R^{D-k} \int_{\|v\| < \epsilon} \left(1 - \frac{\|\delta_{J_k} + v\|^2}{R^2}\right)^\alpha dv.$$

On $\|v\| \leq \epsilon$ we have $t(v) := \|\delta_{J_k} + v\|^2 / R^2 \leq ((\|\delta_{J_k}\| + \epsilon)/R)^2$, hence

$$\left(1 - \frac{\|\delta_{J_k} + v\|^2}{R^2}\right)^\alpha = 1 + O\left(\frac{(\|\delta_{J_k}\| + \epsilon)^2}{R^2}\right) \qquad \text{when } \|v\| \leq \epsilon.$$

Integrating over the $k$–ball $B^k(0, \epsilon)$ gives

$$I_k = \omega_{D-k} R^{D-k} \left[\lambda^k(B^k(0, \epsilon)) + O\left(\frac{(\|\delta_{J_k}\| + \epsilon)^2}{R^2}\right) \lambda^k(B^k(0, \epsilon))\right] = \omega_{D-k}\omega_k \, \epsilon^k \, R^{D-k} \left[1 + O\big(\tfrac{(\|\delta_{J_k}\| + \epsilon)^2}{R^2}\big)\right].$$

$$(\star)$$

Apply previous Equation with $k = n$ and $k = m$:

$$I_n = \omega_{D-n}\omega_n \epsilon^n R^{D-n} \left[1 + O\big(\tfrac{(\|\delta_{J_n}\| + \epsilon)^2}{R^2}\big)\right], \qquad I_m = \omega_{D-m}\omega_m \epsilon^m R^{D-m} \left[1 + O\big(\tfrac{(\|\delta_{J_m}\| + \epsilon)^2}{R^2}\big)\right].$$

Let

$$u_n = O\left(\frac{(\|\delta_{J_n}\| + \epsilon)^2}{R^2}\right), \qquad v_m = O\left(\frac{(\|\delta_{J_m}\| + \epsilon)^2}{R^2}\right).$$

Hence

$$\frac{I_n}{I_m} = \frac{\omega_{D-n}\omega_n}{\omega_{D-m}\omega_m} \left(\frac{\epsilon}{R}\right)^{n-m} \frac{1 + u_n}{1 + v_m}.$$

We have the identity

$$\frac{1}{1 + v_m} = 1 - v_m + \frac{v_m^2}{1 + v_m} = 1 + O(v_m).$$

Therefore

$$\frac{1 + u_n}{1 + v_m} = (1 + u_n)\big(1 + O(v_m)\big) = 1 + u_n + O(v_m) + O(u_n v_m) = 1 + O\left(\frac{(\|\delta_{J_n}\| + \epsilon)^2}{R^2}\right) + O\left(\frac{(\|\delta_{J_m}\| + \epsilon)^2}{R^2}\right),$$

Consequently,

$$\frac{I_n}{I_m} = \frac{\omega_{D-n}\omega_n}{\omega_{D-m}\omega_m} \left(\frac{\epsilon}{R}\right)^{n-m} \left[1 + O\left(\frac{(\|\delta_{J_n}\| + \epsilon)^2 + (\|\delta_{J_m}\| + \epsilon)^2}{R^2}\right)\right].$$

$\square$

**Upper bound on the measure of the $\epsilon$-thickening set of the discontinuities $T_\epsilon(\Gamma^{(r)})$.**
We begin with a lemma that provides asymptotic bounds on the measure of a polyhedral set $P$ lying in a subspace of codimension $r$, defined by a system of linear inequalities. This result will later allow us to pass from the measure of the polyhedral region carved out by the top-$k$ constraints to the measure of a bounded ball in the subspace.

**Lemma A.16** (Slice density with mixed inequalities). *Let $S \subset \mathbb{R}^D$ be an affine subspace of codimension $r$ and set $d := D - r$. Let $P \subset S$ be a (nonempty) polyhedral set given by the system of linear inequalities:*

$$P = \left\{ x \in S : \; c_j^\top x < b_j \;\; (j = 1, \ldots, p), \;\; d_m^\top x > e_m \;\; (m = 1, \ldots, q) \right\}.$$

*Define the (asymptotic) slice density*

$$\alpha(P) \; := \; \lim_{R \to \infty} \frac{\lambda^d\big(P \cap B^D(0, R)\big)}{\omega_d \, R^d}.$$

*Suppose there exists $u \in Lin(S) \setminus \{0\}$ such that*

$$c_j^\top u < 0 \quad \text{for all } j = 1, \ldots, p, \qquad \text{and} \qquad d_m^\top u > 0 \quad \text{for all } m = 1, \ldots, q.$$

*Set $\widehat{u} := u/\|u\|$ and*

$$\rho_1 := \min_{1 \leq j \leq p} \{-c_j^\top \widehat{u}\}, \quad \rho_2 := \min_{1 \leq m \leq q} \{d_m^\top \widehat{u}\}, \quad \rho := \min\{\rho_1, \rho_2\} > 0, \quad L := \max\Big\{ \max_j \|c_j\|, \; \max_m \|d_m\| \Big\}.$$

*Let*

$$s \; := \; \min\Big\{ \tfrac{1}{\sqrt{2}}, \; \tfrac{\rho}{4L} \Big\} \in (0, 1/\sqrt{2}], \qquad \theta \; := \; 2 \arcsin(s) \; \in (0, \pi/2].$$

*Then $\alpha(P)$ satisfies the two–sided bounds*

$$\tfrac{1}{2} I_{4s^2(1-s^2)}\Big( \tfrac{d-1}{2}, \tfrac{1}{2} \Big) \; \leq \; \alpha(P) \; \leq \; \tfrac{1}{2},$$

*where $I_x(a, b)$ is the regularized incomplete beta function.*

*Proof.* By hypothesis, $-c_j^\top \widehat{u} > 0$ for all $j$ and $d_m^\top \widehat{u} > 0$ for all $m$, hence $\rho > 0$ is well-defined. For unit vectors $w$, the linear forms vary continuously in $w$:

$$|c_j^\top w - c_j^\top \widehat{u}| \; \leq \; 2\|c_j\| \sin\Big( \tfrac{\angle(w, \widehat{u})}{2} \Big), \qquad |d_m^\top w - d_m^\top \widehat{u}| \; \leq \; 2\|d_m\| \sin\Big( \tfrac{\angle(w, \widehat{u})}{2} \Big).$$

Set

$$s := \min\Big\{ \tfrac{1}{\sqrt{2}}, \tfrac{\rho}{4L} \Big\}, \qquad \theta := 2 \arcsin(s).$$

Then whenever $\angle(w, \widehat{u}) \leq \theta$ we have

$$c_j^\top w \leq -\rho + 2Ls \leq -\tfrac{\rho}{2}, \qquad d_m^\top w \geq \rho - 2Ls \geq \tfrac{\rho}{2}.$$

Fix $x_0 \in S$. For such $w$ and all sufficiently large $t$,

$$c_j^\top (x_0 + tw) \leq b_j, \qquad d_m^\top (x_0 + tw) \geq e_m,$$

so the ray $x_0 + tw$ eventually lies in $P$. Thus every $w$ in the spherical cap

$$\mathcal{C} := \{ w \in Lin(S) : \|w\| = 1, \; \angle(w, \widehat{u}) \leq \theta \}$$

contributes to $P$, giving for large $R$,

$$\frac{\lambda^d(P \cap B_R)}{\omega_d R^d} \; \geq \; \sigma_{d-1}(\mathcal{C}).$$

In spherical coordinates, using the result from (Li, 2011), the cap area ratio is

$$\sigma_{d-1}(\mathcal{C}) = \frac{1}{2} I_{\sin^2 \theta}\Big( \tfrac{d-1}{2}, \tfrac{1}{2} \Big).$$

Since $\sin^2\theta = \sin^2(2\arcsin s) = 4s^2(1-s^2)$, this yields the explicit lower bound

$$\alpha(P) \;\geq\; \tfrac{1}{2} I_{4s^2(1-s^2)}\left(\tfrac{d-1}{2}, \tfrac{1}{2}\right) \;>\; 0.$$

*Upper bound.* The feasible cone $\{c_j^\top w < 0,\ d_m^\top w > 0\}$ is an intersection of hemispheres. Any such intersection is contained in some hemisphere, so its normalized measure cannot exceed that of a hemisphere:

$$\alpha(P) \;=\; \sigma_{d-1}(C_\infty \cap S^{d-1}) \;\leq\; \tfrac{1}{2}.$$

This proves the claimed two-sided bounds. $\qquad\square$

To invoke Lemma A.16 in the Top-$k$ setting, we first show that the system of linear inequalities induced by the Top-$k$ constraints indeed satisfies the hypothesis of Lemma A.16.

**Lemma A.17** (Top-$k$ slices satisfy Lemma A.16). *Assume affine scores $z_i(x) = a_i^\top x + b_i$ with $\{a_i\}_{i=1}^M$ linearly independent. Fix an order-$r$ tie set $J = \{i_1, \ldots, i_{r+1}\}$ and let*

$$S := S_J^{(r)} = \{x \in \mathbb{R}^D : A_J x = d_J\}, \qquad V := \mathrm{Lin}(S) = \ker A_J.$$

*For any admissible top-k index set $\mathbb{S} \subset \{1, \ldots, M\}$, define the polyhedral slice*

$$\Gamma_{J,\mathbb{S}}^{(r)} = \Big\{ x \in S : (a_j - a_{i_1})^\top x > b_{i_1} - b_j \ \forall j \in \mathbb{S} \setminus J, \quad (a_m - a_{i_1})^\top x < b_{i_1} - b_m \ \forall m \in \mathbb{S}^\complement \setminus J \Big\}.$$

*Then $\Gamma_{J,\mathbb{S}}^{(r)}$ satisfies the condition of Lemma A.16.*

*In particular, writing $c_j := a_j - a_{i_1}$ for $j \in \mathbb{S} \setminus J$ and $d_m := a_m - a_{i_1}$ for $m \in \mathbb{S}^\complement \setminus J$, there exists $u \in V \setminus \{0\}$ such that*

$$c_j^\top u < 0 \quad \forall j, \qquad d_m^\top u > 0 \quad \forall m,$$

*Proof.* Work inside $S$ and write each $x \in S$ as $x = x_0 + v$ with $v \in V$ (for an arbitrary $x_0 \in S$). Only the $V$–components of normals matter, so define the orthogonal projection $\Pi_V : \mathbb{R}^D \to V$ and set

$$n_j := \Pi_V(a_j - a_{i_1}) \in V \quad (j \in \mathbb{S} \setminus J), \qquad m_\ell := \Pi_V(a_\ell - a_{i_1}) \in V \quad (\ell \in \mathbb{S}^\complement \setminus J).$$

For $x = x_0 + v$ we have for all index $\star$:

$$(a_\star - a_{i_1})^\top x = (a_\star - a_{i_1})^\top x_0 + \Pi_V(a_\star - a_{i_1})^\top v$$

so the slice inequalities reduce on $V$ to

$$n_j^\top v > \beta_j \ (j \in \mathbb{S} \setminus J), \qquad m_\ell^\top v < \gamma_\ell \ (\ell \in \mathbb{S}^\complement \setminus J),$$

where

$$\beta_j := (b_{i_1} - b_j) - (a_j - a_{i_1})^\top x_0, \qquad \gamma_\ell := (b_{i_1} - b_\ell) - (a_\ell - a_{i_1})^\top x_0.$$

**Step 1 (Nondegeneracy of projected normals).** We claim $n_j \neq 0$ for all $j \in \mathbb{S} \setminus J$ and $m_\ell \neq 0$ for all $\ell \in \mathbb{S}^\complement \setminus J$. If, say, $n_j = 0$, then $a_j - a_{i_1} \in V^\perp = \mathrm{row}(A_J) = \mathrm{span}\{a_{i_s} - a_{i_1} : s = 2, \ldots, r+1\}$, yielding a nontrivial linear dependence among $\{a_{i_1}, \ldots, a_{i_{r+1}}, a_j\}$, contradicting the independence of $\{a_i\}_{i=1}^M$. The same argument applies to each $m_\ell$.

**Step 2 (A single cone collecting all signs).** Introduce the finitely generated cone

$$K := \mathrm{Cone}\Big( \{-n_j : j \in \mathbb{S} \setminus J\} \cup \{m_\ell : \ell \in \mathbb{S}^\complement \setminus J\} \Big) \subset V.$$

We show $K$ is pointed. If $K$ contained a line, then there exist nonzero coefficients $\alpha_j, \beta_\ell \geq 0$, not all zero, such that

$$\sum_\ell \beta_\ell m_\ell - \sum_j \alpha_j n_j \;=\; 0.$$

Lift this identity back to the original normals: since $n_j = \Pi_V(a_j - a_{i_1})$ and $m_\ell = \Pi_V(a_\ell - a_{i_1})$, we get

$$\Pi_V\Big( \sum_\ell \beta_\ell(a_\ell - a_{i_1}) - \sum_j \alpha_j(a_j - a_{i_1}) \Big) = 0,$$

hence the bracketed vector lies in $V^\perp = \text{row}(A_J) = \text{span}\{a_{i_s} - a_{i_1}\}_{s=2}^{r+1}$. Therefore there exist coefficients $\gamma_s$ such that

$$\sum_\ell \beta_\ell(a_\ell - a_{i_1}) - \sum_j \alpha_j(a_j - a_{i_1}) = \sum_{s=2}^{r+1} \gamma_s(a_{i_s} - a_{i_1}).$$

Rearranging terms gives a nontrivial linear dependence among distinct vectors from $\{a_i\}$:

$$\sum_\ell \beta_\ell a_\ell - \sum_j \alpha_j a_j - \sum_{s=2}^{r+1} \gamma_s a_{i_s} - \Big( \sum_\ell \beta_\ell - \sum_j \alpha_j - \sum_{s=2}^{r+1} \gamma_s \Big)a_{i_1} = 0,$$

with coefficients not all zero (since some $\alpha$ or $\beta$ is nonzero). This contradicts the linear independence of $\{a_i\}$. Hence $K$ is pointed.

**Step 3 (Strict separating functional).** Because $K$ is a pointed polyhedral cone, its polar $K^\circ = \{u \in V : \langle g, u \rangle \le 0 \,\forall g \in K\}$ has nonempty interior. Equivalently, there exists $u \in V \setminus \{0\}$ such that

$$\langle g, u \rangle < 0 \quad \text{for every generator } g \in \{-n_j\} \cup \{m_\ell\}.$$

Unpacking the generators, we have

$$\langle -n_j, u \rangle < 0 \Rightarrow \langle n_j, u \rangle > 0 \quad \forall j, \qquad \langle m_\ell, u \rangle < 0 \quad \forall \ell.$$

**Step 4 (Sign alignment with Lemma A.16).** Define $u' := -u \in V \setminus \{0\}$. Then

$$\langle n_j, u' \rangle = -\langle n_j, u \rangle < 0 \quad \forall j, \qquad \langle m_\ell, u' \rangle = -\langle m_\ell, u \rangle > 0 \quad \forall \ell.$$

Recalling $c_j = a_j - a_{i_1}$ and $d_m = a_m - a_{i_1}$, and that only their $V$–components act on $V$, we obtain

$$c_j^\top u' = n_j^\top u' < 0 \quad (\forall j \in \mathbb{S} \setminus J), \qquad d_m^\top u' = m_\ell^\top u' > 0 \quad (\forall m \in \mathbb{S}^{\complement} \setminus J).$$

**Conclusion.** We have constructed $u' \in V \setminus \{0\}$ satisfying the mixed strict sign conditions required by Lemma A.16. Therefore that lemma applies to the slice $\Gamma_{J,\mathbb{S}}^{(r)}$. $\qquad\square$

Building on Lemma A.16 and Lemma A.17, we conclude that each Top-$k$ slice $\Gamma_{J,\mathbb{S}}^{(r)} \subset S_J^{(r)}$ has positive slice density

$$\alpha(\Gamma_{J,\mathbb{S}}^{(r)}) \in \left[ \tfrac{1}{2} I_{4s_{\mathbb{S},J,r}^2(1-s_{\mathbb{S},J,r}^2)}\Big(\tfrac{d-1}{2}, \tfrac{1}{2}\Big), \tfrac{1}{2} \right],$$

where $d = D - r$ and $I_x(a,b)$ is the regularized incomplete beta function. Since, for fixed $J$, the order-$r$ slice is a finite union $\Gamma_J^{(r)} = \bigcup_{\mathbb{S}} \Gamma_{J,\mathbb{S}}^{(r)}$, its density

$$\alpha_{J,r} := \lim_{R \to \infty} \frac{\lambda^{D-r}\big(\Gamma_J^{(r)} \cap B^D(0,R)\big)}{\omega_{D-r}R^{D-r}}$$

is strictly positive and satisfies the trivial bounds

$$\alpha_{J,r} \in \left[ \max_{\mathbb{S}} \tfrac{1}{2} I_{4s_{\mathbb{S},J,r}^2(1-s_{\mathbb{S},J,r}^2)}\Big(\tfrac{d-1}{2}, \tfrac{1}{2}\Big), 1 \right].$$

We now establish asymptotic bounds for the ratio of $\epsilon$–thickenings of discontinuity sets of different orders. The argument proceeds in 4 steps:

1. Relate the measure of the $\epsilon$–thickening $\lambda^D\big(T_\epsilon(\Gamma_J^{(r)}) \cap B_R\big)$ to the base measure of the slice $\lambda^d\big(\Gamma_J^{(r)} \cap B_R\big)$.

2. Derive the asymptotics of a single thickened slice using definition of $\alpha_{J,r}$:

$$\lambda^D\big(T_\epsilon(\Gamma_J^{(r)}) \cap B^D(0,R)\big) = \omega_{D-r}\,\omega_r\,\alpha_{J,r}\,\epsilon^r\,R^{D-r} + O(\epsilon^r R^{D-r-1}).$$

3. Estimate overlaps between distinct thickenings $T_\epsilon(\Gamma_J^{(r)})$ and $T_\epsilon(\Gamma_{J'}^{(r)})$ for $J \neq J'$, showing they are bounded by

$$O(\epsilon^{r+1} R^{D-r-1}).$$

4. Assemble the contributions of all slices $J \in \mathcal{J}_r$ to obtain

$$U_r(R) = \lambda^D\big(T_\epsilon(\Gamma^{(r)}) \cap B^D(0,R)\big),$$

and then compare the cases $r = n$ and $r = m$ to deduce the asymptotic ratio

$$\frac{U_n(R)}{U_m(R)}.$$

We are now ready to state and prove the main theorem.

**Theorem A.18** (Ratio of $\epsilon$-thickening of order-$n$ discontinuity vs. $\epsilon$-thickening of order-$m$ discontinuity). *Fix integers $1 \leq m, n < D$ and $\epsilon > 0$. For each $r \in \{m, n\}$, suppose*

$$\Gamma^{(r)} \subseteq \bigcup_{J \in \mathcal{J}_r} S_J^{(r)}, \qquad S_J^{(r)} = \{x \in \mathbb{R}^D : A_J^{(r)} x = d_J^{(r)}\}, \ \ \mathrm{rank}(A_J^{(r)}) = r,$$

*with finite $\mathcal{J}_r$. Assume moreover that each slice $\Gamma_J^{(r)} := \Gamma^{(r)} \cap S_J^{(r)}$ is a (possibly unbounded) polyhedral subset of the flat $S_J^{(r)}$. Define*

$$U_r(R) := \lambda^D\big(T_\epsilon(\Gamma^{(r)}) \cap B^D(0,R)\big), \qquad \omega_d := \lambda^d\big(B^d(0,1)\big).$$

*For each $J \in \mathcal{J}_r$, set*

$$\alpha_{J,r} := \lim_{R \to \infty} \frac{\lambda^{D-r}\big(\Gamma_J^{(r)} \cap B^D(0,R)\big)}{\omega_{D-r}\,R^{D-r}} \in \Big[\ \max_{\mathbb{S}}\ \tfrac{1}{2}\,I_{4s_{\mathbb{S},J,r}^2(1-s_{\mathbb{S},J,r}^2)}\big(\tfrac{d-1}{2}, \tfrac{1}{2}\big),\ 1\Big],$$

*with $s_{\mathbb{S},J,r}$ defined as in Lemma A.16 and Lemma A.17.*

*Then*

$$U_r(R) = \omega_{D-r}\,\omega_r\Big(\sum_{J \in \mathcal{J}_r} \alpha_{J,r}\Big)\,\epsilon^r\,R^{D-r}\ +\ O(\epsilon^r R^{D-r-1}),$$

*and*

$$\frac{U_n(R)}{U_m(R)}\ =\ \frac{\sum_{J \in \mathcal{J}_n} \alpha_{J,n}}{\sum_{J \in \mathcal{J}_m} \alpha_{J,m}}\,\frac{\omega_{D-n}\,\omega_n}{\omega_{D-m}\,\omega_m}\,\Big(\frac{\epsilon}{R}\Big)^{n-m}\Big(1 + O\Big(\frac{1}{R}\Big)\Big).$$

*Proof.* We write $B_R := B^D(0,R)$ and $d := D - r$ when considering a fixed order $r$.

**Step 1 (relation between thickening and polyhedral slice).** Fix a codimension-$r$ flat $S \subset \mathbb{R}^D$ and a measurable $P \subset S$. Choose an orthogonal decomposition $\mathbb{R}^D = S \oplus S^\perp$ and write $x = (y, u)$ with $y \in S$, $u \in S^\perp$. Then

$$T_\epsilon(P) \cap B_R = \Big\{(y,u) :\ y \in P,\ \|u\| < \epsilon,\ \|y\|^2 + \|u\|^2 < R^2\Big\}.$$

Fubini theorem gives us the identity

$$\lambda^D\big(T_\epsilon(P) \cap B_R\big) = \int_{y \in P} \lambda^r\Big(B^r\big(0, \rho_R(y)\big)\Big)\,d\lambda^d(y) = \int_{y \in P} \omega_r\,\rho_R(y)^r\,d\lambda^d(y), \tag{1}$$

where $\rho_R(y) := \min\{\epsilon, \sqrt{R^2 - \|y\|^2}\} \in [0, \epsilon]$.

Split the base $P$ into the interior band $I_R := \{y : \|y\| \leq R - \epsilon\}$ and the boundary band $B_R^\partial := \{y : R - \epsilon < \|y\| < R\}$. On $I_R$ we have $\rho_R(y) = \epsilon$; on $B_R^\partial$ we only know $0 \leq \rho_R(y) \leq \epsilon$. Thus

$$\omega_r\,\epsilon^r\,\lambda^d\big(P \cap B_{R-\epsilon}\big)\ \leq\ \lambda^D\big(T_\epsilon(P) \cap B_R\big)\ \leq\ \omega_r\,\epsilon^r\,\lambda^d\big(P \cap B_R\big). \tag{2}$$

The $d$-volume of the annulus $B_R \setminus B_{R-\epsilon}$ is $\lambda^d(B_R \setminus B_{R-\epsilon}) \leq d\,\omega_d\,R^{d-1}\epsilon$, so subtracting the bounds in equation 2 yields the explicit error

$$\left| \lambda^D\big(T_\epsilon(P) \cap B_R\big) - \omega_r\,\epsilon^r\,\lambda^d\big(P \cap B_R\big) \right| \;\leq\; d\,\omega_d\,\omega_r\,\epsilon^{r+1}\,R^{d-1}. \tag{3}$$

**Step 2 (asymptotics of one thickened polyhedral slice).** By definition of $\alpha_{J,r}$, we obtain:

$$\lambda^d\big(\Gamma_J^{(r)} \cap B_R\big) = \alpha_{J,r}\,\omega_d\,R^d + O(R^{d-1}). \tag{4}$$

For fixed $r$ and $J \in \mathcal{J}_r$, put $S := S_J^{(r)}$, $P := \Gamma_J^{(r)}$, and $d = D - r$. Combining equation 3 and equation 4 yields

$$\lambda^D\big(T_\epsilon(\Gamma_J^{(r)}) \cap B_R\big) = \omega_r\,\epsilon^r\Big(\alpha_{J,r}\,\omega_{D-r}\,R^{D-r} + O(R^{D-r-1})\Big) + O(\epsilon^{r+1}R^{D-r-1}),$$

i.e.

$$\lambda^D\big(T_\epsilon(\Gamma_J^{(r)}) \cap B_R\big) = \omega_{D-r}\,\omega_r\,\alpha_{J,r}\,\epsilon^r\,R^{D-r} \;+\; O(\epsilon^r R^{D-r-1}), \tag{5}$$

with the $O(\cdot)$ uniform over $J \in \mathcal{J}_r$ (finite family).

**Step 3 (overlap estimate between slices).** Let $J \neq J'$. Since $S_J^{(r)}$ and $S_{J'}^{(r)}$ are distinct codimension-$r$ flats, their intersection $L := S_J^{(r)} \cap S_{J'}^{(r)}$ (if nonempty) has codimension at least $r + 1$. There exists a constant $c = c(D, \{S_J^{(r)}\})$ such that

$$T_\epsilon(S_J^{(r)}) \cap T_\epsilon(S_{J'}^{(r)}) \;\subset\; T_{c\epsilon}(L)$$

(geometrically: the distance to $L$ is bounded by a fixed multiple of the sum of distances to $S_J^{(r)}$ and $S_{J'}^{(r)}$, with the constant depending only on the angle between the two flats; a finite family gives a uniform $c$). Hence, by the single-flat tube estimate (Proposition A.14),

$$\lambda^D\big(T_\epsilon(S_J^{(r)}) \cap T_\epsilon(S_{J'}^{(r)}) \cap B_R\big) \;\leq\; C_{D,r}\,\epsilon^{r+1}\,R^{D-r-1}.$$

Since $\Gamma_J^{(r)} \subset S_J^{(r)}$, the same bound holds with $T_\epsilon(\Gamma^{(r)})$ in place of $T_\epsilon(S^{(r)})$. Summing over the finitely many pairs,

$$\left| \lambda^D\Big(\bigcup_J T_\epsilon(\Gamma_J^{(r)}) \cap B_R\Big) - \sum_J \lambda^D\big(T_\epsilon(\Gamma_J^{(r)}) \cap B_R\big) \right| \;\leq\; C'_{D,r}\,\epsilon^{r+1}\,R^{D-r-1}. \tag{6}$$

(Higher-order intersections are even smaller-codimension $\geq r + 2$-and are absorbed into the same bound.)

**Step 4 (Measure ratio across thickened different orders).** Because $T_\epsilon(\Gamma^{(r)}) = \bigcup_{J \in \mathcal{J}_r} T_\epsilon(\Gamma_J^{(r)})$, combining equation 5 over $J$ with equation 6 gives

$$U_r(R) = \omega_{D-r}\,\omega_r\Big( \sum_{J \in \mathcal{J}_r} \alpha_{J,r} \Big)\epsilon^r\,R^{D-r} \;+\; O(\epsilon^r R^{D-r-1}). \tag{7}$$

Apply equation 7 with $r = n$ and $r = m$ with the similar asymptotic division argument as in Proposition A.15:

$$\frac{U_n(R)}{U_m(R)} = \frac{\omega_{D-n}\omega_n\big(\sum_{J \in \mathcal{J}_n} \alpha_{J,n}\big)\epsilon^n R^{D-n}\big(1 + O(R^{-1})\big)}{\omega_{D-m}\omega_m\big(\sum_{J \in \mathcal{J}_m} \alpha_{J,m}\big)\epsilon^m R^{D-m}\big(1 + O(R^{-1})\big)}$$

$$= \frac{\sum_{J \in \mathcal{J}_n} \alpha_{J,n}}{\sum_{J \in \mathcal{J}_m} \alpha_{J,m}}\,\frac{\omega_{D-n}\,\omega_n}{\omega_{D-m}\,\omega_m}\,\left(\frac{\epsilon}{R}\right)^{n-m}\big(1 + O(R^{-1})\big).$$

$\square$

Building on Lemma A.16, we also establish an asymptotic ratio for $\ell_\infty$–tubes around discontinuity slices of different orders. The proof follows the same multi–step strategy as before, adapted to the $\ell_\infty$ geometry:

1. Derive the fiber decomposition of a slice $\Gamma_J^{(r)} \subset S_J^{(r)}$ in the subspace $S_J^{(r)}$.

2. Establish explicit two–sided bounds for the measure of the $\ell_\infty$–tube $\lambda^D\big(T_\epsilon^{(\infty)}(\Gamma_J^{(r)}) \cap B_R\big)$ in terms of the subspace volume $\lambda^d(\Gamma_J^{(r)} \cap B^D(0,R))$.

3. Reduce to base volumes in the subspace by evaluating $\lambda^d(\Gamma_J^{(r)} \cap B^D(0,R))$ and derive the asymptotic expansion of $\lambda^D(T_\epsilon^{(\infty)}(\Gamma_J^{(r)}) \cap B^D(0,R))$.

4. Control overlaps between distinct tubes $T_\epsilon^{(\infty)}(\Gamma_J^{(r)})$ and $T_\epsilon^{(\infty)}(\Gamma_{J'}^{(r)})$ for $J \neq J'$, showing their contribution is $O(\epsilon^{r+1}R^{D-r-1})$.

5. Derive the asymptotic measure of the union $\bigcup_{J \in \mathcal{J}_r} T_\epsilon^{(\infty)}(\Gamma_J^{(r)})$ for fixed $r$, and then compare $U_n(R)$ and $U_m(R)$ to obtain the asymptotic ratio

$$\frac{U_n(R)}{U_m(R)}.$$

**Theorem A.19** (Weighted union–$\ell_\infty$ tube ratio for orders $n$ vs. $m$)**.** *Fix integers $1 \leq m,n < D$ and $\epsilon > 0$. For each $r \in \{m,n\}$, suppose*

$$\Gamma^{(r)} \subseteq \bigcup_{J \in \mathcal{J}_r} S_J^{(r)}, \qquad S_J^{(r)} = \{x \in \mathbb{R}^D : A_J^{(r)}x = d_J^{(r)}\}, \ \ \mathrm{rank}(A_J^{(r)}) = r,$$

*with finite $\mathcal{J}_r$. Assume moreover that each slice $\Gamma_J^{(r)} := \Gamma^{(r)} \cap S_J^{(r)}$ is a (possibly unbounded) polyhedral subset of the flat $S_J^{(r)}$. Define the $\ell_\infty$–tube around $S_J^{(r)}$ by*

$$T_\epsilon^{(\infty)}(S_J^{(r)}) := \{x \in \mathbb{R}^D : \|A_J^{(r)}x - d_J^{(r)}\|_\infty \leq \epsilon\}, \quad T_\epsilon^{(\infty)}(\Gamma_J^{(r)}) := \{x : \mathrm{dist}_\infty(x, \Gamma_J^{(r)}) \leq \epsilon\},$$

*where $\mathrm{dist}_\infty(x,\Gamma) := \inf_{y \in \Gamma} \|A_J^{(r)}x - A_J^{(r)}y\|_\infty$ (so the normal thickening is measured via $A_J^{(r)}$). Set*

$$U_r(R) := \lambda^D\big(T_\epsilon^{(\infty)}(\Gamma^{(r)}) \cap B^D(0,R)\big), \qquad \omega_d := \lambda^d\big(B^d(0,1)\big),$$

*and for each $J \in \mathcal{J}_r$ let*

$$\alpha_{J,r} := \lim_{R \to \infty} \frac{\lambda^{D-r}\big(\Gamma_J^{(r)} \cap B^D(0,R)\big)}{\omega_{D-r} R^{D-r}} \in \Big[\max_{\mathbb{S}} \tfrac{1}{2} I_{4s_{\mathbb{S},J,r}^2(1-s_{\mathbb{S},J,r}^2)}\Big(\tfrac{d-1}{2}, \tfrac{1}{2}\Big), \ 1\Big],$$

$$\kappa_{J,r} := \big(\det(A_J^{(r)}(A_J^{(r)})^\top)\big)^{-1/2},$$

*with $s_{\mathbb{S},J,r}$ defined as in Lemma A.16 and Lemma A.17*

*Then*

$$\frac{U_n(R)}{U_m(R)} = \frac{\displaystyle\sum_{J \in \mathcal{J}_n} \kappa_{J,n}\,\alpha_{J,n}}{\displaystyle\sum_{J \in \mathcal{J}_m} \kappa_{J,m}\,\alpha_{J,m}} \frac{\omega_{D-n}}{\omega_{D-m}} \left(\frac{2\epsilon}{R}\right)^{n-m} \left(1 + O\Big(\frac{1}{R}\Big)\right).$$

*Proof.* Fix $r \in \{m,n\}$ and abbreviate $d := D - r$, $B_R := B^D(0,R)$. We prove

$$U_r(R) = \omega_{D-r}\Big(\sum_{J \in \mathcal{J}_r} \kappa_{J,r}\alpha_{J,r}\Big)(2\epsilon)^r R^{D-r} + O(\epsilon^{r+1}R^{D-r-1}), \tag{8}$$

which yields the ratio in the statement after applying it with $r = n$ and $r = m$.

**Step 1 (fiber decomposition of a slice in the subspace).** Fix one slice index $J$ and write $S := S_J^{(r)} = \{x : Ax = d\}$ with $\mathrm{rank}(A) = r$. Let $V := \ker A$ and $V^\perp = \mathrm{row}(A)$. Choose an orthonormal basis $N \in \mathbb{R}^{D \times r}$ for $V^\perp$ and complete with an orthonormal basis for $V$ so that every $x \in \mathbb{R}^D$ decomposes uniquely as $x = y + Nz$ with $y \in S$ and $z \in \mathbb{R}^r$. Then for $y \in S$ we have $Ay = d$, hence

$$Ax - d = A(y + Nz) - d = ANz.$$

Because $N$ has orthonormal columns, $AN \in \mathbb{R}^{r \times r}$ is invertible and

$$|\det(AN)| = \sqrt{\det(AA^\top)}.$$

Define

$$\kappa_{J,r} := \big(\det(AA^\top)\big)^{-1/2} = \frac{1}{|\det(AN)|}.$$

The $\ell_\infty$-tube fiber over any base point $y \in S$ is the linear preimage

$$\{z \in \mathbb{R}^r : \|AN\,z\|_\infty \le \epsilon\} = (AN)^{-1}\big([-\epsilon,\epsilon]^r\big),$$

whose $r$-volume equals

$$\lambda^r\big((AN)^{-1}([-\epsilon,\epsilon]^r)\big) = \frac{\lambda^r([-\epsilon,\epsilon]^r)}{|\det(AN)|} = \kappa_{J,r}\,(2\epsilon)^r.$$

claimed *Size of the fiber in the ambient norm.* Since $\|w\|_2 \le \sqrt{r}\,\|w\|_\infty$ for $w \in \mathbb{R}^r$, any $z$ in the fiber satisfies

$$\|z\| = \|Nz\| \le \|(AN)^{-1}\|_2\,\|ANz\|_2 \le \|(AN)^{-1}\|_2\,\sqrt{r}\,\epsilon.$$

Set the slice-dependent constant

$$C_J := \sqrt{r}\,\|(AN)^{-1}\|_2.$$

Then every point $y + Nz$ in the fiber over $y$ lies within ambient distance $\le C_J\epsilon$ of $y$.

**Step 2 (two–sided bounds for $\ell_\infty$–tubes in terms of subspace volumes).** By Fubini in the orthogonal splitting $\mathbb{R}^D = S \oplus V^\perp$,

$$\lambda^D\big(T_\epsilon^{(\infty)}(\Gamma_J^{(r)}) \cap B_R\big) = \int_{y \in \Gamma_J^{(r)}} \lambda^r\Big(\big\{z : \|ANz\|_\infty \le \epsilon,\ \|y + Nz\| \le R\big\}\Big)\,d\lambda^d(y).$$

Let

$$I_R := \{y \in \Gamma_J^{(r)} : \|y\| \le R - C_J\epsilon\}, \qquad B_R^\partial := \{y \in \Gamma_J^{(r)} : R - C_J\epsilon < \|y\| < R\}.$$

For $y \in I_R$, the entire *full* fiber fits in $B_R$ (triangle inequality), so its $r$-volume equals $\kappa_{J,r}(2\epsilon)^r$. For $y \in B_R^\partial$, the fiber volume is bounded above by the full fiber volume. Therefore,

$$\kappa_{J,r}(2\epsilon)^r\,\lambda^d(I_R) \le \lambda^D\big(T_\epsilon^{(\infty)}(\Gamma_J^{(r)}) \cap B_R\big) \le \kappa_{J,r}(2\epsilon)^r\,\lambda^d(I_R) + \kappa_{J,r}(2\epsilon)^r\,\lambda^d(B_R^\partial). \quad (9)$$

Since $I_R \cup B_R^\partial = \Gamma_J^{(r)} \cap B_R$ and $I_R = \Gamma_J^{(r)} \cap B_{R-C_J\epsilon}$, we can rewrite equation 9 as the *two-sided inequality*

$$\kappa_{J,r}(2\epsilon)^r\,\lambda^d\big(\Gamma_J^{(r)} \cap B_{R-C_J\epsilon}\big) \le \lambda^D\big(T_\epsilon^{(\infty)}(\Gamma_J^{(r)}) \cap B_R\big) \le \kappa_{J,r}(2\epsilon)^r\,\lambda^d\big(\Gamma_J^{(r)} \cap B_R\big). \quad (10)$$

**Step 3 (reduce to base volumes and apply polyhedral asymptotics).** The difference between the upper and lower terms in equation 10 is supported on the base annulus of thickness $C_J\epsilon$ in $S$:

$$\lambda^d\big(B_R \setminus B_{R-C_J\epsilon}\big) = \omega_d\big(R^d - (R - C_J\epsilon)^d\big) \le d\,\omega_d\,R^{d-1}\,C_J\epsilon.$$

Multiplying by the constant fiber volume $\kappa_{J,r}(2\epsilon)^r$ gives

$$\left| \lambda^D\big(T_\epsilon^{(\infty)}(\Gamma_J^{(r)}) \cap B_R\big) - \kappa_{J,r}(2\epsilon)^r\,\lambda^d\big(\Gamma_J^{(r)} \cap B_R\big) \right| \le d\,\omega_d\,\kappa_{J,r}\,C_J\,(2\epsilon)^r\,\epsilon\,R^{d-1}. \quad (11)$$

In particular,

$$\lambda^D\big(T_\epsilon^{(\infty)}(\Gamma_J^{(r)}) \cap B_R\big) = \kappa_{J,r}(2\epsilon)^r\,\lambda^d\big(\Gamma_J^{(r)} \cap B_R\big) + O\big(\epsilon^{r+1}R^{d-1}\big),$$

where the big–$O$ constant may depend on $J$ through $\kappa_{J,r}$ and $C_J$.

From the Equation:

$$\alpha_{J,r} := \lim_{R \to \infty} \frac{\lambda^{D-r}\big(\Gamma_J^{(r)} \cap B^D(0,R)\big)}{\omega_{D-r}\,R^{D-r}},$$

we obtain:

$$\lambda^d\big(\Gamma_J^{(r)} \cap B_R\big) = \alpha_{J,r}\,\omega_d\,R^d \;+\; O(R^{d-1}). \tag{12}$$

Combining equation 11 and equation 12 yields

$$\lambda^D\big(T_\epsilon^{(\infty)}(\Gamma_J^{(r)}) \cap B_R\big) = \kappa_{J,r}\,\alpha_{J,r}\,\omega_d\,(2\epsilon)^r\,R^d \;+\; O(\epsilon^{r+1}R^{d-1}). \tag{13}$$

Since $\mathcal{J}_r$ is finite, we can take the $O(\cdot)$ uniform in $J$ by enlarging the implicit constant to the maximum over $J$.

**Step 4 (control overlaps between different $\ell_\infty$–tubes).** Fix $J \neq J'$ and set $S := S_J^{(r)}$, $S' := S_{J'}^{(r)}$, and $L := S \cap S'$. Let $V_L := \{u \in \mathbb{R}^D : A_J u = 0,\ A_{J'} u = 0\}$ be the direction space of $L$, and let $N := V_L^\perp$ (so every $x \in \mathbb{R}^D$ decomposes uniquely as $x = y + v$ with $y \in L$, $v \in N$). Define the linear map

$$T : N \longrightarrow \mathbb{R}^r \times \mathbb{R}^r, \qquad T(v) := \big(A_J v,\ A_{J'} v\big).$$

If $T(v) = (0,0)$ then $A_J v = A_{J'} v = 0$, so $v \in V_L$. Since also $v \in N = V_L^\perp$, we get $v = 0$. Thus $T$ is injective on the finite-dimensional space $N$; hence there exists $c_0 > 0$ (such as $c_0 = 1/\sigma_{\min}(T)$) with

$$\|v\| \;\leq\; c_0\,\|T(v)\|_2 \;=\; c_0\left(\|A_J v\|_2^2 + \|A_{J'} v\|_2^2\right)^{1/2} \qquad \forall v \in N. \tag{14}$$

Now take any $x \in T_\epsilon^{(\infty)}(S) \cap T_\epsilon^{(\infty)}(S')$. Write $x = y + v$ with $y \in L$, $v \in N$. Because $A_J y = d_J$ and $A_{J'} y = d_{J'}$, we have

$$A_J v = A_J x - d_J, \qquad A_{J'} v = A_{J'} x - d_{J'}.$$

Using $\|w\|_2 \leq \sqrt{r}\,\|w\|_\infty$ in $\mathbb{R}^r$,

$$\|A_J v\|_2 \leq \sqrt{r}\,\|A_J x - d_J\|_\infty \leq \sqrt{r}\,\epsilon, \qquad \|A_{J'} v\|_2 \leq \sqrt{r}\,\epsilon.$$

Plugging into equation 14 gives

$$\mathrm{dist}(x, L) = \|v\| \;\leq\; c_0\,\sqrt{(\sqrt{r}\,\epsilon)^2 + (\sqrt{r}\,\epsilon)^2} = c_0\,\sqrt{2r}\,\epsilon \;=:\; c\epsilon.$$

Therefore we have the set inclusion

$$T_\epsilon^{(\infty)}(S) \cap T_\epsilon^{(\infty)}(S') \;\subset\; T_{c\epsilon}^{(2)}(L), \tag{15}$$

where $T_{c\epsilon}^{(2)}(L)$ denotes the *Euclidean* tube of radius $c\epsilon$ around $L$, and $c = c(J, J') := c_0\sqrt{2r}$ depends only on the pair $(J, J')$.

Since $L$ has codimension at least $r + 1$, the Euclidean tube estimate (Proposition A.14) yields

$$\lambda^D\big(T_{c\epsilon}^{(2)}(L) \cap B_R\big) \;\leq\; C\,\epsilon^{r+1}\,R^{D-r-1}$$

for some constant $C = C(D, r, \{S, S'\})$. By equation 15, the same bound holds for $\lambda^D\big(T_\epsilon^{(\infty)}(S) \cap T_\epsilon^{(\infty)}(S') \cap B_R\big)$. Because $\Gamma_J^{(r)} \subset S$ and $\Gamma_{J'}^{(r)} \subset S'$, intersecting with the slices can only decrease the measure; hence

$$\lambda^D\big(T_\epsilon^{(\infty)}(\Gamma_J^{(r)}) \cap T_\epsilon^{(\infty)}(\Gamma_{J'}^{(r)}) \cap B_R\big) \;\leq\; C\,\epsilon^{r+1}\,R^{D-r-1}.$$

Summing this over the finitely many unordered pairs $(J, J')$ and applying inclusion–exclusion truncated at first order gives

$$\left|\lambda^D\Big(\bigcup_J T_\epsilon^{(\infty)}(\Gamma_J^{(r)}) \cap B_R\Big) - \sum_J \lambda^D\big(T_\epsilon^{(\infty)}(\Gamma_J^{(r)}) \cap B_R\big)\right| \;\leq\; C'\,\epsilon^{r+1}\,R^{D-r-1}, \tag{16}$$

with $C'$ depending only on $(D, r)$ and the finite family $\{S_J^{(r)}\}_{J \in \mathcal{J}_r}$.

**Step 5 (union asymptotics and ratio for orders $n$ vs. $m$).** Summing equation 13 over $J \in \mathcal{J}_r$ and invoking equation 16 gives

$$U_r(R) = \omega_{D-r}\Big(\sum_{J \in \mathcal{J}_r} \kappa_{J,r}\alpha_{J,r}\Big)(2\epsilon)^r\,R^{D-r} \;+\; O(\epsilon^{r+1}R^{D-r-1}),$$

which is equation 8.

Applying equation 8 with $r = n$ and $r = m$ and dividing using the same argument as Proposition A.15 yields

$$\frac{U_n(R)}{U_m(R)} = \frac{\sum_{J \in \mathcal{J}_n} \kappa_{J,n} \alpha_{J,n}}{\sum_{J \in \mathcal{J}_m} \kappa_{J,m} \alpha_{J,m}} \frac{\omega_{D-n}}{\omega_{D-m}} \left(\frac{2\epsilon}{R}\right)^{n-m} \left(1 + O\left(\frac{1}{R}\right)\right),$$

as claimed. □

**Proposition A.20** (Characterization of $\ell_\infty$–thickening). *An input $x$ belongs to $T_\epsilon^{(\infty)}(\Gamma)$ if and only if there exists an index $i$ such that $0 \leq z_{[k]}(x) - z_i(x) < \epsilon$. In other words, at least one non top-$k$ logit lies within $\epsilon$ of the $k$-th logit $z_{[k]}(x)$.*

*Proof.* ($\Rightarrow$) Assume $x \in T_\epsilon^{(\infty)}(\Gamma)$. By the definition of $T_\epsilon^{(\infty)}(\Gamma)$, there exist a tie set $J$ and a top-$k$ active set $\mathbb{S}$ such that $J \setminus \mathbb{S} \neq \varnothing$. Let $i \in J \setminus \mathbb{S}$. Then $0 \leq z_{[k]}(x) - z_i(x) < \epsilon$.

($\Leftarrow$) Assume there exists $i$ with $0 \leq z_{[k]}(x) - z_i(x) < \epsilon$. Let $J = \{[k], i\}$. Then $x \in T_\epsilon^{(\infty)}(\Gamma_J^{(2)}) \subseteq T_\epsilon^{(\infty)}(\Gamma)$. □

### A.5    HITTING AND OCCUPATION TIME NEAR DISCONTINUITIES

Suppose we wish to study an adversarial process that drives the input $x_0 \in \mathcal{C}_\mathbb{S}$ toward a discontinuity boundary. We model this process by the stochastic differential equation

$$dx_t = \gamma(t, x_t) \, dt + \sigma(t, x_t) \, dB_t,$$

where $B_t$ is a standard $n$-dimensional Brownian motion. The drift term $\gamma(t, x)$ represents the adversarial drive, while the diffusion term $\sigma(t, x)$ models uncertainty and random perturbations. Such noise may arise from stochastic gradient descent when the adversarial direction is estimated from minibatches, from measurement errors in the input, or from inherent randomness injected into the system.

#### A.5.1    RANDOMLY PERTURBED DIFFUSION PROCESS IS GUARANTEED TO HIT THE TOP-K CELL BOUNDARY

We consider the stochastic dynamic that consist only of the diffusion term. In this case, the evolution of the system is driven purely by random perturbations. For simplicity, we assume that the diffusion coefficient is time-independent, i.e., $\sigma(t, x_t) = \sigma$ for all $t$, with invertible $\sigma \in \mathbb{R}^{d \times d}$. Then $x_t$ is an Itô process with initial condition $\boldsymbol{x}_0 \in \mathcal{C}_\mathbb{S}$ satisfying

$$dx_t = \sigma \, dB_t.$$

A key step in our analysis is to understand the hitting time of such processes against linear boundaries. The Proposition A.21 is a classical result that provides a probabilistic bound for the hitting time, and it will later be applied to establish the exit-time behavior from the polyhedral cell $\mathcal{C}_\mathbb{S}$.

**Proposition A.21** (Probabilistic bound for the hitting time). *Let $Y_t = Y_0 + c\,\widetilde{B}_t$ with $Y_0 > 0$ and $c > 0$, and define*

$$\tau := \inf\{t \geq 0 : Y_t \leq 0\}.$$

*Then, for every $t > 0$,*

$$\mathbb{P}(\tau \leq t) = 2\left(1 - \Phi\left(\frac{Y_0}{c\sqrt{t}}\right)\right),$$

*and hence for any $\delta \in (0, 1)$,*

$$\mathbb{P}\left(\tau \leq \left(\frac{Y_0}{c\,q_\delta}\right)^2\right) = 1 - \delta, \qquad q_\delta := \Phi^{-1}\left(\frac{1+\delta}{2}\right),$$

*where $\Phi(x) = \frac{1}{\sqrt{2\pi}} \int_{-\infty}^x e^{-u^2/2} \, du$ is the standard normal cumulative distribution function.*

*Proof.* Write $Y_t = Y_0 + c\widetilde{B}_t$ with $Y_0 > 0$. Then

$$\tau = \inf\{t \geq 0 : Y_t \leq 0\} = \inf\{t \geq 0 : \widetilde{B}_t \leq -Y_0/c\}.$$

For standard Brownian motion $\widetilde{B}_t$, the reflection principle gives

$$\mathbb{P}\left(\min_{0 \leq s \leq t} \widetilde{B}_s \leq -a\right) = 2\mathbb{P}(\widetilde{B}_t \leq -a) = 2\left(1 - \Phi(a/\sqrt{t})\right), \quad a > 0.$$

Where the second equality terms from the fact that $\widetilde{B}_t \sim \mathcal{N}(0, t)$.

Applying the equality with $a = Y_0/c$ yields

$$\mathbb{P}(\tau \leq t) = 2\left(1 - \Phi\left(\frac{Y_0}{c\sqrt{t}}\right)\right).$$

Let $q_\delta = \Phi^{-1}\left(\frac{1+\delta}{2}\right)$. Setting $t_\delta = \left(\frac{Y_0}{cq_\delta}\right)^2$ gives $\Phi\left(\frac{Y_0}{c\sqrt{t_\delta}}\right) = \frac{1+\delta}{2}$, hence

$$\mathbb{P}(\tau \leq t_\delta) = 2(1 - \frac{1+\delta}{2}) = 1 - \delta.$$

$\square$

The above proposition shows that for a one-dimensional diffusion of the form $Y_t = Y_0 + c\widetilde{B}_t$, the first hitting time of zero admits an explicit probabilistic bound. In our multidimensional setting, each face of the polyhedral cell $\mathcal{C}_\mathbb{S}$ is described by a linear inequality $a^{(i,j)\top}x > d^{(i,j)}$, and projecting the diffusion $x_t$ onto the normal direction $a^{(i,j)}$ reduces the problem to exactly this one-dimensional case. Applying Proposition A.21 to all such faces yields the following bound for the exit time from $\mathcal{C}_\mathbb{S}$.

**Theorem A.22** (Probabilistic bound of the cell boundary hitting time). *Assume $x_t$ follows the diffusion equation $dx_t = \sigma\, dB_t$ with $\sigma \in \mathbb{R}^{d \times d}$ and initial condition $x_0 \in \mathcal{C}_\mathbb{S}$, the open polyhedral cell associated with the $k$-subset $\mathbb{S}$,*

$$\mathcal{C}_\mathbb{S} = \bigcap_{i \in \mathbb{S}, \, j \notin \mathbb{S}} \left\{ x \in \mathbb{R}^d : (W_g^{(i)} - W_g^{(j)})^\top x > b_g^{(j)} - b_g^{(i)} \right\}.$$

*Denote $a^{(i,j)} := W_g^{(i)} - W_g^{(j)}$, $d^{(i,j)} := b_g^{(j)} - b_g^{(i)}$, and $c^{(i,j)} := \|\sigma^\top a^{(i,j)}\|$, and assume uniform nondegeneracy $c^{(i,j)} > 0$ for all $i, j$. Define*

$$r_{\min} := \min_{i \in \mathbb{S}, \, j \notin \mathbb{S}} \frac{a^{(i,j)\top}x_0 - d^{(i,j)}}{\|\sigma^\top a^{(i,j)}\|} > 0.$$

*The hitting time of $\mathcal{C}_\mathbb{S}$ is*

$$\tau_{\mathcal{C}_\mathbb{S}} := \inf\{t \geq 0 : x_t \notin \mathcal{C}_\mathbb{S}\}.$$

*Then for every $t > 0$,*

$$\mathbb{P}(\tau_{\mathcal{C}_\mathbb{S}} \leq t) \geq 2\left(1 - \Phi\left(\frac{r_{\min}}{\sqrt{t}}\right)\right),$$

*where $\Phi(x) = \frac{1}{\sqrt{2\pi}}\int_{-\infty}^x e^{-u^2/2}\, du$ is the standard normal CDF.*

*Moreover, by continuity of the sample paths, $x_{\tau_\mathbb{S}} \in \partial\mathcal{C}_\mathbb{S}$ almost surely.*

*Proof.* By the uniform nondegeneracy assumption, we have $\|\sigma^\top a_{ij}\| = c^{(ij)} > 0$.

Consider the gap process

$$Y_t^{(i,j)} := a_{ij}^\top X_t - d^{(i,j)}.$$

We observe that $Y_t^{(i,j)} = 0$ when $x_t$ is on the boundary created by experts $i, j$.

Applying Itô's formula and using the equation $dx_t = \sigma \, dB_t$, we obtain

$$dY_t^{(i,j)} = a_{ij}^\top \sigma \, dB_t$$

Let $u := \frac{\sigma^\top a_{ij}}{\|\sigma^\top a_{ij}\|} \in \mathbb{R}^d$, so $\|u\| = 1$, and define

$$\widetilde{B}_t^{(i,j)} := u^\top B_t \quad \text{(i.e., } \widetilde{B}_t^{(i,j)} = \frac{a_{ij}^\top \sigma}{\|\sigma^\top a_{ij}\|} \cdot B_t).$$

Since $B_t$ is a $d$-dimensional Brownian motion and $u$ is constant, $\widetilde{B}_t^{(i,j)}$ is a continuous local martingale with $\widetilde{B}_0^{(i,j)} = 0$. Its quadratic variation is $\langle \widetilde{B}_t^{(i,j)} \rangle_t = \int_0^t \|u\|^2 \, ds = t$.

By Lévy's characterization for Brownian motion, a continuous local martingale starting at $0$ with quadratic variation $t$ is a standard one-dimensional Brownian motion; hence $\widetilde{B}_t^{(i,j)}$ is a standard 1-dimensional Brownian motion.

We can rewrite:

$$dY_t^{(i,j)} = a_{ij}^\top \sigma \, dB_t = \|\sigma^\top a_{ij}\| d\widetilde{B}_t^{(i,j)}$$

Thus $Y^{(i,j)}$ is a nondegenerate 1-dimensional Brownian motion starting from

$$Y_0^{(i,j)} = a_{ij}^\top x_0 - d^{(i,j)} > 0.$$

Define the stopping time

$$\tau_{ij} := \inf\{t \geq 0 : Y_{ij}(t) \leq 0\}.$$

Intuitively, $\tau_{ij}$ is the first time the process $Y_{ij}(t)$, which starts positive, touches zero; i.e., the random moment when expert $i$ and $j$'s scores become equal and the trajectory hits the boundary.

In summary, we have the following:

$$dY_t^{(i,j)} = a_{ij}^\top \sigma \, dB_t = \|\sigma^\top a_{ij}\| d\widetilde{B}_t^{(i,j)}$$

$$Y_0^{(i,j)} = a_{ij}^\top x_0 - d^{(i,j)}.$$

$$\tau_{ij} := \inf\{t \geq 0 : Y_{ij}(t) \leq 0\}.$$

We want to bound the hitting time $\tau_{ij}$ using Proposition A.21.

Apply Proposition A.21 with $Y_0 = Y_0^{(i,j)} = a_{ij}^\top x_0 - d^{(i,j)}, c = \|\sigma^\top a_{ij}\|$ we obtain:

For every $t > 0$,

$$\mathbb{P}(\tau_{ij} \leq t) = 2\left(1 - \Phi\left(\frac{a_{ij}^\top x_0 - d^{(i,j)}}{\|\sigma^\top a_{ij}\|\sqrt{t}}\right)\right),$$

Since the first exit time $\tau_{\mathcal{C}_\mathbb{S}}$ from the open cell $\mathcal{C}_\mathbb{S}$ is the infimum of the exit times through all boundary faces, we have

$$\tau_{\mathcal{C}_\mathbb{S}} = \inf_{i \in \mathbb{S}, \, j \notin \mathbb{S}} \tau_{ij}.$$

Thus

$$\{\tau_{\mathcal{C}_\mathbb{S}} \leq t\} = \bigcup_{i \in \mathbb{S}, j \notin \mathbb{S}} \{\tau_{ij} \leq t\}.$$

The probability of the union is at least as large as the maximum probability of its members. Therefore

$$\mathbb{P}(\tau_{\mathcal{C}_\mathbb{S}} \leq t) \geq \max_{i \in \mathbb{S}, \, j \notin \mathbb{S}} \mathbb{P}(\tau_{ij} \leq t).$$

Let

$$r_{\min} := \min_{i \in \mathbb{S}, \, j \notin \mathbb{S}} \frac{a_{ij}^\top x_0 - d^{(i,j)}}{\|\sigma^\top a_{ij}\|} > 0.$$

Since $\mathbb{P}(\tau_{ij} \leq t) = 2\left(1 - \Phi\left(\frac{a_{ij}^\top x_0 - d^{(i,j)}}{\|\sigma^\top a_{ij}\|\sqrt{t}}\right)\right)$ decreases as $r_{ij}$ increases, the maximum is attained at $r_{\min}$. Hence, for every $t > 0$,

$$\mathbb{P}(\tau_{\mathcal{C}_{\mathbb{S}}} \leq t) \; \geq \; 2\left(1 - \Phi\left(\tfrac{r_{\min}}{\sqrt{t}}\right)\right).$$

By construction, at the exit time $\tau_{\mathbb{S}}$ at least one inequality becomes tight, i.e. $Y_{\tau_{\mathbb{S}}}^{(i,j)} = 0$ for some pair $(i,j)$, so $x_{\tau_{\mathbb{S}}} \in \partial\mathcal{C}_{\mathbb{S}}$. Since $x_t$ has continuous sample paths, the exit occurs on $\partial\mathcal{C}_{\mathbb{S}}$ almost surely. $\qquad\square$

From Theorem A.22, we can establish that the exit time $\tau_{\mathbb{S}}$ is finite almost surely in the next corollary.

**Corollary A.23.** *The exit time $\tau_{\mathbb{S}}$ of the diffusion process from the polyhedral cell $\mathcal{C}_{\mathbb{S}}$ is finite almost surely; that is,*

$$\mathbb{P}(\tau_{\mathbb{S}} < \infty) = 1.$$

*Proof.* By Theorem A.22 we have for every $t > 0$,

$$\mathbb{P}(\tau_{\mathbb{S}} \leq t) \; \geq \; 2\left(1 - \Phi\left(\tfrac{r_{\min}}{\sqrt{t}}\right)\right) \xrightarrow[t \to \infty]{} 1,$$

which implies $\mathbb{P}(\tau_{\mathbb{S}} < \infty) = 1$. $\qquad\square$

*Remark* A.24. Theorem A.22 and Corollary A.23 asserts two key properties of the randomly perturbed diffusion process $dx_t = \sigma\, dB_t$ in relation to the polyhedral cell $\mathcal{C}_{\mathbb{S}}$: (i) the exit time $\tau_{\mathbb{S}}$ is finite almost surely, so the process cannot remain in $\mathcal{C}_{\mathbb{S}}$ indefinitely; (ii) due to continuity of the sample paths, the exit occurs on the boundary $\partial\mathcal{C}_{\mathbb{S}}$.

### A.5.2 Equivalence between cell boundaries and the discontinuity set

In Section A.5, we established that a randomly perturbed diffusion process starting inside any top-$k$ cell $\mathcal{C}_{\mathbb{S}}$ with fixed diffusion coefficient almost surely exits the cell in finite time, i.e., it hits the boundary $\partial\mathcal{C}_{\mathbb{S}}$ with probability one. However, we have not proved that the union of all such cell boundaries coincides with the discontinuity set $\Gamma$. This result can be proved directly from the definitions, which we provide a proof in Lemma A.25.

**Lemma A.25** (Union of all boundaries and the discontinuous set coincides). *For each $k$-subset $\mathbb{S}$, let the open cell be*

$$\mathcal{C}_{\mathbb{S}} = \{x : z_i(x) > z_j(x) \;\; \forall i \in \mathbb{S},\; j \notin \mathbb{S}\}, \quad \overline{\mathcal{C}_{\mathbb{S}}} = \{x : z_i(x) \geq z_j(x) \;\; \forall i \in \mathbb{S},\; j \notin \mathbb{S}\}.$$

*Define the switching facets*

$$\mathbb{F}_{\mathbb{S},i,j} = \Big\{ x : z_i(x) = z_j(x),\; z_i(x) \leq z_\ell(x) \;\forall \ell \in \mathbb{S} \setminus \{i\},\; z_m(x) \leq z_j(x) \;\forall m \notin (\mathbb{S} \cup \{j\}) \Big\},$$

*and*

$$\Gamma = \bigcup_{|\mathbb{S}| = k} \; \bigcup_{i \in \mathbb{S},\, j \notin \mathbb{S}} \mathbb{F}_{\mathbb{S},i,j}.$$

*Then*

$$\Gamma \;=\; \bigcup_{|\mathbb{S}| = k} \partial\mathcal{C}_{\mathbb{S}}.$$

*Proof.* (i) $\Gamma \subseteq \bigcup_{|\mathbb{S}|=k} \partial\mathcal{C}_{\mathbb{S}}$.

Fix $\mathbb{S}$ and $i \in \mathbb{S}$, $j \notin \mathbb{S}$. If $x \in \mathbb{F}_{\mathbb{S},i,j}$, then $z_i(x) = z_j(x)$ and $z_i(x) \leq z_\ell(x)$ for all $\ell \in \mathbb{S} \setminus \{i\}$ while $z_m(x) \leq z_j(x)$ for all $m \notin (\mathbb{S} \cup \{j\})$. Hence $x \in \overline{\mathcal{C}_{\mathbb{S}}}$ and $x \notin \mathcal{C}_{\mathbb{S}}$, so $x \in \partial\mathcal{C}_{\mathbb{S}}$. Thus $\mathbb{F}_{\mathbb{S},i,j} \subseteq \partial\mathcal{C}_{\mathbb{S}}$, and the union gives the inclusion.

(ii) $\bigcup_{|\mathbb{S}|=k} \partial\mathcal{C}_{\mathbb{S}} \subseteq \Gamma$.

Let $x \in \partial\mathcal{C}_{\mathbb{S}}$ for some $\mathbb{S}$. Then $x \in \overline{\mathcal{C}_{\mathbb{S}}}$ but $x \notin \mathcal{C}_{\mathbb{S}}$, so there exists an inside–outside pair with equality: $\exists\, i \in \mathbb{S},\, j \notin \mathbb{S}$ such that $z_i(x) = z_j(x)$. Let $i^\star \in \arg\min_{\ell \in \mathbb{S}} z_\ell(x)$ and $j^\star \in \arg\max_{m \notin \mathbb{S}} z_m(x)$.

Since $x \in \overline{\mathcal{C}_{\mathbb{S}}}$, we have $\min_{\ell \in \mathbb{S}} z_\ell(x) \geq \max_{m \notin \mathbb{S}} z_m(x)$; because $x \notin \mathcal{C}_{\mathbb{S}}$, the strict inequality fails, hence

$$z_{i^\star}(x) = \min_{\ell \in \mathbb{S}} z_\ell(x) = \max_{m \notin \mathbb{S}} z_m(x) = z_{j^\star}(x).$$

By construction, $z_{i^\star}(x) \leq z_\ell(x)$ for all $\ell \in \mathbb{S} \setminus \{i^\star\}$ and $z_m(x) \leq z_{j^\star}(x)$ for all $m \notin (\mathbb{S} \cup \{j^\star\})$, i.e. $x \in \mathbb{F}_{\mathbb{S}, i^\star, j^\star}$. Therefore $x \in \Gamma$.

Combining (i) and (ii) yields $\Gamma = \bigcup_{|\mathbb{S}| = k} \partial \mathcal{C}_{\mathbb{S}}$. $\qquad\square$

*Remark* A.26. Using Lemma A.25 and Theorem A.22, we can conclude that a randomly perturbed diffusion process initiated inside any top-$k$ cell $\mathcal{C}_{\mathbb{S}}$ with fixed, nondegenerate diffusion coefficient almost surely reaches a discontinuity boundary in finite time.

### A.5.3 FIRST EXIT ALMOST SURELY AS ORDER-1 DISCONTINUITY

From Theorem A.22 and Lemma A.25, we know that the first hitting time of the discontinuity set is almost surely finite. What remains unclear is the type of discontinuity reached at exit. In the next part, we show that, with probability one, the process exits through an order-1 discontinuity. The key tool is a classical lemma: an $r$-dimensional Brownian motion ($r \geq 2$) almost surely never hits a fixed point in $\mathbb{R}^r$ at any time.

**Lemma A.27.** *Let $(B_t)_{t \geq 0}$ be standard $d$-dimensional Brownian motion with $d \geq 2$ and $B_0 = 0$. For any fixed $a \in \mathbb{R}^d$ with $a \neq 0$,*

$$\mathbb{P}(\exists\, t > 0: \ B_t = a) = 0.$$

*Proof.* Let $r < |a| < R$ and define $R_t = \|B_t - a\|$. Set the stopping times

$$\tau_r := \inf\{t \geq 0: \ R_t = r\}, \qquad \tau_R := \inf\{t \geq 0: \ R_t = R\}.$$

*Case $d \geq 3$:*

Let $u(x) = \|x - a\|^{2-d}$, which is harmonic on $\mathbb{R}^d \setminus \{a\}$.

Applying Itô's formula,

$$du(B_t) = \nabla u(B_t) \cdot dB_t,$$

so $u(B_t)$ is a local martingale.

By optional stopping theorem for the bounded stopping time $\tau_r \wedge \tau_R$, we obtain

$$\mathbb{E}\big[u(B_{\tau_r \wedge \tau_R})\big] = u(B_0) = |a|^{2-d}.$$

Since $B_{\tau_r \wedge \tau_R}$ lies on the sphere of radius $r$ or $R$ centered at $a$, we have

$$\mathbb{P}(\tau_r < \tau_R)\, r^{2-d} + \mathbb{P}(\tau_R < \tau_r)\, R^{2-d} = |a|^{2-d}.$$

Thus

$$\mathbb{P}(\tau_r < \tau_R) = \frac{|a|^{2-d} - R^{2-d}}{r^{2-d} - R^{2-d}}.$$

Letting $r \downarrow 0, R \uparrow \infty$ gives

$$\mathbb{P}(\tau_r < \infty) \xrightarrow[r \downarrow 0]{} 0.$$

That is, the probability that the Brownian path ever enters an arbitrarily small neighborhood of $a$ vanishes. Consequently, the event of hitting the exact point $a$ has probability zero, and hence

$$\mathbb{P}(\exists\, t > 0: \ B_t = a) = 0.$$

*Case $d = 2$:*

Let $v(x) = \log \|x - a\|$, which is harmonic on $\mathbb{R}^2 \setminus \{a\}$.

By Itô's formula, $v(B_t)$ is a local martingale, hence by optional stopping at $\tau_r \wedge \tau_R$,

$$\log |a| = \mathbb{E}\big[v(B_{\tau_r \wedge \tau_R})\big] = (\log r)\, \mathbb{P}(\tau_r < \tau_R) + (\log R)\, \mathbb{P}(\tau_R < \tau_r).$$

Therefore

$$\mathbb{P}(\tau_r < \tau_R) = \frac{\log R - \log|a|}{\log R - \log r} \xrightarrow[r\downarrow 0]{} 0,$$

Letting $r \downarrow 0, R \uparrow \infty$ gives

$$\mathbb{P}(\tau_r < \infty) \xrightarrow[r\downarrow 0]{} 0.$$

and, as above, $\mathbb{P}(\exists\, t > 0 : B_t = a) = 0$.

$\square$

**Corollary A.28.** *By translation invariance of Brownian motion, Lemma A.27 implies that if a standard $d$-dimensional Brownian motion $(B_t)_{t\geq0}$ starts at $B_0 = a$ with $a \neq 0$, then it almost surely never hits the origin:*

$$\mathbb{P}\big(\exists\, t > 0 : B_t = 0\big) = 0.$$

**Lemma A.29** (Linear image of Brownian motion). *Let $(B_t)_{t\geq0}$ be a standard $d$-dimensional Brownian motion and let $A \in \mathbb{R}^{n\times d}$ have rank $n \leq d$. Define $Z_t := AB_t$. Then $(Z_t)_{t\geq0}$ is an $n$-dimensional Brownian motion with covariance matrix $AA^\top$, i.e.*

$$Z_0 = 0, \quad Z \text{ has continuous paths}, \quad Z_t - Z_s \sim \mathcal{N}(0, (t-s)\,AA^\top)$$

*with independent, stationary increments. In particular, $\widetilde{B}_t := (AA^\top)^{-1/2}Z_t$ is a standard $n$-dimensional Brownian motion.*

*Proof.* Since $B_0 = 0$ and $t \mapsto B_t$ is continuous, we have $Z_0 = AB_0 = 0$ and $t \mapsto Z_t = AB_t$ is continuous.

For $0 \leq s < t$, the increment $B_t - B_s$ is independent of $\mathcal{F}_s := \sigma(B_u : u \leq s)$ and has law $\mathcal{N}(0, (t-s)I_d)$. Applying the linear map $A$,

$$Z_t - Z_s = A(B_t - B_s),$$

which is (joint) Gaussian with mean 0 and covariance

$$\mathrm{Cov}(Z_t - Z_s) = A\,\mathrm{Cov}(B_t - B_s)\,A^\top = A\big((t-s)I_d\big)A^\top = (t-s)\,AA^\top.$$

Independence of increments is preserved under linear maps: if $(X_1, \ldots, X_m)$ are independent, then so are $(AX_1, \ldots, AX_m)$. Hence $(Z_t)$ has independent, stationary Gaussian increments with the stated covariance, and is adapted with continuous paths.

By the characterization of Brownian motion as a continuous Gaussian process with independent, stationary increments and covariance $\mathbb{E}[Z_t Z_s^\top] = (t \wedge s)\,AA^\top$, we conclude that $Z$ is an $n$-dimensional Brownian motion with covariance $AA^\top$. Finally, since $AA^\top$ is symmetric positive definite (rank $n$), $(AA^\top)^{-1/2}$ exists and

$$\widetilde{B}_t := (AA^\top)^{-1/2}Z_t$$

has covariance $(t-s)I_n$ for each increment, hence is standard $n$-dimensional Brownian motion. $\square$

We now use Corollary A.28 to show that the exit almost surely occurs on an order-1 discontinuity. The Corollary A.28 is applied here to rule out the simultaneous satisfaction of multiple independent boundary equalities, which almost surely does not occur.

**Theorem A.30** (Exit occurs on an order-1 discontinuity). *Let $x_t$ solve $dx_t = \sigma\,dB_t$ with invertible $\sigma \in \mathbb{R}^{d\times d}$ and $x_0 \in \mathcal{C}_\mathbb{S}$, and let $\tau_\mathbb{S} := \inf\{t \geq 0 : x_t \notin \mathcal{C}_\mathbb{S}\}$. Then*

$$\mathbb{P}\big(x_{\tau_\mathbb{S}} \in \Gamma^{(1)}\big) = 1 \quad \text{and} \quad \mathbb{P}\big(x_{\tau_\mathbb{S}} \in \Gamma^{(n)}\big) = 0 \text{ for all } n \geq 2.$$

*Proof.* Define $y_t := \sigma^{-1}x_t$; then $y_t$ is a standard Brownian motion in $\mathbb{R}^d$ (denoted $B_t$), and $\mathcal{D} := \sigma^{-1}\mathcal{C}_\mathbb{S}$ is a polyhedral domain.

Suppose the exit occurs at an order-$n$ discontinuity with $n \geq 2$. Then there exists an index set $I = \{i_1, \ldots, i_{n+1}\}$ with $|I| = n + 1$ such that

$$z_{i_1}(x_{\tau_\mathbb{S}}) = \cdots = z_{i_{n+1}}(x_{\tau_\mathbb{S}}),$$

and this common value coincides with the $k \to k+1$ threshold. Equivalently, at $y_{\tau_{\mathbb{S}}}$ we have $n$ independent equalities

$$z_{i_2}(y) - z_{i_1}(y) = \cdots = z_{i_{n+1}}(y) - z_{i_1}(y) = 0.$$

Define the $n$-dimensional process

$$U_t := \big(z_{i_2}(y_t) - z_{i_1}(y_t), \ \ldots, \ z_{i_{n+1}}(y_t) - z_{i_1}(y_t)\big).$$

Since each $z_i$ is affine, there exist $A \in \mathbb{R}^{n \times d}$ and $b \in \mathbb{R}^n$ such that

$$U_t = Ay_t + b = AB_t + b.$$

By construction, $A$ is obtained by taking $n$ independent row differences of $W_g$; since $W_g$ has full row rank, it follows that $\mathrm{rank}(A) = n$. Consequently, $AA^\top$ is symmetric positive definite, and $(AA^\top)^{-1/2}$ exists uniquely.

By Lemma A.29, $AB_t$ is an $n$-dimensional Brownian motion with covariance $AA^\top$. Hence the centered process

$$\widetilde{U}_t := U_t - b$$

is an $n$-dimensional Brownian motion with nonstandard covariance $AA^\top$. Define

$$\widehat{B}_t := (AA^\top)^{-1/2}\,\widetilde{U}_t,$$

which has the law of a standard $n$-dimensional Brownian motion (this follows by the same reasoning as Lemma A.29).

Because $x_0 \in \mathcal{C}_{\mathbb{S}}$, we have $U_0 = b \neq 0$, hence

$$\{\exists\, t > 0 : \ U_t = 0\} = \{\exists\, t > 0 : \ \widetilde{U}_t = -b\} = \Big\{\exists\, t > 0 : \ \widehat{B}_t = -(AA^\top)^{-1/2}b\Big\}.$$

Since $n \geq 2$ and $-(AA^\top)^{-1/2}b \neq 0$, Corollary A.28 yields

$$\mathbb{P}\big(\exists\, t > 0 : \ U_t = 0\big) = 0.$$

Exiting at an order-$n$ discontinuity would necessarily require that the process $U_t$ reaches the origin, i.e. $U_{\tau_{\mathbb{S}}} = 0$. However, as shown in Corollary A.28, an $n$-dimensional Brownian motion with $n \geq 2$ almost surely never hits any fixed point distinct from its initial condition. Since $U_0 \neq 0$, the probability of $U_t$ ever reaching 0 is therefore zero. It follows that exits through order-$n$ discontinuities with $n \geq 2$ occur with probability zero, and consequently the exit must almost surely take place on an order-1 discontinuity, that is,

$$\mathbb{P}\big(x_{\tau_{\mathbb{S}}} \in \Gamma^{(1)}\big) = 1 \quad \text{and} \quad \mathbb{P}\big(x_{\tau_{\mathbb{S}}} \in \Gamma^{(n)}\big) = 0 \ \text{ for all } n \geq 2.$$

$\square$

### A.5.4 Occupation time near discontinuity sets

Fix $\epsilon > 0$ and, for each order $n \geq 1$, let $T_\epsilon(\Gamma^{(n)})$ be the $\epsilon$–tube around the order-$n$ discontinuity set $\Gamma^{(n)}$.

Let $(X_t)_{t \geq 0}$ be an Itô process in $\mathbb{R}^D$ with initial condition $X_0 = x_0 \in \mathcal{C}_{\mathbb{S}}$ for some polyhedral cell $\mathcal{C}_{\mathbb{S}}$,

$$dX_t = \sigma\, dB_t,$$

where $B_t$ is a standard $D$–dimensional Brownian motion and $\sigma \in \mathbb{R}^{D \times D}$ is constant.

The *occupation time* of $X$ in the tube of order $r$ up to horizon $T$ is

$$A_\epsilon^{(r)}(T; x_0) \ := \ \int_0^T \mathbf{1}\big\{X_t \in T_\epsilon(\Gamma^{(n)})\big\}\, dt,$$

and its time-average (fraction of time spent in the tube) is

$$L_\epsilon^{(r)}(T; x_0) \ := \ \frac{1}{T}\, A_\epsilon^{(n)}(T; x_0).$$

For expectations,

$$\mathbb{E}_{x_0}\big[A_\epsilon^{(n)}(T)\big] = \int_0^T \mathbb{P}_{x_0}\big\{X_t \in T_\epsilon(\Gamma^{(n)})\big\}\, dt, \qquad \mathbb{E}_{x_0}\big[L_\epsilon^{(n)}(T)\big] = \frac{1}{T}\int_0^T \mathbb{P}_{x_0}\big\{X_t \in T_\epsilon(\Gamma^{(n)})\big\}\, dt.$$

**Proposition A.31** (Occupation time near one codimension-$n$ flat). *Let $1 \leq n < D$ and let $S = \{x \in \mathbb{R}^D : Ax = d\}$ be an affine flat with $\mathrm{rank}(A) = n$. Let $X_t$ solve $dX_t = \sigma \, dB_t$, $X_0 = x_0$, where $B_t$ is standard $D$–dimensional Brownian motion and $\Sigma := \sigma\sigma^\top \succ 0$. Choose an orthonormal basis $N \in \mathbb{R}^{D \times n}$ of $S^\perp$ and set*

$$\Sigma_\perp := N^\top \Sigma N \in \mathbb{R}^{n \times n}, \qquad \lambda_{\min} := \lambda_{\min}(\Sigma_\perp).$$

*Fix any $y_0 \in S$ and write $s_0 := N^\top y_0$ and $\mu := N^\top x_0$. For $\epsilon > 0$ and $T > 0$, define the occupation time*

$$A_\epsilon^{(n)}(T; S) := \int_0^T \mathbf{1}\{\mathrm{dist}(X_t, S) < \epsilon\} \, dt.$$

*Let*

$$\omega_n := \lambda^r(B^n(0,1)), \quad K_n := \frac{\omega_n}{(2\pi)^{n/2}\sqrt{\det(\Sigma_\perp)}}, \quad \delta_\epsilon := \left\| \Sigma_\perp^{-1/2}(s_0 - \mu) \right\| - \frac{\epsilon}{\sqrt{\lambda_{\min}}}, \quad b_\epsilon := \frac{(\delta_\epsilon)_+^2}{2}.$$

*Then, for all $T > 0$,*

$$\mathbb{E}\left[ A_\epsilon^{(n)}(T; S) \right] \ \leq \ K_n \, \epsilon^n \int_0^T t^{-n/2} \, e^{-b_\epsilon/t} \, dt = \begin{cases} K_n \, \epsilon^n \, b_\epsilon^{1 - \frac{n}{2}} \, \Gamma\big(\frac{n}{2} - 1, \frac{b_\epsilon}{T}\big), & n > 2, \\[2mm] K_2 \, \epsilon^2 \, E_1\big(\frac{b_\epsilon}{T}\big), & n = 2, \\[2mm] \leq \ 2K_1 \, \epsilon \, \sqrt{T}, & n = 1, \end{cases}$$

*where $\Gamma(\cdot, \cdot)$ is the upper incomplete gamma function and $E_1(z) = \int_z^\infty e^{-u} u^{-1} \, du$.*

*Proof.* **Step 1 (normal coordinates).** Because $N$ has orthonormal columns spanning $S^\perp$, every $x \in \mathbb{R}^D$ decomposes as $x = y + Nv$ with $y \in S$ and $v \in \mathbb{R}^n$; moreover $\mathrm{dist}(x, S) = \|v\|$ and $N^\top y = s_0$ (independent of $y \in S$).

**Step 2 (projected process and its density).** Define the normal projection $Z_t := N^\top X_t \in \mathbb{R}^n$. Since $dZ_t = N^\top \sigma \, dB_t$, we have

$$Z_t \sim \mathcal{N}(\mu, \, t\,\Sigma_\perp), \qquad \mu := N^\top x_0, \quad \Sigma_\perp := N^\top \Sigma N.$$

Hence the transition density of $Z_t$ is

$$g_t(z) = \frac{1}{(2\pi t)^{n/2}\sqrt{\det(\Sigma_\perp)}} \exp\left( -\frac{1}{2t} \left\| \Sigma_\perp^{-1/2}(z - \mu) \right\|^2 \right).$$

**Step 3 (event $\{\mathrm{dist}(X_t, S) < \epsilon\}$ in normal coords).** We have

$$\mathrm{dist}(X_t, S) < \epsilon \quad \Longleftrightarrow \quad \|Z_t - s_0\| < \epsilon.$$

Therefore

$$\mathbb{P}\{\mathrm{dist}(X_t, S) < \epsilon\} = \int_{\|z - s_0\| < \epsilon} g_t(z) \, dz.$$

**Step 4 (uniform bound on the integrand over the ball).** Let $B := \{z \in \mathbb{R}^n : \|z - s_0\| < \epsilon\}$. By the triangle inequality in the Mahalanobis norm,

$$\inf_{z \in B} \left\| \Sigma_\perp^{-1/2}(z - \mu) \right\| \ \geq \ \left\| \Sigma_\perp^{-1/2}(s_0 - \mu) \right\| - \sup_{z \in B} \left\| \Sigma_\perp^{-1/2}(z - s_0) \right\|.$$

Since $\|\Sigma_\perp^{-1/2} w\| \leq \|\Sigma_\perp^{-1/2}\|_2 \|w\|$ and $\|\Sigma_\perp^{-1/2}\|_2 = 1/\sqrt{\lambda_{\min}}$, we get

$$\sup_{z \in B} \left\| \Sigma_\perp^{-1/2}(z - s_0) \right\| \ \leq \ \frac{\epsilon}{\sqrt{\lambda_{\min}}}.$$

Hence

$$\inf_{z \in B} \left\| \Sigma_\perp^{-1/2}(z - \mu) \right\| \ \geq \ \delta_\epsilon := \left\| \Sigma_\perp^{-1/2}(s_0 - \mu) \right\| - \frac{\epsilon}{\sqrt{\lambda_{\min}}}.$$

**Step 5 (probability bound at time $t$).** Using the bound from Step 4 in the density from Step 2, for all $z \in B$,

$$g_t(z) \;\leq\; \frac{1}{(2\pi t)^{n/2}\sqrt{\det(\Sigma_\perp)}} \; \exp\!\left(-\frac{(\delta_\epsilon)_+^2}{2t}\right).$$

Therefore

$$\mathbb{P}\{\mathrm{dist}(X_t, S) < \epsilon\} \leq \lambda^n(B) \cdot \frac{e^{-(\delta_\epsilon)_+^2/(2t)}}{(2\pi t)^{n/2}\sqrt{\det(\Sigma_\perp)}} = K_n\,\epsilon^n\,t^{-n/2}\,e^{-b_\epsilon/t},$$

since $\lambda^n(B) = \omega_r \epsilon^n$ and $b_\epsilon = \frac{1}{2}(\delta_\epsilon)_+^2$.

**Step 6 (time integration).** Integrating from $0$ to $T$,

$$\mathbb{E}\Big[A_\epsilon^{(n)}(T; S)\Big] = \int_0^T \mathbb{P}\{\mathrm{dist}(X_t, S) < \epsilon\}\, dt \leq K_n\,\epsilon^n \int_0^T t^{-n/2}\,e^{-b_\epsilon/t}\,dt.$$

**Step 7 (evaluation of the integral).**

- If $n > 2$, substitute $u = b_\epsilon/t$ (so $t = b_\epsilon/u$, $dt = -b_\epsilon u^{-2}du$):

$$\int_0^T t^{-n/2}e^{-b_\epsilon/t}\,dt = b_\epsilon^{1-\frac{n}{2}}\int_{b_\epsilon/T}^\infty u^{\frac{n}{2}-2}e^{-u}\,du = b_\epsilon^{1-\frac{n}{2}}\,\Gamma\!\left(\frac{n}{2}-1, \frac{b_\epsilon}{T}\right).$$

- If $n = 2$, the integral equals $E_1(b_\epsilon/T)$ (the exponential integral).

- If $n = 1$, drop the exponential to get the simple bound $\int_0^T t^{-1/2}e^{-b_\epsilon/t}\,dt \leq \int_0^T t^{-1/2}\,dt = 2\sqrt{T}$.

Multiplying by $K_n\epsilon^n$ gives the stated bounds in all cases. $\qquad\square$

Apply the previous proposition to the union over all tie sets $J$ of an order-$n$ discontinuity gives us the next theorem.

**Theorem A.32** (Occupation time near order-$n$ discontinuities). *Assume* $\Gamma^{(n)} \subseteq \bigcup_{J \in \mathcal{J}_n} S_J^{(n)}$ *with* $S_J^{(n)} = \{x \in \mathbb{R}^D : A_J^{(n)}x = d_J^{(n)}\}$, $\mathrm{rank}(A_J^{(n)}) = r$. *Let* $X_t$ *solve* $dX_t = \sigma\, dB_t$, $X_0 = x_0$, *with* $\Sigma := \sigma\sigma^\top \succ 0$. *For each $J$, choose an orthonormal basis $N_J$ of $(S_J^{(n)})^\perp$ and set*

$$\Sigma_{\perp, J} := N_J^\top \Sigma N_J, \quad \lambda_{\min, J} := \lambda_{\min}(\Sigma_{\perp, J}), \quad s_J := N_J^\top y \;(y \in S_J^{(n)}), \quad \mu_J := N_J^\top x_0.$$

*Define*

$$K_{J,n} := \frac{\omega_n}{(2\pi)^{n/2}\sqrt{\det(\Sigma_{\perp, J})}}, \qquad \delta_{J,\epsilon} := \left\|\Sigma_{\perp, J}^{-1/2}(s_J - \mu_J)\right\| - \frac{\epsilon}{\sqrt{\lambda_{\min, J}}}, \qquad b_{J,\epsilon} := \frac{(\delta_{J,\epsilon})_+^2}{2}.$$

*Let*

$$A_\epsilon^{(n)}(T; \Gamma) := \int_0^T \mathbf{1}\{X_t \in T_\epsilon(\Gamma^{(n)})\}\, dt.$$

*Then, for all $T > 0$,*

$$\mathbb{E}\big[A_\epsilon^{(n)}(T; \Gamma)\big] \;\leq\; \sum_{J \in \mathcal{J}_n} K_{J,n}\,\epsilon^n \int_0^T t^{-n/2}e^{-b_{J,\epsilon}/t}\,dt. \qquad (17)$$

*In particular,*

$$\mathbb{E}\big[A_\epsilon^{(n)}(T; \Gamma)\big] \;\leq\; \begin{cases} \displaystyle\sum_J K_{J,n}\,\epsilon^n\, b_{J,\epsilon}^{1-\frac{n}{2}}\,\Gamma\!\left(\frac{n}{2}-1, \frac{b_{J,\epsilon}}{T}\right), & n > 2, \\[2ex] \displaystyle\sum_J K_{J,2}\,\epsilon^2\, E_1\!\left(\frac{b_{J,\epsilon}}{T}\right), & n = 2, \\[2ex] \displaystyle 2\Big(\sum_J K_{J,1}\Big)\epsilon\sqrt{T}, & n = 1. \end{cases}$$

*A coarser but convenient bound (using $K_n^{\mathrm{sum}} := \sum_J K_{J,n}$ and $b_{\min} := \min_J b_{J,\epsilon}$) is*

$$\mathbb{E}\big[A_\epsilon^{(n)}(T;\Gamma)\big] \;\leq\; K_n^{\mathrm{sum}}\,\epsilon^n \int_0^T t^{-n/2} e^{-\,b_{\min}/t}\,dt,$$

*Proof.* (*Step 1: union domination*) Since $\Gamma^{(n)} \subseteq \bigcup_J S_J^{(n)}$,

$$T_\epsilon(\Gamma^{(n)}) \;\subseteq\; T_\epsilon\Big(\bigcup_J S_J^{(n)}\Big) \;\subseteq\; \bigcup_J T_\epsilon(S_J^{(n)}),$$

hence pointwise $\mathbf{1}_{\{X_t \in T_\epsilon(\Gamma^{(n)})\}} \leq \sum_J \mathbf{1}_{\{X_t \in T_\epsilon(S_J^{(n)})\}}$.

(*Step 2: integrate and take expectations*) Integrate in $t \in [0, T]$ and take expectations:

$$\mathbb{E}[A_\epsilon^{(n)}(T;\Gamma)] \;\leq\; \sum_J \mathbb{E}\left[\int_0^T \mathbf{1}\{X_t \in T_\epsilon(S_J^{(n)})\}\,dt\right] = \sum_J \int_0^T \mathbb{P}\{X_t \in T_\epsilon(S_J^{(n)})\}\,dt.$$

(*Step 3: apply the single–flat bound to each $J$*) For each fixed $J$, apply Proposition A.31 (with $N_J, \Sigma_{\perp,J}, s_J, \mu_J$). This gives

$$\mathbb{P}\{X_t \in T_\epsilon(S_J^{(n)})\} \;\leq\; K_{J,n}\,\epsilon^n\,t^{-n/2} e^{-\,b_{J,\epsilon}/t},$$

hence equation 17. Evaluating the time integral case-wise yields the formulas. For the coarser bound, use $b_{\min} \leq b_{J,\epsilon}$ so that $e^{-b_{J,\epsilon}/t} \leq e^{-b_{\min}/t}$, factor out $\sum_J K_{J,r}$, and integrate. $\qquad\square$

Table 4: Bits-per-character (BPC) of SmoothSMoE compared to baseline model on EnWiki-8 dataset.

| Model | Test BPC ↓ |
|---|---|
| *SMoE* | 1.153 |
| SmoothSMoE | **1.122** |

# B   FURTHER THEORETICAL ANALYSIS AND ABLATION STUDIES

## B.1   GEOMETRIC INTUITION BEHIND THEORETICAL ANALYSIS

Geometrically, the Top-$k$ SMoE gate partitions the input space into polyhedral regions (cells) where the active expert set is fixed. Inside each cell, the MoE map is a smooth combination of a fixed subset of experts; all nonsmooth behavior comes from crossing the boundaries between cells, where the Top-$k$ set changes. These boundaries are given by hyperplanes of the form $z_i(x) = z_j(x)$, that is, the locations where at least two experts tie. The order of a discontinuity simply counts how many experts tie exactly at the Top-$k$ score. For a simple illustration, consider in three-dimensional space, order-1 sets can be understood as "walls" partitioning the space where one active and one inactive expert swap. Higher-order sets correspond to intersections of several such walls, forming "edges" and "corners".

Our theoretical volume results explicitly formalize the intuition that, in a bounded region where the data live and are perturbed randomly, collisions with walls occur with probability 1, while collisions with "edges" and "corners" essentially do not occur (probability 0). Moreover, the distribution of the first collision time is closely linked to the shortest normalized distance from the starting point to these "walls" (Theorem 5.1). If we take a thin band of thickness $\epsilon$ around these sets inside a ball of radius $R$, the fraction of the band volume contributed by higher-order intersections shrinks polynomially in $\epsilon/R$ (Theorem 4.4), so as we increase $R$ or decrease $\epsilon$, almost all near-boundary

mass concentrates on simple walls. Finally, for a randomly perturbed process, the upper bound on the occupation time inside the $\epsilon$-band of an $n$-th order intersection decays exponentially as $\epsilon^n$ in the small-$\epsilon$ regime (Theorem 5.3).

Our smoothing layer is designed exactly around this picture. Instead of letting the output jump abruptly when crossing a wall, we replace the hard switch by a narrow transition band around the corresponding hyperplanes. Within this band, the contributions of the involved experts vary smoothly with the logits, so that moving across a wall interpolates between experts rather than flipping them discretely. Outside these bands, the model behaves like the original Top-$k$ gate, preserving sparsity and the usual MoE structure. Due to the dominant geometry of lower order discontinuities (For example "walls" compared to "edges" and "corners"), the number of additionally activated experts, which equals the order of the discontinuity (1 for walls, 2 for edges, and 3 for corners) is small, providing a theoretical guarantee for the efficiency of our smoothing mechanism.

### B.2  SMOOTHSMOE VS. OTHER DIFFERENTIABLE ROUTING METHODS

Recent works such as Soft MoE (Puigcerver et al., 2024), SMEAR (Muqeeth et al., 2023), and ReMoE (Wang et al., 2024) enforce full differentiability of the MoE routing map by altering the routing mechanism itself. Soft MoE and SMEAR achieve differentiability via token or expert merging, effectively replacing the sparse Top-$k$ selection map by a dense, smooth probability assignment over experts. From a functional perspective, this turns the piecewise-constant Top-$k$ map into a globally smooth map into the probability simplex, at the expense of token-wise sparsity and causality for autoregressive tasks. ReMoE instead replaces Top-$k$ and Softmax with a ReLU-based router equipped with an $\ell_1$-type load-balancing regularizer, thereby producing continuous gating scores but changing the underlying Top-$k$-induced polyhedral structure and requiring a different gating mechanism to be trained. In contrast, our SmoothSMoE keeps the original Top-$k$ gate and its polyhedral partition of the input space and only modifies logits for tokens whose scores fall inside an $\ell_{\infty,\epsilon}$–thickening of the discontinuity set. That is, we leave the routing map unchanged away from boundaries and apply smoothing only to near-ties $0 < z_{[k]}(x) - z_i(x) < \varepsilon$, which activates at most $n$ additional experts on an order-$n$ discontinuity (Proposition A.20). This design preserves sparsity and causal routing, while our measure-theoretic and stochastic analysis quantifies the volume of these thickened regions and the occupation time of a diffusion near them, providing explicit bounds on how frequently smoothing is used and hence limiting the extra computation it incurs. Thus SmoothSMoE is complementary to prior differentiable routing: it achieves continuity of the SMoE map locally around theoretically characterized discontinuity sets, rather than globally replacing Top-$k$ routing with a different differentiable router.

### B.3  DETAILED ANALYSIS ON $\ell_{\infty,\epsilon}$ LOCAL SMOOTHING VS. VANILLA SMoE NEAR DISCONTINUITY BOUNDARIES

To provide a concrete, empirical counterpart to our theoretical findings, this section presents a targeted experiment designed to visualize the behavior of SMoE and our proposed SmoothSMoE at the decision boundary. The primary objective is to isolate and illustrate the direct architectural impact of our smoothing mechanism on the model's output function, independent of other training dynamics.

**Experiment setup.**  Our experiment utilizes a multi-layer SMoE model pre-trained on the CIFAR-10 dataset. The architecture for each MoE layer consists of an input dimension of $D = 3072$, $E = 32$ experts, and a Top-$k$ gating mechanism with $k = 4$. Each expert is a standard two-layer MLP with a hidden size of 128.

The analysis proceeds on a per-layer basis. For a given layer, we first instantiate the original SMoE using its pre-trained weights. We then create an instance of our SmoothSMoE. To ensure a controlled comparison, the SmoothSMoE's weights are directly copied from the pretrained SMoE. This setup guarantees that any observed differences in behavior are attributable solely to our proposed smoothing architecture.

The core of our methodology is to identify and analyze a critical order-1 discontinuity. An order-1 discontinuity boundary is defined by the hyperplane where the gating scores of the $k$-th active expert and the highest-scoring inactive expert are equal ($z_{[k+1]}(x) = z_{[k]}(x)$). We employ Monte Carlo sampling strategy, generating thousands of random input vectors $x_0$ to locate a boundary region that

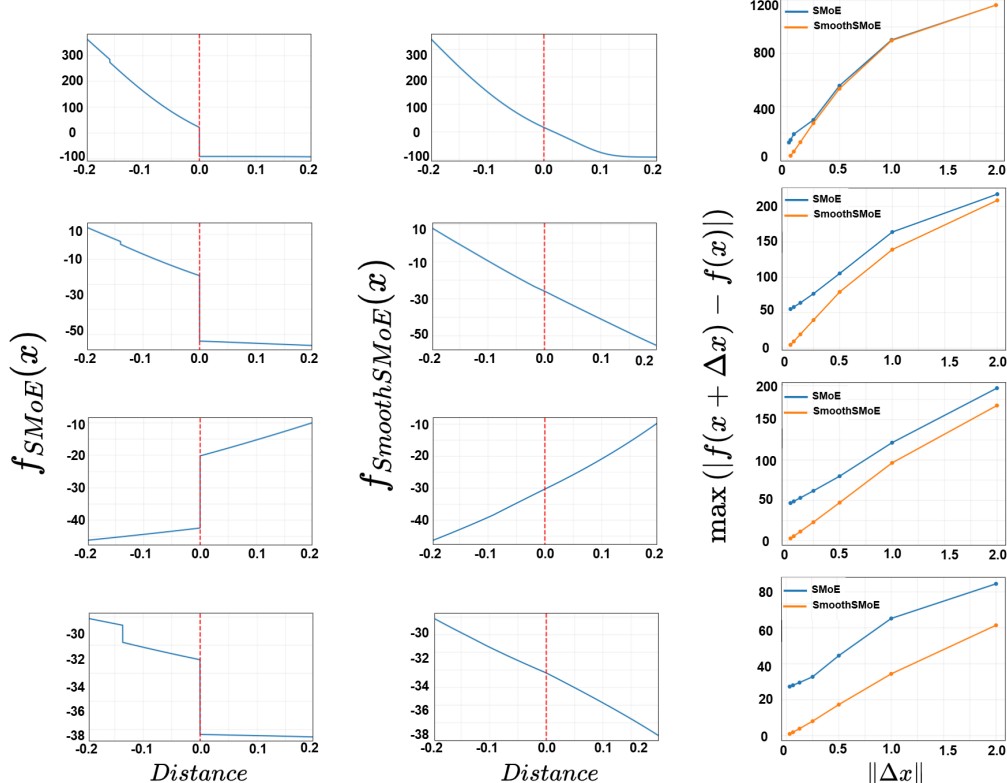

Figure 3: Visualizing the effect of our smoothing mechanism on SMoE layer outputs. Each row corresponds to a different SMoE layer from a pre-trained model. The columns show the standard SMoE, our SmoothSMoE, and the maximum output change, respectively. Left Column (SMoE): The standard SMoE exhibits sharp discontinuities as the input crosses the decision boundary. Middle Column (SmoothSMoE): Our SmoothSMoE, using identical weights, eliminates these jumps and produces a continuous output. Right Column: The maximum output gap $\max(|f(x+\Delta x)-f(x)|)$ is plotted against the perturbation size $\|\Delta x\|$. Our method shows the gap converging to zero, confirming continuity, while the SMoE maintains a large gap.

exhibit significant output jumps along a specific dimension. We then analyze the MoE map restricted to the chosen dimension, denoted $f_{\text{SMoE}} : \mathbb{X} \to \mathbb{R}$ for the Sparse MoE and $f_{\text{SmoothSMoE}} : \mathbb{X} \to \mathbb{R}$ for the SmoothSMoE. For each selected boundary, we compute the exact orthogonal projection, obtaining the point $x^{\perp}$.

To visualize the function's behavior when the input passing a discontinuity boundary, we analyze the output along a line $x = x^{\perp} + l\hat{\mathbf{n}}$ passing thought the discontinuity boundary, where $\hat{\mathbf{n}}$ is the unit normal vector to the boundary hyperplane and $l \in \mathbb{R}$. This line represents the traversal across the discontinuity. The variable $l$ (the horizontal axis in our plots) corresponds to the signed Euclidean distance from the boundary, with the boundary itself precisely at $l = 0$.

**Results**  Figure 3 presents the comparative results for four distinct layers of the model. The left column visualizes the output of the standard SMoE. As predicted by our analysis, the SMoE map is piecewise continuous but exhibits a pronounced jump discontinuity at the boundary. The magnitude of this jump is non-trivial, highlighting a potential source of instability for gradient-based optimization, reduced robustness to adversarial perturbations, and unpredictable outputs behavior when inputs are near these boundaries.

The middle column shows the output of our SmoothSMoE on the exact same line in the input space. The effect of our mechanism is immediately apparent: the discontinuity is completely removed. SmoothSMoE transitions smoothly and continuously across the boundary. This is a direct consequence of our method's ability to create a "soft" handoff between experts by continuous re-weighting, rather than the abrupt expert swapping inherent to Top-$k$ gating.

The right column provides a quantitative analysis of this smoothness. It plots the maximum output difference, $\max \|f(x + \Delta x) - f(x)\|$, against the magnitude of the input perturbation $\|\Delta x\|$ within a shrinking window around $x^\perp$. For the SMoE, the output gap plateaus at a large, non-zero value, confirming that the discontinuity persists even for infinitesimally small perturbations. In stark contrast, the plot for our SmoothSMoE shows the output difference converging to zero as $\|\Delta x\| \to 0$. This behavior provides a visual confirmation of the continuity induced by our method which is formally proved in Proposition A.7, a critical property for model stability and generalization that the standard SMoE lacks.

### B.4 How Boundary Loss Controls $\epsilon$ and the Average Number of Activated Experts

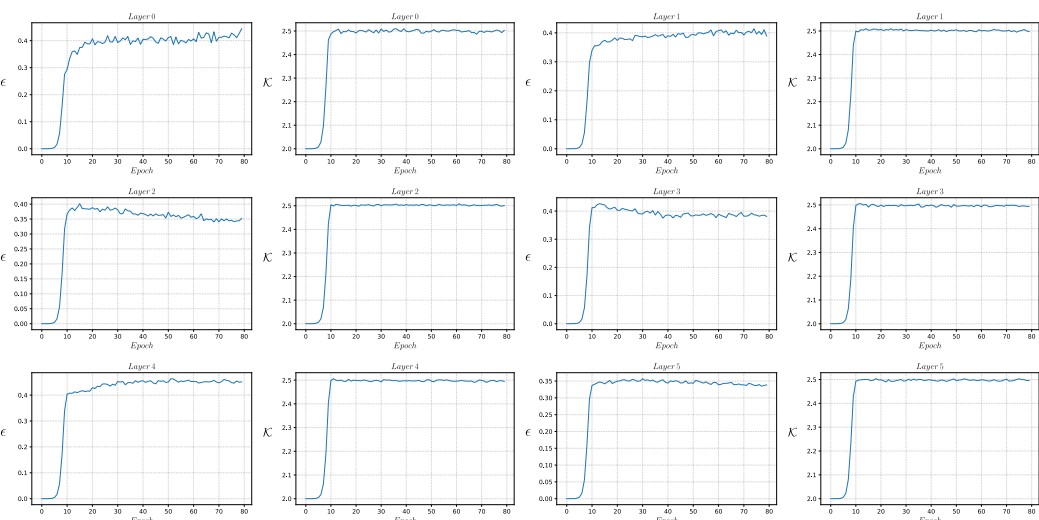

Figure 4: The effect of boundary loss on controlling $\epsilon$ and the average number of activated experts ($\mathcal{K}$) across various layers.

In this study, we analyze the training log from pretraining a 6-layer SmoothSMoE on WikiText-103 for 80 epochs, recording at each epoch the boundary threshold $\epsilon$ and the average number of activated experts $\mathcal{K}$ for every layer. Figure 4 shows how $\epsilon$ and $\mathcal{K}$ evolve during training. At the start, both values are close to $0$, since $\epsilon$ is initialized small to ensure efficiency. They initially grow slowly due to the learning-rate warmup, after which $\epsilon$ increases sharply until $\mathcal{K}$ approaches the target budget ($k^* = 2.5$ experts on average). This marks an adjustment phase where the model tunes $\epsilon$ so that $\mathcal{K}$ converges toward $k^*$. Once this balance is reached, both $\epsilon$ and $\mathcal{K}$ stabilize, with $\epsilon$ exhibiting only small fluctuations to keep $\mathcal{K}$ near the budget as training dynamics evolve. These observations confirm that the boundary loss effectively updates $\epsilon$ to maintain the desired average number of activated experts.

### B.5 Annealing Boundary Smoothing to Hard Top-$k$

In this analysis, we investigate the hypothesis that boundary smoothing makes the loss landscape more amenable to optimization, improves training dynamics and final performance. To test this hypothesis, we adopt an 80-epoch annealing schedule in which smoothing is progressively removed. For the first 40 epochs, we set the target budget to $k^* = 2.5$ to warm up the model, so that the smoothing mechanism can stabilize optimization by activating additional experts near switching surfaces. For

Table 5: Perplexity (PPL) of annealed SmoothSMoE compared to baseline SMoE and SmoothSMoE on clean and attacked WikiText-103 datasets.

| Model | WikiText-103 | | Attacked WikiText-103 | |
|---|---|---|---|---|
| | Valid PPL ↓ | Test PPL ↓ | Valid PPL ↓ | Test PPL ↓ |
| SMoE ($k = 2$) | 33.79 | 35.52 | 42.21 | 44.18 |
| SmoothSMoE annealed ($k = 2$) | 32.97 | 34.59 | 41.14 | 42.91 |
| SmoothSMoE ($k = 2.5$) | **32.72** | **34.35** | **40.99** | **42.85** |

the next 20 epochs, we linearly anneal $k^*$ from 2.5 down to 2, so that the routing gradually converges toward the target hard Top-$k$ regime. In the final 20 epochs, we fix $k^* = 2$, which effectively turns off smoothing and forces the learned $\epsilon$ parameter to converge to 0, allowing the parameters to fully adapt to hard Top-$k$ gating and eliminating train-inference mismatch. We refer to this training protocol as SmoothSMoE annealed. At inference time, we completely remove smoothing and evaluate with a standard Top-2 SMoE router.

As shown in Table 5, the SmoothSMoE annealed model achieves test perplexity 34.59 on WikiText-103, improving over the baseline SMoE (35.52) and placing its performance between SMoE and the full SmoothSMoE model. The same behaviour can be observed on Attacked WikiText-103 dataset. These results confirm our hypothesis that boundary smoothing, by allowing experts near routing boundaries to contribute and by making the loss landscape easier to optimize, improves the final Top-$k$ SMoE performance even when smoothing is completely removed at inference.

Figure 5 reports the average number of activated experts $\mathcal{K}$ across layers under the three-stage training schedule: $\mathcal{K}$ quickly rises and stabilizes around 2.5 during the warm-up stage, then is linearly reduced to 2 as smoothing is annealed, and finally remains at 2 throughout the hard Top-2 adaptation stage.

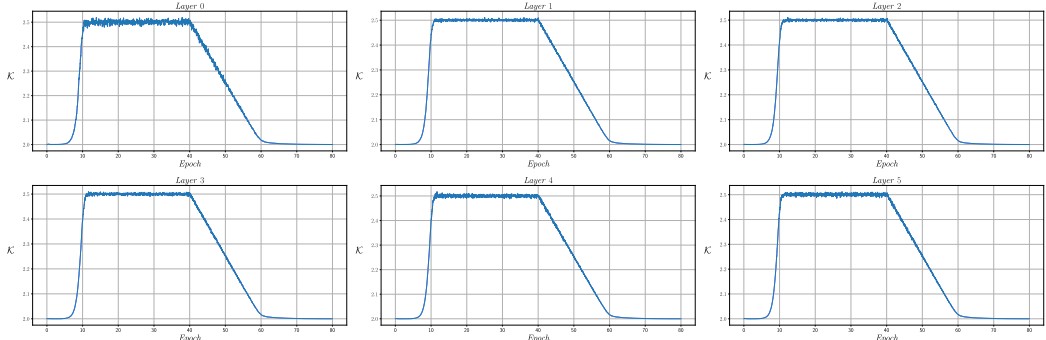

Figure 5: Average number of activated experts $\mathcal{K}$ training dynamic across layers under the three-stage smoothing schedule.

# C  EXPERIMENTAL DETAILS

Before proceeding to the experiments, we establish the choice of coefficients for the log-smoothstep function $h$ defined in Section 6. We have experimented with various values for the coefficients $a$ and $b$, and found that setting $a = 1$ and $b = 50$ provides consistent and effective smoothing behavior across the evaluation. Therefore, we use it for all experiments presented below.

## C.1 LANGUAGE MODELING

### C.1.1 DATASET.

We evaluate our approach on two widely used language modeling benchmarks: WikiText-103 and EnWik-8. The WikiText-103 dataset (Merity et al., 2017b) contains Wikipedia articles with the training set consisting of about 28K articles and 103M tokens in total. The validation and test sets each contain 60 held-out articles, corresponding to 218K and 246K tokens, respectively. The EnWik-8 dataset is a byte-level benchmark derived from a compressed dump of English Wikipedia. It consists of 100 million bytes of data, including not only English text but also markup, special characters, and snippets in other languages. The dataset is split into 90M characters for training, 5M for validation, and 5M for testing.

We follow the experimental setup of Pham et al. (2024) for pretraining on WikiText-103 (Merity et al., 2017a) and EnWik-8 (Mahoney, 2006). For WikiText-103, we report perplexity (PPL) on both validation and test sets. Additionally, we evaluate robustness using the Attacked WikiText-103 dataset constructed by replacing random words with the generic token "AAA" at a rate of 2.5%, following Han et al. (2024); Teo & Nguyen (2024); Abdullaev & Nguyen (2025). For EnWik-8, we evaluate using bits-per-character (BPC) as the primary metric, consistent with prior work on byte-level language modeling.

### C.1.2 IMPLEMENTATION DETAILS.

We employ a standard Switch Transformer (Fedus et al., 2022) as our backbone, with 16 experts and top-2 routing. The model specifications are summarized in Table 6.

Table 6: Backbone specifications for language modeling tasks. All models use 16 experts with top-2 routing.

| Model | SA Layers | FFN Layers | MoE Layers | Att. Span | Embed Size |
|---|---|---|---|---|---|
| Switch Transformer (WikiText-103) | 6 | – | 6 | 1024 | 352 |
| Switch Transformer (EnWik-8) | 8 | – | 8 | 2048 | 352 |

We use the Adam optimizer (Kingma & Ba, 2015) with a base learning rate of $7 \times 10^{-4}$. A linear warmup schedule is applied for 4,000 steps for both models. For WikiText-103, the Switch-medium backbone is trained for 80 epochs with batch size 48. For EnWik-8, the Switch-small backbone is trained for 80 epochs with batch size 48. In all cases, we apply an auxiliary load-balancing loss with weight 0.01 to encourage balanced expert utilization. All models are trained on $2 \times$ NVIDIA H100 80GB GPUs using mixed-precision training.

## C.2 VISION TASK ON DOMAINBED BENCHMARK

### C.2.1 DATASET.

We evaluate on the standard DomainBed benchmark (Gulrajani & Lopez-Paz, 2020), which includes the datasets: PACS (Li et al., 2017), VLCS (Fang et al., 2013), OfficeHome (Venkateswara et al., 2017), TerraIncognita (Beery et al., 2018), and DomainNet (Peng et al., 2019). The statistics of these datasets, including the number of domains, classes, and examples, are summarized in Table 7.

Table 7: Statistics of DomainBed datasets used in our experiments.

| Dataset | PACS | VLCS | OfficeHome | TerraInc | DomainNet |
|---|---|---|---|---|---|
| # Domains | 4 | 4 | 4 | 4 | 6 |
| # Classes | 7 | 5 | 65 | 10 | 345 |
| # Examples | 9,991 | 10,729 | 15,588 | 24,788 | 586,575 |

In detail, the five multi-domain image classification datasets are comprised of:

1. PACS (Li et al., 2017) comprises four domains: art, cartoons, photos, sketches. This dataset contains 9,991 examples of dimension $(3, 224, 224)$ and 7 classes.

2. VLCS (Fang et al., 2013) comprises photographic domains: Caltech101, LabelMe, SUN09, VOC2007. This dataset contains 10,729 examples of dimension $(3, 224, 224)$ and 5 classes.

3. Office-Home (Venkateswara et al., 2017) includes domains: art, clipart, product, real. This dataset contains 15,588 examples of dimension $(3, 224, 224)$ and 65 classes.

4. TerraIncognita (Beery et al., 2018) contains photographs of wild animals taken by camera traps at locations: L100, L38, L43, L46. This dataset contains 24,788 examples of dimension $(3, 224, 224)$ and 10 classes.

5. DomainNet (Peng et al., 2019) has six domains: clipart, infograph, painting, quickdraw, real, sketch. This dataset contains 586,575 examples of size $(3, 224, 224)$ and 345 classes.

We follow the standard DomainBed evaluation protocol using train-domain validation. For each test domain, we train on the remaining domains and use the left-out domain for validation. We select the model maximizing validation accuracy and report the final accuracy on the held-out test domain.

### C.2.2 IMPLEMENTATION DETAILS.

We adopt a ViT-S/16 backbone (Dosovitskiy et al., 2021) pretrained on ImageNet-1K following Li et al. (2023). Images are processed into patch embeddings by ViT-S/16 with a patch size of $16 \times 16$, 6 attention heads, and 12 transformer blocks. Each MoE block contains 6 experts, and the cosine router selects the top-2 experts for each patch. Experts are initialized from the corresponding pretrained ViT blocks, while cosine routers are randomly initialized to ensure even routing at the start.

Training uses the Adam optimizer (Kingma & Ba, 2015) with dataset-specific hyperparameters, as shown in Table 8. Batch size is fixed to 32 per domain. For DomainNet, we train for 15,000 iterations to compare fairly with prior work, while for the other datasets, we train for 5,000 iterations.

Table 8: Hyperparameters for different datasets in DomainBed.

| Dataset | PACS | VLCS | OfficeHome | TerraInc | DomainNet |
|---|---|---|---|---|---|
| Learning Rate | 3e-5 | 3e-5 | 1e-5 | 5e-5 | 5e-5 |
| Weight Decay | 0 | 1e-6 | 1e-6 | 1e-4 | 0 |

### C.3 LANGUAGE TASK (GLUE BENCHMARK)

### C.3.1 DATASET.

We evaluate on a subset of the General Language Understanding Evaluation (GLUE) benchmark (Wang et al., 2018), selecting five representative tasks: CoLA, MRPC, MNLI, QNLI, and RTE. These tasks cover a wide range of linguistic phenomena including grammatical acceptability, paraphrase detection, question answering, and textual entailment. The tasks are briefly summarized as follows:

- CoLA (Corpus of Linguistic Acceptability) (Warstadt et al., 2019): A binary classification task assessing whether a sentence is grammatically acceptable.

- MRPC (Microsoft Research Paraphrase Corpus) (Dolan & Brockett, 2005): A paraphrase identification task determining whether two sentences are semantically equivalent.

- MNLI (Multi-Genre Natural Language Inference) (Xu et al., 2020): A large-scale three-way natural language inference task (entailment, contradiction, neutral) spanning multiple domains.

- QNLI (Question Natural Language Inference) (Wang et al., 2018): A binary classification task derived from the Stanford Question Answering Dataset (SQuAD), reformulated as a sentence pair classification problem.

- RTE (Recognizing Textual Entailment) (Bentivogli et al., 2009): A binary entailment classification task combining several RTE challenges (RTE1–RTE5).

Dataset statistics, including sizes, task types, and domains, are summarized in Table 9.

Table 9: Overview of selected GLUE benchmark tasks. Sizes follow Wang et al. (2018).

| Task | Domain | Train / Dev / Test Size | Task Type | Metric |
|------|--------|-------------------------|-----------|--------|
| CoLA | Miscellaneous | 8.5k / 1k / 1k | Acceptability classification | MCC |
| MRPC | News | 3.7k / 408 / 1.7k | Paraphrase detection | Acc/F1 |
| MNLI | Multi-genre text | 393k / 20k / 20k | Natural language inference | Acc (m/mm) |
| QNLI | Wikipedia QA | 105k / 5.5k / 5.4k | QA/NLI conversion | Accuracy |
| RTE | News/Wikipedia | 2.5k / 276 / 3k | Textual entailment | Accuracy |

We follow the official GLUE evaluation protocols (Wang et al., 2018). Specifically, we use Matthew's correlation coefficient (MCC) for CoLA, accuracy and F1-score for MRPC, matched and mismatched accuracy for MNLI, and accuracy for both QNLI and RTE. Each task is fine-tuned independently, and the best-performing checkpoint on the validation set is used for final test submission. Experiments are repeated with five random seeds, and we report the best validation result for each configuration.

### C.3.2 IMPLEMENTATION DETAILS.

We adopt BERT-large (Devlin et al., 2019b) as the backbone model, augmented with our MoE design. We replace the FFN layer in one Transformer block of BERT-large with an MoE layer containing 16 experts, using top-$k$ routing strategies with $k = 2$ and $k = 4$. To encourage balanced expert utilization, we incorporate the GShard load balancing loss (Lepikhin et al., 2021) with auxiliary loss weight 0.01. We also set gate noise to 1.0 and capacity factor to 1.5 to stabilize routing and mitigate expert overflows.

Fine-tuning is performed with the Adam optimizer (Kingma & Ba, 2015). A grid search over learning rates $\{2 \times 10^{-5}, 3 \times 10^{-5}, 5 \times 10^{-5}\}$ is conducted, while the batch size is fixed at 32. Training is run for up to 10 epochs with early stopping on validation performance. We apply a linear learning rate scheduler. All experiments are executed on NVIDIA H100 80GB GPUs with mixed-precision training. Checkpoints are saved and evaluated every epoch, with the best validation checkpoint retained for testing.

