# OpenReview forum: "Discontinuities in Sparse Mixture-of-Experts: A Measure-Stochastic Analysis"
_ICLR.cc/2026/Conference — ICLR 2026 Conference Desk Rejected Submission_

### Official Review · Reviewer_aFnx · 2025-10-21

**Soundness:** 3
**Presentation:** 2
**Contribution:** 3
**Rating:** 6
**Confidence:** 2

**Summary:**

This paper conducts a rigorous geometric and stochastic analysis of the inherent discontinuities in MoE architectures caused by Top-k routing, classifying these discontinuities by order (based on the number of tied experts) and proving that lower-order (especially order-1) discontinuities dominate in terms of volume and probability under random perturbations. Motivated by this theoretical insight, the authors propose SmoothSMoE, a lightweight smoothing mechanism that softly incorporates experts near discontinuity regions, ensuring the SMoE input-output map becomes continuous while maintaining low computational overhead. Extensive experiments across language modeling (WikiText-103, EnWiki-8), image classification (DomainBed), and natural language understanding (GLUE) tasks demonstrate that SmoothSMoE outperforms vanilla SMoE.

**Strengths:**

1. The theoretical verification is sufficient.

2. Inherent discontinuities in Top-k routing is an interesting topic, which may be related to discussion of hard samples in classification.

**Weaknesses:**

1. Does similar scoring imply that either choice is equally acceptable? If similar scoring imply that either choice is equally acceptable, why is this smoothing mechanism necessary?

2. What is the value of k in the Top-k reported in Table 1 and Table 2?

I am not particularly familiar with the theoretical details, but intuitively, if a token receives similar scores from expert 1 and expert 3, it suggests that assigning it to either expert 1 or expert 3 might make little difference. Based on this assumption, many prior works have proposed dynamic routing strategies. Given that, why is this smoothing mechanism necessary?

**Questions:**

Please refer to Weaknesses.

---

> ### Author Response · Authors · 2025-11-19
> **Rebuttal by Authors**
>
> We thank the Reviewer for the thoughtful comment and constructive feedback. Below, we address the concerns raised in the review.
>
> **W1: Does similar scoring imply that either choice is equally acceptable? If similar scoring imply that either choice is equally acceptable, why is this smoothing mechanism necessary?**
>
> **Answer:** Let us clarify that the goal of our smoothing mechanism is not to decide between “equally acceptable” experts based on similar scores, but to control the behavior of the MoE mapping near switching boundaries. In vanilla SMoE, two inputs that fall on different sides of a score-tied boundary can produce a discontinuous jump in the output, even if their top experts have very similar scores. Our smoothing makes the SMoE map continuous, so outputs vary continuously as inputs cross expert-switching boundaries. As a beneficial side effect, when several experts have similar scores near the top-$k$ cutoff, the mechanism can temporarily allow more of them to contribute, rather than forcing an arbitrary hard tie-break. For a more extensive discussion of whether similar scoring implies that either choice is equally acceptable and the necessity of our smoothing mechanism, please see our answer to W3.
>
> **W2: What is the value of k in the Top-k reported in Table 1 and Table 2?**
>
> **Answer:** For the experiments shown in the Table 1 (WikiText-103) and Table 2 (DomainBed, which is moved to Table 3 in the revised manuscript), we use $k=2$. The Top-$k$ values for all experiments along with other hyperparameters are detailed in Appendix C.
>
> **W3: I am not particularly familiar with the theoretical details, but intuitively, if a token receives similar scores from expert 1 and expert 3, it suggests that assigning it to either expert 1 or expert 3 might make little difference. Based on this assumption, many prior works have proposed dynamic routing strategies. Given that, why is this smoothing mechanism necessary?**
>
> **Answer:** For an intuitive understanding of our theretical analysis, let us first summarize our theoretical contributions in a top-down view:
> (i) We show that a Top-$k$ MoE gate induces a polyhedral partition of the input space into regions where the active expert set is constant, separated by switching (discontinuous) surfaces where logit orderings change.
> (ii) We classify these surfaces by their order (how many experts participate in the tie) and derive volume estimates for the $\epsilon$-thickening around them inside a bounded region where the data effectively lives, with explicit ratios showing that, lower-order thickened boundaries dominate while higher-order ones have vanishing relative volume.
> (iii) Using a random perturbation process modeled by Brownian motion, we obtain explicit bounds on the hitting time of these switching surfaces and on the occupation time spent in their $\epsilon$-thickened neighborhoods.
> In addition to these theoretical results, we propose a simple logit-based local smoothing scheme around the switching surfaces that makes the SMoE map provably continuous with favorable performance.
>
> Intuitively, we agree that when a token receives very similar scores from multiple experts, each of them is a plausible choice. The main problem we target in our work is quite a different one: the instability around expert switching surfaces that arises in vanilla SMoE due to the hard Top-$k$ selection. In this setting, the model inherits sharp switching surfaces (Figure 1 in our paper): as soon as logits cross, the selected expert set changes discretely, and this can induce large jumps in the output even though the underlying expert scores are nearly identical. Our smoothing mechanism is designed to address this instability rather than aiming to answer the question of which expert is the better choice when they have similar scores. It replaces these hard expert switches by a narrow transition region in which the contributions of near-tied experts vary continuously, making the SMoE map itself continuous. At the same time, when several experts have similar scores near the Top-$k$ cutoff, the smoothed gate naturally assigns nonzero mass to multiple plausible experts instead of enforcing an essentially arbitrary hard decision.
>
> In addition to the empirical improvements observed across our experiments when comparing SmoothSMoE to SMoE and ReMoE (added as a new continous routing baseline in Tables 1 and 2 of Section 7 in the revised manuscript), we also include an additional result showing that our method can be used purely as a regularization mechanism. Specifically, we begin training with SmoothSMoE and then gradually turn off smoothing, ending with a standard SMoE router at the end of training. This variant achieves better performance than the SMoE baseline (Table 5 in Appendix B.2 of the revised manuscript), demonstrating that our smoothing mechanism improves training dynamics and final performance.

---

> > ### Author Response · Authors · 2025-11-26
> > **Gentle Reminder and Follow-Up on Rebuttal**
> >
> > This is a gentle reminder regarding our rebuttal. We understand that the review process is demanding and apologize if you were already planning to revisit this review. We are very grateful for the time and care you have devoted to our submission, and for the insightful comments that helped us refine and strengthen our work. We hope that our responses have addressed your concerns; if so, we would kindly ask you to consider increasing your score. Please do not hesitate to reach out if any questions remain, as we would be happy to provide further clarification.

---

### Official Review · Reviewer_rUVp · 2025-10-27

**Soundness:** 4
**Presentation:** 4
**Contribution:** 3
**Rating:** 6
**Confidence:** 4

**Summary:**

This paper tackles the problem of discontinuities in expert weighting in sparse MoEs, which occur where gating logits are tied. A series of theorems bounds the neighborhood volumes of discontinuities of different orders (defined by the number of tied experts). Then a smooth MoE method is proposed that adds nonzero weight for experts having gating logits within $\epsilon$ of the top $k$, making the weights continuous in the inputs. This method shows strong gains over the natural baseline on several tasks.

**Strengths:**

Rigorous and well-written. Although the results on higher-order ties are not surprising (see below), the detailed theorems are a useful contribution.

The proposed smooth MoE is intuitive and a simple modification to existing methods, achieving strong experimental results.

**Weaknesses:**

The result that first-order discontinuities dominate is fairly obvious. (In fact on reading the 5th sentence of the abstract (“We first classify”) my reaction was why bother with the classification since all higher-order ones have negligible probability.) The result about finite hitting time is also unsurprising since Brownian motion hits any hyperplane in finite time with probability 1. Moreover since diffusion per se is not of interest and the real question is how close an input is to a discontinuity, the result is just that any input is a finite distance from a discontinuity.

Lemma A.20 holds only because $T_\epsilon^{(\infty)}$ is defined wrt $z$ space (compared to $T_\epsilon$ which is defined wrt $x$ space). So the fact that distance only needs to be checked in the logits (e.g., line 358) is just a trick of the definition. Moreover what matters in practice, e.g. in an adversarial setting or for gradients under the proposed smoothing method, is distance in $x$ space.

**Questions:**

Do you renormalize the gating weights after adding boundary experts?

Is the smoothing useful at test or does it primarily help by giving more informative gradients during training? What happens to performance if you turn off the smoothing at test (or late in training to let the model fine-tune to the no-smoothing setup)?

---

> ### Author Response · Authors · 2025-11-19
> **Rebuttal by Authors**
>
> We thank the Reviewer for the thoughtful comment and constructive feedback. Below, we address the concerns raised in the review.
>
> **W1: The result that first-order discontinuities dominate is fairly obvious. (In fact on reading the 5th sentence of the abstract (“We first classify”) my reaction was why bother with the classification since all higher-order ones have negligible probability.) The result about finite hitting time is also unsurprising since Brownian motion hits any hyperplane in finite time with probability 1. Moreover since diffusion per se is not of interest and the real question is how close an input is to a discontinuity, the result is just that any input is a finite distance from a discontinuity.**
>
> **Answer:** The reviewer raises three related concerns in this weakness, which we address in turn: (i) The dominance of first-order discontinuities is obvious, (ii) The finite hitting-time result is unsurprising, and (iii) Diffusion is not the right lens compared to simply measuring distance to a discontinuity.
>
> First, for the dominance of first-order discontinuities: while this may seem intuitive on all of $\mathbb{R}^d$, in practice the input distribution is effectively supported in a bounded region $B^D(0,R)$ due to normalization, weight decay, and other forms of implicit regularization. In this setting, Theorem 4.4 makes the analysis precise: we demonstrate the ratio between order-$n$ and order-$m$ $\epsilon$-thickenings depends on the Top-$k$ pattern $J$ via the slice densities $\alpha_{J,n}$ of the top-$k$ induced polyhedra (Lemma A.16–A.17), and we prove that within $B^D(0,R)$ the volume ratio of order-$n$ versus order-$m$ thickened discontinuities scales as $(\epsilon/R)^{n-m}$, making the dependence on $R$ and on the combinatorics of $J$ explicit.
>
> Second, for the hitting-time statement, Theorem 5.1 goes beyond the fact that Brownian motion hits a hyperplane in finite time. It provides an explicit probabilistic bound on the discontinuity boundary hitting time, $\mathbb{P}(\tau_{\mathbb{S}} \le t) \ge 2\bigl(1 - \Phi(r_{\min}/t)\bigr)$, where $r_{\min}$ is computed in closed form from the gating weights $W_g$ and biases $b$. This makes the dependence on the actual MoE parameters explicit, rather than relying only on a generic finite hitting-time argument.
>
> Third, regarding how close in an input to discontinuity and the diffusion formulation. If we consider an arbitrary input $x$, the precise distance from $x$ to the discontinuity can be computed explicitly using the gating weights and biases. The diffusion formulation is introduced to capture random perturbations around $x$ (for instance due to stochastic optimization, noisy inputs), and our results further analyze how a randomly perturbed trajectory starting at $x$ hits the discontinuity set (Theorem 5.1) and how much time it spends in the $\epsilon$-thickened neighborhoods of different discontinuity orders (Theorem 5.3).
>
> **W2: Lemma A.20 holds only because $T_{\epsilon}^{(\infty)}$ is defined wrt $z$ space (compared to $T_{\epsilon}$ which is defined wrt $x$ space). So the fact that distance only needs to be checked in the logits (e.g., line 358) is just a trick of the definition. Moreover what matters in practice, e.g. in an adversarial setting or for gradients under the proposed smoothing method, is distance in $x$ space.**
>
> **Answer:** Most of our theoretical analysis (classification of discontinuity order, Euclidean thickening volumes, hitting and occupation times) is carried out directly in $x$-space, except Theorem 4.7 in $z$-space. We introduce $T_{\epsilon}^{(\infty)}$ when analyzing the smoothing, because for a linear gate $z(x) = W_g x + b$ the logit gaps $z_{[k]}(x) - z_i(x)$ naturally encode distance to the gating hyperplanes in the norm induced by $W_g$. By standard equivalence of norms result in finite-dimensional spaces, working with this induced norm is comparable to working with Euclidean distances in $x$-space up to multiplicative factors. This allows us to detect whether $x$ lies in an neighborhood around the discontinuity set by simply checking conditions of the form $0 < z_{[k]}(x) - z_i(x) < \epsilon$, without explicitly computing distances in $x$-space (which noted can be computed explicitly with the expense of added computation). We use this logit-based notion of distance in practice mainly because it reuses the already-available logits and is computationally effective for our smoothing construction.

---

> ### Author Response · Authors · 2025-11-19
> **Rebuttal by Authors (cont.)**
>
> **Q1: Do you renormalize the gating weights after adding boundary experts?**
>
> **Answer:** Our smoothing is applied directly to the logits $z$ before the softmax, so the softmax automatically renormalizes the gating weights and no additional renormalization step is needed.
>
> **Q2: Is the smoothing useful at test or does it primarily help by giving more informative gradients during training? What happens to performance if you turn off the smoothing at test (or late in training to let the model fine-tune to the no-smoothing setup)?**
>
> **Answer:** This is an interesting question, since smoothing allows more experts near routing boundaries to contribute and yields a continuous routing map, which can make the loss landscape more amenable to optimization; we therefore hypothesize that it improves training dynamics and boosts final performance. To test this hypothesis, we use a three-stage schedule on WikiText-103. For the first $40$ epochs, we set $k^{\*} = 2.5$ to warm up the model and allow extra experts near switching surfaces. Over the next $20$ epochs, we linearly anneal $k^{*}$ from $2.5$ to $2$ to gradually turn off the smoothing, and in the final $20$ epochs we fix $k^{\*} = 2$, which turns off smoothing and lets the model fully adapt to hard Top-$2$ gating. This schedule directly tests how much of the benefit comes from smoother optimization during training. Results in Table 5 (Section B.5, revised manuscript) show that even after turning off boundary smoothing at the end, the training landscape induced by SmoothSMoE leads to an improved SMoE model with higher performance.

---

> > ### Comment · Reviewer_rUVp · 2025-11-24
> >
> > I appreciate the mathematical detail in the formal results which goes beyond my characterization. I still think those details are ancillary to the paper's main argument but that isn't a big criticism.
> >
> > Apologies for the dumb renormalization question.
> >
> > Thanks for the new experiment turning off smoothing after initial training. It's nice to see that prediction confirmed.

---

> ### Author Response · Authors · 2025-11-24
> **Reply to Reviewer rUVp**
>
> We wholeheartedly thank you for your thoughtful follow-up and for engaging closely with both the theoretical and experimental aspects of the paper. The discussions made it clear that more accessible explanations of the formal results would substantially improve readability for a broader audience, so we have further revised Section B.1 to incorporate several of the key geometric and analytic intuitions that emerged from our discussion. We are also glad to hear that the new experiment where smoothing is turned off after initial training addresses your question. It provides an interesting and concrete application we had previously overlooked, and we are grateful for your prompting us to explore this direction.

---

> > ### Author Response · Authors · 2025-11-25
> > **Reply to Reviewer rUVp (cont.)**
> >
> > We greatly appreciate your engagement and constructive feedback. If our revisions and clarifications satisfactorily address the concerns you raised, we kindly hope that this may be taken into consideration when adjusting the score. We remain open to further discussion in the rebuttal phase.

---

### Official Review · Reviewer_VNBy · 2025-10-30

**Soundness:** 3
**Presentation:** 3
**Contribution:** 2
**Rating:** 4
**Confidence:** 2

**Summary:**

This paper presents a rigorous theoretical analysis of the discontinuities inherent in Sparse Mixture-of-Experts (SMoE) models that utilize Top-k gating. The authors' contributions can be categorized into three main areas. First, from a geometric perspective, they classify discontinuities by an "order" corresponding to the number of simultaneously tied expert scores at the selection boundary. Using measure-theoretic arguments, they demonstrate that the $\epsilon$-thickened volume of these discontinuity surfaces decays as $(\epsilon/R)^n$ for an order-n discontinuity, implying that lower-order (simpler) discontinuities are geometrically dominant. Second, from a stochastic perspective, they model random input perturbations as an Itô diffusion process. They prove that such a process, starting within a region of a fixed expert set, will almost surely hit a boundary, and that this first hit will occur on an order-1 discontinuity. They further provide bounds on the occupation time near these surfaces, again showing that the process spends less time near higher-order discontinuities. Finally, motivated by these theoretical insights, they propose a practical and efficient smoothing mechanism called SmoothSMoE. This method applies a localized smoothing to non-top-k expert logits that fall within an $l_{\infty,\epsilon}$ distance of the k-th logit, effectively enforcing continuity while incurring minimal overhead. The authors validate their method with experiments on language modeling, image classification, and natural language understanding tasks, showing modest but consistent improvements over the vanilla SMoE baseline

**Strengths:**

- The core strength of the paper is its deep and formal analysis of SMoE discontinuities. The use of measure theory to quantify the size of different orders of discontinuities and the stochastic analysis of hitting times are highly original and provide insights into the behavior of these models.
- The paper clearly identifies a fundamental problem (discontinuities from Top-k gating) and proposes a solution (local smoothing) directly motivated by a rigorous analysis of that problem.
- The proposed SmoothSMoE is simple, requires no re-training from scratch, and is computationally efficient. The adaptive loss for tuning the hyperparameter $\epsilon$ makes the method more practical.

**Weaknesses:**

- The authors mention several related works on differentiable or smooth MoE routing in the introduction (e.g., SMEAR, Soft MoE, ReMoE) but dismiss them on conceptual grounds (e.g., breaking causality for generation, requiring costly retraining). However, they fail to provide any direct empirical comparisons, even on tasks where those methods are perfectly applicable, such as the GLUE benchmark or DomainBed classification.
- The reported improvements, while consistent, are quite small across the board (e.g., ~1-1.5 PPL on WikiText-103, ~0.9% average accuracy on DomainBed, <0.5% on GLUE). While theoretical justification can make small gains more compelling, the authors should be more circumspect in their claims. The lack of error bars or statistical significance testing further weakens the empirical claims. Are these small gains reproducible or just noise from a single run?

**Questions:**

- Can the authors justify the omission of experimental comparisons to other relevant smoothing methods like Soft MoE on the non-autoregressive tasks (DomainBed, GLUE)?

- The log-smoothstep function seems somewhat arbitrary. Did you experiment with other smoothing functions (e.g., based on splines, sigmoids)? Is there any part of your theoretical analysis that suggests this specific functional form is optimal or preferable in some way?

- Could you please provide standard deviations across multiple runs for your key results in Tables 1, 2, and 4?

---

> ### Author Response · Authors · 2025-11-19
> **Rebuttal by Authors**
>
> We thank the Reviewer for the thoughtful comment and constructive feedback. Below, we address the concerns raised in the review.
>
> **W1: The authors mention several related works on differentiable or smooth MoE routing in the introduction (e.g., SMEAR, Soft MoE, ReMoE) but dismiss them on conceptual grounds (e.g., breaking causality for generation, requiring costly retraining). However, they fail to provide any direct empirical comparisons, even on tasks where those methods are perfectly applicable, such as the GLUE benchmark or DomainBed classification.**
>
> **Q1: Can the authors justify the omission of experimental comparisons to other relevant smoothing methods like Soft MoE on the non-autoregressive tasks (DomainBed, GLUE)?**
>
> **Answer W1 + Q1:** Regarding the reviewer’s concern, we compare our method with ReMoE [1], a recent continuous routing approach with sparsely activated experts. Experiments are conducted on GLUE for the non-autoregressive task and WikiText-103 for the autoregressive task. The results (In table 1 and 2 section 7) show that ReMoE slightly outperforms SMoE, while our SmoothSMoE further improves over ReMoE under comparable settings.
>
> **W2: The reported improvements, while consistent, are quite small across the board (e.g., ~1-1.5 PPL on WikiText-103, ~0.9% average accuracy on DomainBed, <0.5% on GLUE). While theoretical justification can make small gains more compelling, the authors should be more circumspect in their claims. The lack of error bars or statistical significance testing further weakens the empirical claims. Are these small gains reproducible or just noise from a single run?**
>
> **Q3: Could you please provide standard deviations across multiple runs for your key results in Tables 1, 2, and 4?**
>
> **Answer W2 + Q3:** To address the reviewer’s concern about the lack of error bars, we run all comparative experiments with multiple random seeds and report mean performance along with standard deviation for each run in the revised manuscript. Specifically, we use $5$ random seeds for GLUE and DomainBed, and $3$ random seeds for WikiText-103 language modeling (due to the high computational cost of language modeling training from scratch). The updated results are reported in Tables 1, 2, and 3 of the revised manuscript. These results show that the gains remain consistent across runs, indicating that the improvements are clearly reproducible rather than noise.
>
> In addition to these strengthened empirical results, we include an ablation showing that our method can also be used as a regularization mechanism. We begin training with SmoothSMoE and gradually turn off smoothing, ending with a standard SMoE router at the end of training. This annealed variant achieves better performance than the SMoE baseline (Table 5 in Appendix B.2 of the revised manuscript), which suggests that the smoothing mechanism improves both training dynamics and final performance.
>
> **Q2: The log-smoothstep function seems somewhat arbitrary. Did you experiment with other smoothing functions (e.g., based on splines, sigmoids)? Is there any part of your theoretical analysis that suggests this specific functional form is optimal or preferable in some way?**
>
> **Answer:** We choose log-smoothstep because our  smoothing function works directly in logit space: we add a mask $m(z)$ to each gate logit so that $m(z)$ decays smoothly from $0$ to $-\infty$ as $z$ moves from $z_{[k]}$ down to $z_{[k]} - \epsilon$, effectively turning non top-$k$ experts off outside the $\epsilon$-tube (see Fig. 2 in Appendix A.2). A natural candidate function is $h(u) = \log(u)$ with $u = \frac{z_i(x) - z_{[k]}(x) + \epsilon}{\epsilon}$, but this does not allow us to control the decay rate. We therefore use the more flexible family $h(u) = \log \big(\frac{u^a}{u^a + (1-u)^b}\big)$, which has the same behavior, recovers $\log(u)$ as a special case when $a = b = 1$, and lets us tune how sharply the mask decays via $(a,b)$. Standard spline or sigmoid functions do not reach $-\infty$ at finite input, so they cannot zero out weighting score in the same way from logit space. We do not claim this particular form is optimal; our analysis only requires a monotone $C^\infty$ mask with the right limiting behavior, and the proposed log-smoothstep family is a convenient and natural choice that satisfies these conditions.
>
> [1] Ziteng Wang, Jun Zhu, and Jianfei Chen. ReMoE: Fully Differentiable Mixture-of-Experts with ReLU Routing. International Conference on Learning Representations, 2025.

---

> > ### Author Response · Authors · 2025-11-26
> > **Gentle Reminder and Follow-Up on Rebuttal**
> >
> > This is a gentle reminder regarding our rebuttal. We understand that the review process is demanding and apologize if you were already planning to revisit this review. We are very grateful for the time and care you have devoted to our submission, and for the insightful comments that helped us refine and strengthen our work. We hope that our responses have addressed your concerns; if so, we would kindly ask you to consider increasing your score. Please do not hesitate to reach out if any questions remain, as we would be happy to provide further clarification.

---

### Official Review · Reviewer_ue9P · 2025-10-31

**Soundness:** 3
**Presentation:** 3
**Contribution:** 2
**Rating:** 6
**Confidence:** 3

**Summary:**

This paper presents a rigorous geometric and stochastic characterization of discontinuities in Sparse Mixture-of-Experts (SMoE) models caused by Top-$k$ gating.
The authors:
- Classify discontinuities by order (number of tied experts) and derive asymptotic volume bounds for their ϵ-thickened neighborhoods using measure-theoretic slicing arguments.
- Analyze stochastic dynamics of random perturbations near these boundaries, proving that diffusion paths almost surely hit order-1 discontinuities first, with explicit hitting-time and occupation-time bounds.
- Propose a smoothing mechanism—$\ell_{\infty, \epsilon}$-local smoothing—that restores continuity in SMoEs by softly activating near-boundary experts.
Empirical results across language (WikiText-103, EnWiki-8, GLUE) and vision (DomainBed) tasks demonstrate measurable robustness and accuracy gains.

**Strengths:**

- Foundational theory: Provides the first quantitative geometry of MoE discontinuities.
- Stochastic insight: Connects random perturbation dynamics to boundary order and frequency.
- Elegant theorems: Clean scaling laws (Thm. 4.4, 4.7) and finite-time hitting bounds.
- Actionable smoothing: $\ell_{\infty, \epsilon}$-smoothing has provable continuity and empirical benefit.
- Cross-domain validation: Demonstrated on both NLP and vision benchmarks.

**Weaknesses:**

- High mathematical density: May deter non-theory readers; geometric intuition could be expanded.
- Scope of stochastic model: Assumes isotropic Brownian motion—structured or adversarial noise remains open.
- Empirical analysis limited: Smoothing ablation on very large LLMs (e.g., >7B) not shown.
- Adaptive $\epsilon$ mechanism: Boundary loss behavior could be theoretically analyzed beyond heuristics.
- No direct link to differentiable routing methods: A comparative theoretical discussion (SoftMoE, ReMoE) would strengthen positioning.

**Questions:**

- How sensitive are asymptotic ratios (Theorem 4.4) to non-affine gating networks or nonlinear activations?
- Does the stochastic hitting analysis extend to anisotropic σ or multiplicative noise processes?
- Is the occupation-time bound tight in the small-$\epsilon$ limit, or could there be sharper constants?
- In adaptive ϵ optimization, is $\mathcal{L}_{\text{boundary}}$ stable under mini-batch estimation noise?
- Can the smoothing be viewed as a local convex relaxation of Top-$k$ gating?
- How does smoothing interact with expert load balancing in large-scale training?
- Could discontinuity order distribution be empirically estimated in trained models?
- Are there guarantees of gradient continuity for backprop through $\ell_{\infty, \epsilon}$ smoothstep?
- Would adversarially aligned perturbations (not Brownian) still almost surely hit order-1 boundaries?
- Could these geometric insights inform new regularizers for MoE fairness or specialization?

⸻

---

> ### Author Response · Authors · 2025-11-19
> **Rebuttal by Authors**
>
> We thank the Reviewer for the thoughtful comment and constructive feedback. Below, we address the concerns raised in the review.
>
> **W1: High mathematical density: May deter non-theory readers; geometric intuition could be expanded.**
>
> **Answer:** To better convey the geometric intuition behind our theory and smoothing mechanism, we have added a dedicated discussion in Section B.1 of the revised manuscript.
>
> **W2: Scope of stochastic model: Assumes isotropic Brownian motion—structured or adversarial noise remains open.**
>
> **Q2: Does the stochastic hitting analysis extend to anisotropic $\sigma$ or multiplicative noise processes?**
>
> **Answer W2 + Q2:** Our theoretical results already cover anisotropic diffusion matrices $\sigma \in \mathbb{R}^{d \times d}$ and only require nondegeneracy, as in Theorem 5.1. For multiplicative noise processes of the form $dX_t = \sigma(X_t, t) dB_t$, each coordinate is a continuous local martingale and can be represented as a time-changed Brownian motion via the Dubins–Schwarz theorem. Under mild regularity and nondegeneracy assumptions on $\sigma(X_t, t)$, this can guarantee that discontinuity hyperplanes are hit in finite time, although obtaining explicit bounds on the hitting time becomes more involved due to the involvement of $X_t$ and would require a finer analysis beyond the scope of the current paper.
>
> **W3: Empirical analysis limited: Smoothing ablation on very large LLMs (e.g., >7B) not shown.**
>
> **Answer:** Due to limited academic compute resources, full training ablations on very large LLMs are beyond our current scope, so we strengthen the empirical analysis in two complementary ways: (i) In Section B.5, we add an experiment where SmoothSMoE is used as a training-time regularizer on Wikitext-103 task and smoothing is gradually turned off to obtain a vanilla top-$k$ SMoE, showing that smoothing makes the loss landscape more amenable to optimization and improves performance; (ii) we include ReMoE [1] as an recent differentiable-routing baseline in our rebuttal experiments; across WikiText-103 and GLUE, SmoothSMoE consistently outperforms both SMoE and ReMoE.
>
> **W4: Adaptive mechanism: Boundary loss behavior could be theoretically analyzed beyond heuristics.**
>
> **Answer:** By construction, the boundary loss satisfies $\frac{\partial L_{\text{boundary}}}{\partial \epsilon} = \alpha(\mathcal{K} - k^{\*})$ where $\mathcal{K}$ is the average number of active experts and $k^{\*}$ is the target budget. If we update $\epsilon$ by gradient descent with learning rate $\eta$, and denote by $\epsilon_n$ the value of $\epsilon$ after $n$-update step, then $\epsilon_{n+1} = \epsilon_n - \eta \frac{\partial L_{\text{boundary}}}{\partial \epsilon} =  \epsilon_n - \eta \alpha(\mathcal{K} - k^{\*})$, which decreases when $\mathcal{K} > k^{\*}$ and increases when $\mathcal{K} < k^{\*}$. Thus the boundary loss implements a simple feedback mechanism that adapts $\epsilon$ to keep the average number of active experts close to the budget $k^*$. Since $\mathcal{K}$ increases with $\epsilon$ but generally does not admit a closed form, a more detailed theoretical characterization of the relation between $\mathcal{K}$ and $\epsilon$ is an interesting direction for future work.
>
> **W5: No direct link to differentiable routing methods: A comparative theoretical discussion (SoftMoE, ReMoE) would strengthen positioning.**
>
> **Answer:** We agree that an explicit theoretical comparison is helpful and have added a discussion of the theoretical differences between SmoothSMoE and other differentiable routing method in Section B.2 of the revised manuscript. We have also added an empirical comparison with ReMoE in our rebuttal, showing that SmoothSMoE outperforms both SMoE and ReMoE on WikiText-103 language modeling and GLUE benchmarks.
>
> **Q1: How sensitive are asymptotic ratios (Theorem 4.4) to non-affine gating networks or nonlinear activations?**
>
> **Answer:** The asymptotic ratios in our proof are obtained by evaluating the integral of $\epsilon$-thickening volume of the union of all discontinous hyperplanes, hence the proof techniques can be extend to piecewise linear like ReLU gating with finer analysis. For nonlinear activation, the description of the resulting manifold is more complex and is an challenging theoretical problem.
>
> **Q3: Is the occupation-time bound tight in the small-$\epsilon$ limit, or could there be sharper constants?**
>
> **Answer:** Proposition A.31 shows that for a single codimension-$n$ flat the occupation time scales as $\mathbb{E}[A^{(n)}_\epsilon(T;S)] \sim C \epsilon^n$ as $\epsilon\to 0$, and is tight with small-$\epsilon$ scaling. In principle, the numerical constants in Theorem 5.3 can be further refined by replacing the supremum estimate on the Gaussian density and tracking overlaps between $\epsilon$-tubes more carefully, such refinements would only affect lower-order terms and not the leading $\epsilon^n$ behavior that we use in the paper.

---

> ### Author Response · Authors · 2025-11-19
> **Rebuttal by Authors (cont.)**
>
> **Q4: In adaptive $\epsilon$ optimization, is $L_{boundary}$ stable under mini-batch estimation noise?**
>
> **Answer:** Empirically, the adaptive scheme is stable. During training we log the learned threshold $\epsilon$ and the average number of activated experts per token (Fig. 4 in  Appendix B.4). Both quantities evolve in a controlled manner over time, and the average number of active experts stays close to the target $k^*$, indicating that the boundary loss provides a robust control signal even under mini-batch noise.
>
> **Q5: Can the smoothing be viewed as a local convex relaxation of Top-k gating?**
>
> **Answer:** This is an interesting point. Our construction can indeed be interpreted as a local convex relaxation of the discrete Top-k mask. Standard Top-k routing can be written as applying a binary mask $s_i(x) \in \\{0,1\\}$ to the gate score and then normalizing. With log-smoothstep, we effectively replace these indicators by continuous coefficients $\tilde s_i(x) \in [0,1]$ inside an $\ell_{\infty, \epsilon}$-neighborhood of the Top-k boundary. Since $[0,1]^M$ is the convex hull of $\\{0,1\\}^M$, this can be viewed as a convex relaxation of the discrete selection variables, applied only near discontinuities. However, we do not derive $\tilde s_i(x)$ as the optimizer of a specific convex program; our design objective is continuity and a bounded number of extra experts rather than a particular convex surrogate.
>
> **Q6: How does smoothing interact with expert load balancing in large-scale training?**
>
> **Answer:** In all our experiments, SmoothSMoE is trained with the same load balancing loss as the underlying SMoE. The load balancing term is applied to the final active expert set after smoothing, meaning it sees the original Top-$k$ experts together with any additional experts inside the $\ell_{\infty,\epsilon}$ smoothing band, and all of these active experts are treated identically. Designing variants that weight experts in the smoothing band according to their smooth coefficients, or that more tightly couple smoothing and load balancing, is an interesting direction for future work.
>
> **Q7: Could discontinuity order distribution be empirically estimated in trained models?**
>
> **Answer:** One could construct a Monte Carlo estimator by sampling points in input space and, for each sample, determining the discontinuity order it belongs to from the pattern of score ties and near-ties. However, this requires exploring a high-dimensional region in $\mathbb{R}^d$ and repeatedly evaluating and sorting expert scores, which can becomes computationally expensive.
>
> **Q8: Are there guarantees of gradient continuity for backprop through smoothstep?**
>
> **Answer:** Our formal result (Prop. A.7) guarantees continuity of the forward map, moreover from the construction we also obtain almost-everywhere (a.e.) gradient continuity. The smoothed gate is a composition of affine logits $z_i(x)$, the $k$-th order statistic $z_{[k]}(x)$, the log-smoothstep $h$, and a softmax. The softmax is $C^\infty$, and $h$ is $C^\infty$ on the interior of its active region $0 < u < 1$ and constant elsewhere. Away from (i) exact logit ties (where two $z_i$ coincide) and (ii) the boundary of the $\varepsilon$-strip (where $u \in \\{0,1\\}$), the order statistic $z_{[k]}(x)$ coincides with a single affine logit, so the whole gate is a $C^\infty$ composition whose gradient varies continuously. The only regions where gradient discontinuity arises from gating are these tie and strip-boundary surfaces, which form finite unions of affine hyperplanes and hence have Lebesgue measure zero. Thus the smoothed SMoE has gradient that is continuous a.e., which is compatible with backpropagation and the usual assumptions made in practice.
>
> **Q9: Would adversarially aligned perturbations (not Brownian) still almost surely hit order-$1$ boundaries?**
>
> **Answer:** With aligned perturbations, for example a deterministic process $dX_t = b(X_t, t) dt$, the first boundary hit is no longer guaranteed to be of order-$1$. However, we conjecture that the result can be extended to adapted stochastic processes of the form $dX_t = b_t dt + \sigma_t dB_t$ under nondegeneracy conditions on $b_t$ and $\sigma_t$ due to the fluctuation of the diffusion component. When this conjecture holds, it can further be extended to general continuous semimartingales (the class of well-behaved continuous perturbation processes) using the Doob–Meyer decomposition and the martingale representation theorem, thus can be an interesting theoretical work.

---

> ### Author Response · Authors · 2025-11-21
> **Rebuttal by Authors (cont.)**
>
> **Q10: Could these geometric insights inform new regularizers for MoE fairness or specialization?**
>
> **Answer:** The geometric picture suggests several ways to design fairness and specialization regularizers: for fairness, one can penalize experts whose assigned tokens concentrate in unstable $\epsilon$-tubes near top-$k$ score-tied boundaries, so that balanced usage reflects genuinely decisive assignments rather than boundary noise; for specialization, a low-overlap penalty can discourage situations where many experts are simultaneously near top-$k$ on the same tokens, promoting more distinct expert regions. In the same spirit, our additional experiment in Section B.5 treats SmoothSMoE as a geometry-aware training-time regularizer, training with smoothing active and then annealing it away to recover a vanilla top-$k$ SMoE, which improves performance and provides concrete evidence that geometry-motivated regularization is beneficial to SMoE training.
>
> [1] Ziteng Wang, Jun Zhu, and Jianfei Chen. ReMoE: Fully Differentiable Mixture-of-Experts with ReLU Routing. International Conference on Learning Representations, 2025.

---

> > ### Author Response · Authors · 2025-11-26
> > **Gentle Reminder and Follow-Up on Rebuttal**
> >
> > This is a gentle reminder regarding our rebuttal. We understand that the review process is demanding and apologize if you were already planning to revisit this review. We are very grateful for the time and care you have devoted to our submission, and for the insightful comments that helped us refine and strengthen our work. We hope that our responses have addressed your concerns; if so, we would kindly ask you to consider increasing your score. Please do not hesitate to reach out if any questions remain, as we would be happy to provide further clarification.

---

### Author Response · Authors · 2025-11-19
**General Response and Summary of Revisions**

Dear Reviewers,

Thank you for your thoughtful reviews and valuable feedback. We are encouraged by the unanimously positive reception of our theoretical contributions: Reviewer ue9P highlights the foundational theory, stochastic insights, and elegance of the scaling law; Reviewer VNBy comments on the high originality and depth of our analysis of SMoE discontinuities and the practicality of our method; Reviewers rUVp and aFnx emphasize the rigor and sufficiency of our theoretical results.

The main concerns raised by the Reviewers are (i) the high mathematical density of the paper may deter non-theory readers, (ii) the comparison with other differentiable routing method, and (iii) the statistical significance of the empirical results. We have revised the manuscript based on these feedback, and all new or revised content is highlighted in blue. We summarize here the main changes in the revised paper:

1. We added a new Section B.1 to provide a more intuitive, geometry-based explanation of our theoretical analysis for non-theory readers.

2. We added a new Section B.2 to discuss the theoretical differences between SmoothSMoE and other differentiable routing methods.

3. We included ReMoE [1], a recent differentiable routing method as a baseline on WikiText-103 language modeling and GLUE benchmark, and reported all comparison experiments with mean and standard deviation over multiple random seeds.

4. We added an experiment on Wikitext-103 language modeling in Section B.5, demonstrating that SmoothSMoE improves training dynamic and boost performance, and that the improvement is maintained when the smoothing is gradually turned off, yielding an efficient Top-$k$ SMoE at the end of training.

[1] Ziteng Wang, Jun Zhu, and Jianfei Chen. ReMoE: Fully Differentiable Mixture-of-Experts with ReLU Routing. International Conference on Learning Representations, 2025.

---

### Author Response · Authors · 2025-11-26
**Finalized Rebuttal and Additional Results**

Dear Reviewers and Chairs,

Thank you once again for your thoughtful reviews and valuable feedback.

We have now finalized and posted our rebuttal. In response to the reviews, we have added further experimental results and made additional revisions to address the concerns raised. These updates have been incorporated into the revised manuscript and are summarized in "Summary of Revisions".

We would be grateful if you could let us know whether our responses adequately address your concerns, or if there are any remaining questions regarding our submission or rebuttal.

We are happy to provide any further clarification or engage in additional follow-up discussion as needed.

Best regards,

Authors

---

### Author Response · Authors · 2025-12-03
**Submission Briefings for the New AC and Thank You for Overseeing Our Submission**

Dear AC,

Thank you very much for stepping in during this difficult stage of the review process and taking over the handling of our submission. We sincerely appreciate your support.

To make it easier for you to quickly grasp the current status of the paper, we have prepared three short briefings, summarized below and provided in full in the subsequent messages:

- Summary of Additional Theoretical and Empirical Results: On the theoretical side, we added two new sections (B.1 and B.2) in the appendix: Section B.1 provides an intuitive, geometry-based explanation of our analysis for non-theory readers, and Section B.2 discusses the theoretical differences between SmoothSMoE and other differentiable routing methods. On the empirical side, we further added ReMoE as a recent continuous-routing baseline and show that SmoothSMoE achieves superior performance on WikiText-103 and GLUE, while also reporting mean and standard deviation for all benchmark experiments. We added a new experiment in which SmoothSMoE is used purely as a regularization mechanism and confirmed the hypothesis that it improves both training dynamics and final performance.

- Summary of Main Reviewer Concerns and Our Replies: (i) Reviewer ue9P asked us to improve accessibility for non-theory readers and to include a theoretical comparison with other continuous-routing methods; we addressed this by adding Sections B.1 and B.2. (ii) Reviewers VNBy and ue9P asked for strenghthen empirical analysis and statistical significance; we responded by including ReMoE as a new baseline as suggested, reporting standard deviations across multiple seeds for all key experiments, and adding the new regularization/annealing experiment. (iii) Reviewer rUVp asked for clarification of the theoretical contribution and for the regularization experiment; we addressed each point in detail, and in a follow-up comment the reviewer explicitly expressed appreciation for the mathematical detail and confirmed that the new experiment matched their expectations. (iv) Reviewer aFnx questioned whether similar expert scores make the choice of expert essentially indifferent and whether smoothing is necessary; we clarified that SmoothSMoE is designed to address discontinuous instability empirically observed in hard Top-$k$ SMoEs when experts swap, and we highlighted empirical advantages of our method across multiple tasks.

- Summary of Our Key Contributions: Our work provides, to our knowledge, the first extensive and rigorous analysis of discontinuity regions in vanilla SMoEs from both geometric/measure-theoretic and stochastic perspectives. Building directly on this analysis, we propose an efficient smoothing mechanism that makes the SMoE mapping provably continuous and yields consistent empirical improvements over vanilla SMoE and recent continuous-routing baselines across a range of tasks and settings.

We submitted our initial general response and summary of revisions on 20 November 2025, as well as several subsequent replies and gentle reminders to the reviewers. Reviewers ue9P, VNBy, rUVp, and aFnx provided initial reviews; after the rebuttal, only Reviewer rUVp posted a follow-up comment, explicitly appreciating the mathematical contributions and confirming that the new regularization experiment validated their prediction. Unfortunately, the discussion period ended early, and despite several reminders we were unable to continue the discussion or receive updated scores from the other reviewers.

We would be very grateful if you could carefully read our paper, rebuttal, and further replies with additional results, and take into account that we have addressed all of the reviewers’ concerns. We believe the reviewers would likely have updated their evaluations had the discussion continued. We trust that, with your careful review, our submission will be assessed fairly and accurately by the AC, SAC, and PC. Please let us know if you have any questions about the paper or rebuttal; we are more than happy to provide further clarification or engage in additional discussion.

Best regards,

Authors

---

### Note · Program_Chairs · 2026-01-17
**Submission Desk Rejected by Program Chairs**

The following references in this submission do not refer to real documents and/or have major errors in bibliographic information:

 Xinyu Qiu, Yufan Zhang, Rong Zhu, Zhuosheng Zhang, Yang Liu, and Hai Zhao. Emoe: Efficient mixture-of-experts for large language models. In Findings of the Association for Computational Linguistics: ACL, pp. 13944-13957, 2023.